# Optimal Algorithms for Decentralized Stochastic Variational Inequalities

**Dmitry Kovalev**
KAUST,[*] Saudi Arabia
dakovalev1@gmail.com

**Aleksandr Beznosikov**
MIPT,[†] HSE University and Yandex, Russia
anbeznosikov@gmail.com

**Abdurakhmon Sadiev**
MIPT, Russia
sadiev.aa@phystech.edu

**Michael Persiianov**
MIPT, Russia
persiianov.mi@phystech.edu

**Peter Richtárik**
KAUST, Saudi Arabia
peter.richtarik@kaust.edu.sa

**Alexander Gasnikov**
MIPT, HSE University and IITP RAS,[‡] Russia
gasnikov@yandex.ru

## Abstract

Variational inequalities are a formalism that includes games, minimization, saddle point, and equilibrium problems as special cases. Methods for variational inequalities are therefore universal approaches for many applied tasks, including machine learning problems. This work concentrates on the decentralized setting, which is increasingly important but not well understood. In particular, we consider decentralized stochastic (sum-type) variational inequalities over fixed and time-varying networks. We present lower complexity bounds for both communication and local iterations and construct optimal algorithms that match these lower bounds. Our algorithms are the best among the available literature not only in the decentralized stochastic case, but also in the decentralized deterministic and non-distributed stochastic cases. Experimental results confirm the effectiveness of the presented algorithms.

## 1 Introduction

Variational inequalities are a broad and flexible class of problems that includes minimization, saddle point, Nash equilibrium, and fixed point problems as special cases; see [24, 6] for an introduction. Over the long history of modern research on variational inequalities spanning at least half a century, the community developed their own methods and theory, differing from the approaches in their sister field, optimization. The ExtraGradient / MirrorProx methods due to [38, 52, 34] have a similar foundational standing in the variational inequalities field that gradient descent occupies in the optimization literature. As in the case of gradient descent, many modifications [31] and variants [66] of these methods were proposed and studied in the variational inequalities literature.

---

[*]King Abdullah University of Science and Technology
[†]Moscow Institute of Physics and Technology
[‡]Institute for Information Transmission Problems RAS

36th Conference on Neural Information Processing Systems (NeurIPS 2022).

## 1.1 Applications of variational inequalities

In recent years, there has been a significant increase of research activity in the study of variational inequalities due to new connections to reinforcement learning [58, 32], adversarial training [44], and GANs [28]. In particular, [21, 26, 47, 20, 42, 60] show that even if one considers the classical (in the variational inequalities literature) regime involving monotone and strongly monotone inequalities, it is possible to obtain insights, methods and recommendations useful for the GAN community.

In addition to the above modern applications, and besides their many classical applications in applied mathematics that include economics, equilibrium theory, game theory and optimal control [23], variational inequalities remain popular in supervised learning (with non-separable loss [33]; with non-separable regularizer [3]), unsupervised learning (discriminative clustering [69]; matrix factorization [4]), image denoising [22, 18], robust optimization [9], and non-smooth optimization via smooth reformulations [56, 53].

## 1.2 Processing on the edge

With the proliferation of mobile phones, wearables, digital sensors, smart home appliances, and other devices capable of capturing, storing and processing data, there is an increased appetite to mine the richness contained in these sources for the benefit of humanity. However, at the same time, the traditional centralized approach relying on moving the data into a single proprietary warehouse for processing via suitable machine learning methods is problematic, and a new modus operandi is on the rise: processing the data at the source, on the edge, where it was first captured and where it is stored [37, 46, 35], by the client's devices that own the data. There are many reasons for a gradual shift in this direction, including energy efficiency and data privacy.

A key necessary characteristic for any viable algorithmic approach to work in such a massively decentralized regime is the ability to support decentralized processing reflecting the fact that the devices are connected through a network of a potentially complicated topology, possibly varying in time. A central authority may be absent in such a system, and the methods need to rely on communication patters that correspond to the existing connection links.

## 1.3 Decentralized algorithms for variational inequalities

In this paper we study

*algorithms for solving variational inequalities over decentralized communication networks.*

In this regime, a number of nodes (workers, devices, clients) are connected via a communication network, represented by a graph. Each node can perform computations using its local state and data, and is only allowed to communicate with its neighbors in the graph.

Decentralized algorithms over fixed communication networks find their applications in sensor networks [61], network resource allocation [8], cooperative control [27], distributed spectrum sensing [7], power system control [25] and, of course, in machine learning [63]. Recently, decentralized methods over time-varying networks have gained particular popularity due to their relevance to federated learning [37, 35], where communication failures between devices are a common problem.

Decentralized minimization methods are well studied [49, 36]. In particular, lower bounds and optimal algorithms for such problems are known in the *fixed* [63, 30, 40] and *time-varying* [39, 41] network topology regimes.

*However, in significantly more general and hence potentially much more impactful formalism of variational inequalities, the question of optimal and efficient decentralized methods is still open.*

Motivated by these considerations, our work is devoted to advancing the algorithmic and theoretical foundations of decentralized variational inequalities, in both the fixed and time-varying network regimes.

## 1.4 Our contributions and related work

We now briefly summarize our main contributions.

**(a) Lower bounds**

We present the **first lower bounds for the communication and local computation complexities of decentralized variational inequalities in the stochastic (finite sum) case, in both the fixed and time-varying network topology regimes.** See Table 1.

Existing literature contains lower bounds for *non-distributed* finite-sum variational inequalities [29], which we recover as a special case. Existing literature also contains lower bounds for *deterministic* decentralized variational inequalities in the fixed [13] and time-varying [12] regimes. Our bounds covers these results too. See Table 2 (Appendix G).

**(b) Optimal decentralized algorithms**

We construct **four new algorithms for stochastic (finite sum) decentralized variational inequalities: two for fixed networks, and two for time-varying networks. Two of these algorithms match our lower bounds, and are therefore optimal in terms of communication and local iteration complexities.** These are the first algorithms for stochastic (finite sum) decentralized variational inequalities over fixed and time-varying networks. See Table 1.

Moreover, our results offer linear communication complexity for deterministic decentralized strongly monotone variational inequalities, which is an improvement upon the sublinear results of [14, 13, 10, 12]. Additionally, our algorithms have better guarantees on local computations than the methods developed by [62]. See Table 2 (Appendix G).

Let us also single out a number of works on decentralized saddle point problems or VIs which are not suitable for comparison with our results: [67, 5] consider non-monotone(minty) problems, [45] does not prove convergence, [43] assumes data homogeneity, and [64] considers a discrete problem.

Table 1: Summary of upper and lower bounds for communication and local computation complexities for finding an $\varepsilon$-solution for strongly monotone **stochastic (finite-sum) decentralized** variational inequality (1) over fixed and time-varying networks. Convergence is measured by the distance to the solution.

| | | Reference | Communication complexity | Local complexity | Weaknesses |
|---|---|---|---|---|---|
| **Fixed** | **Upper** | Mukherjee and Chakraborty [48] [1,2] | $\mathcal{O}\left(\chi^{\frac{4}{3}}\frac{L^{\frac{4}{3}}}{\mu^{\frac{4}{3}}}\log\frac{1}{\varepsilon}\right)$ | $\mathcal{O}\left(n\chi^{\frac{4}{3}}\frac{L^{\frac{4}{3}}}{\mu^{\frac{4}{3}}}\log\frac{1}{\varepsilon}\right)$ | weak communication rates weak local computation rates |
| | | Beznosikov et al. [14] [1,2] | $\mathcal{O}\left(\sqrt{\chi}\frac{L}{\mu}\log^2\frac{1}{\varepsilon}\right)$ | $\mathcal{O}\left(n\frac{L}{\mu}\log\frac{L+\mu}{\mu}\log\frac{1}{\varepsilon}\right)$ | multiple gossip no linear convergence |
| | | Beznosikov et al. [13] [1,3] | $\mathcal{O}\left(\sqrt{\chi}\frac{L}{\mu}\log^2\frac{1}{\varepsilon}\right)$ | $\mathcal{O}\left(n\frac{L}{\mu}\log\frac{1}{\varepsilon}\right)$ | multiple gossip no linear convergence |
| | | Rogozin et al. [62] [1,2,4] | $\mathcal{O}\left(\sqrt{\chi}\frac{L}{\mu}\log\frac{1}{\varepsilon}\right)$ | $\mathcal{O}\left(n\sqrt{\chi}\frac{L}{\mu}\log\frac{1}{\varepsilon}\right)$ | weak local computation rates |
| | | Alg. 1 (this paper) | $\mathcal{O}\left(\max[\sqrt{n};\sqrt{\chi}]\frac{L}{\mu}\log\frac{1}{\varepsilon}\right)$ [6] | $\mathcal{O}\left(\max[\sqrt{n};\sqrt{\chi}]\frac{L}{\mu}\log\frac{1}{\varepsilon}\right)$ [6] | |
| | | Alg. 1 + Alg. 3 (this paper) | $\mathcal{O}\left(\sqrt{\chi}\frac{L}{\mu}\log\frac{1}{\varepsilon}\right)$ [6] | $\mathcal{O}\left(\sqrt{n}\frac{L}{\mu}\log\frac{1}{\varepsilon}\right)$ [6] | multiple gossip |
| | **Lower** | Beznosikov et al. [13] [3] | $\Omega\left(\sqrt{\chi}\frac{L}{\mu}\log\frac{1}{\varepsilon}\right)$ | $\Omega\left(\frac{L}{\mu}\log\frac{1}{\varepsilon}\right)$ | |
| | | Thm. 3.2 + Cor. 3.3 (this paper) | $\Omega\left(\sqrt{\chi}\frac{L}{\mu}\log\frac{1}{\varepsilon}\right)$ | $\Omega\left(\sqrt{n}\frac{L}{\mu}\log\frac{1}{\varepsilon}\right)$ | |
| **Time-varying** | **Upper** | Beznosikov et al. [10] [3] | $\mathcal{O}\left(\chi\frac{L}{\mu}\log\frac{1}{\varepsilon}+\chi\frac{LD}{\mu^2\sqrt{\varepsilon}}\right)$ [5] | $\mathcal{O}\left(n\chi\frac{L}{\mu}\log\frac{1}{\varepsilon}+n\chi\frac{LD}{\mu^2\sqrt{\varepsilon}}\right)$ | $D$-homogeneity no linear convergence |
| | | Beznosikov et al. [12] [1,2] | $\mathcal{O}\left(\chi\frac{L}{\mu}\log^2\frac{1}{\varepsilon}\right)$ | $\mathcal{O}\left(n\frac{L}{\mu}\log\frac{1}{\varepsilon}\right)$ | multiple gossip no linear convergence |
| | | Alg. 2 (this paper) | $\mathcal{O}\left(\max[\sqrt{n};\chi]\frac{L}{\mu}\log\frac{1}{\varepsilon}\right)$ [5,6] | $\mathcal{O}\left(\max[\sqrt{n};\chi]\frac{L}{\mu}\log\frac{1}{\varepsilon}\right)$ [6] | |
| | | Alg. 2 + Eq. 6 (this paper) | $\mathcal{O}\left(\chi\frac{L}{\mu}\log\frac{1}{\varepsilon}\right)$ [5,6] | $\mathcal{O}\left(\sqrt{n}\frac{L}{\mu}\log\frac{1}{\varepsilon}\right)$ [6] | multiple gossip |
| | **Lower** | Beznosikov et al. [12] [2] | $\Omega\left(\chi\frac{L}{\mu}\log\frac{1}{\varepsilon}\right)$ | $\Omega\left(\frac{L}{\mu}\log\frac{1}{\varepsilon}\right)$ | |
| | | Thm. 3.4 + Cor. 3.5 (This paper) | $\Omega\left(\chi\frac{L}{\mu}\log\frac{1}{\varepsilon}\right)$ [5] | $\Omega\left(\sqrt{n}\frac{L}{\mu}\log\frac{1}{\varepsilon}\right)$ | |

[1] for saddle point problems; [2] deterministic; [3] stochastic, but not finite sum; [4] convex-concave (monotone) case (we re-analyzed for strongly monotone case); [5] $B$-connected graphs [51] are also considered. For simplicity in comparison with other works, we put $B = 1$. To get estimates for $B \neq 1$, one need to change $\chi$ to $B\chi$; [6] can include additional factors such as $n\log\frac{1}{\varepsilon}$, $\chi\log\frac{1}{\varepsilon}$, for full complexities, see details in Section 4.
*Notation:* $\mu$ = constant of strong monotonicity of operator $F$, $L$ = Lipschitz constants of $L_{m,i}$, $\chi$ = characteristic number of the network (see Assumptions 2.3 and 2.4), $n$ = size of the local dataset.

**(c) Optimal non-distributed/centralized algorithms**

We believe it is notable that despite the generality of our setup and algorithms, **our results, when specialized to handle this simpler case, improve upon the current state-of-the-art results in the non-distributed/centralized setting.** In particular, unlike existing methods, our algorithms support *batching*: while the complexity of the best available algorithms grows with the batch size, our algorithms are not sensitive to this. This property is of crucial importance when working in the large batch mode, which is used in the practice [16, 73, 71]. See Table 3 (Appendix G).

**(d) Experiments**

Numerical experiments on bilinear problems and robust regression problems confirm the practical efficiency of our methods, both in the non-distributed stochastic setup and in the decentralized deterministic one.

## 2 Problem Setup and Assumptions

We write $\langle x, y \rangle := \sum_{i=1}^{n} x_i y_i$ to denote the standard inner product of vectors $x, y \in \mathbb{R}^n$, where $x_i$ corresponds to the $i$-th component of $x$ in the standard basis in $\mathbb{R}^n$. This induces the standard $\ell_2$-norm in $\mathbb{R}^n$ in the following way: $\|x\| := \sqrt{\langle x, x \rangle}$. To denote the Kronecker product of two matrices $\mathbf{A} \in \mathbb{R}^{m \times m}$ and $\mathbf{B} \in \mathbb{R}^{n \times n}$, we use $A \otimes B \in \mathbb{R}^{nm \times nm}$. The identity matrix of size $n \times n$ is denoted by $\mathbf{I}_n$. We write $[n] := \{1, 2, \ldots, n\}$. $\mathbb{N}$ is the set of positive integers.

### 2.1 Variational inequality

We study variational inequalities (VI) of the form

$$\text{Find} \quad z^* \in \mathbb{R}^d \quad \text{such that} \quad \langle F(z^*), z - z^* \rangle + g(z) - g(z^*) \geq 0, \quad \forall z \in \mathbb{R}^d, \tag{1}$$

where $F : \mathbb{R}^d \to \mathbb{R}^d$ is an operator, and $g : \mathbb{R}^d \to \mathbb{R} \cup \{+\infty\}$ is a proper lower semicontinuous convex function. We also assume that $g$ is proximal friendly, i.e. the computation of the operator $\text{prox}_{\rho g}(z) = \arg\min_{y \in \mathbb{R}^d} \{\rho g(y) + \frac{1}{2} \|y - z\|^2\}$ (with $\rho > 0$) is done for free or costs very low.

To showcase the expressive power of the formalism (1), we now give a few examples of variational inequalities arising in machine learning.

**Example 1 [Convex minimization].** Consider the convex regularized minimization problem:

$$\min_{z \in \mathbb{R}^d} f(z) + g(z), \tag{2}$$

where $f$ is typically a smooth data-fidelity term, and $g$ a possibly nonsmooth regularizer. If we define $F(z) := \nabla f(z)$, then it can be proved that $z^* \in \text{dom } g$ is a solution for (1) if and only if $z^* \in \text{dom } g$ is a solution for (2). So, the regularized optimization problem (2) can be cast as a VI (1).

While minimization problems are widely studied in a separate literature, the next class of problems is much more strongly tied to variational inequalities.

**Example 2 [Convex-concave saddles].** Consider the convex-concave saddle point problem

$$\min_{x \in \mathbb{R}^{d_x}} \max_{y \in \mathbb{R}^{d_y}} f(x, y) + g_1(x) - g_2(y), \tag{3}$$

where $g_1$ and $g_2$ can also be interpreted as regularizers. If we let $F(z) := F(x, y) = [\nabla_x f(x, y), -\nabla_y f(x, y)]$ and $g(z) = g(x, y) = g_1(x) + g_2(y)$, then it can be proved that $z^* \in \text{dom } g$ is a solution for (1) if and only if $z^* \in \text{dom } g$ is a solution for (3). So, convex-concave saddle point problems (3) can be cast as a VI (1).

Saddle point problems are strongly related to variational inequalities. In particular, lower bounds for the former are also valid for the latter. Moreover, upper bounds for variational inequalities are valid for saddle point problems. However, what is perhaps more important is that these lower and upper bounds match. This is in contrast to minimization, where the lower bounds are weaker.

### 2.2 Decentralized variational inequalities

We consider the decentralized case of problem (1), namely we assume that $F$ is distributed across $M$ workers,

$$F(z) := \sum_{m=1}^{M} F_m(z), \tag{4}$$

while each $F_m : \mathbb{R}^d \to \mathbb{R}^d$, $m \in [M]$, has the finite sum structure

$$F_m(z) := \frac{1}{n} \sum_{i=1}^{n} F_{m,i}(z). \tag{5}$$

The data describing $F_m$ being stored on worker $m$. For example, $F_{m,i}$ can correspond to the value of the operator on the $i$th data point of the $m$-th dataset.

## 2.3 Assumptions

**Assumption 2.1** (Lipschitzness). Each operator $F_m$ is $L$-Lipschitz continuous, i.e. for all $u, v \in \mathbb{R}^d$ we have $\|F_m(u) - F_m(v)\| \le L\|u - v\|$.

Further, the collection of operators is $\overline{L}$-average Lipschitz continuous, i.e., for all $u, v \in \mathbb{R}^d$ it holds $\frac{1}{n} \sum_{i=1}^n \|F_{m,i}(u) - F_{m,i}(v)\|^2 \le \overline{L}^2 \|u - v\|^2$.

In the context of (2) and (3), $L$-Lipschitzness of the operator means that the functions $f(z)$ and $f(x, y)$ are $L$-smooth.

**Assumption 2.2** (Strong monotonicity). Each operator $F_m$ is $\mu$-strongly monotone, i.e., for all $u, v \in \mathbb{R}^d$ we have $\langle F_m(u) - F_m(v); u - v \rangle \ge \mu\|u - v\|^2$.

In the context of (2) and (3), strong monotonicity of $F$ means strong convexity of $f(z)$ and strong convexity-strong concavity of $f(x, y)$.

## 2.4 Communication and gossip

Typically, decentralized communication is realized via a *gossip protocol* [68, 15, 50], which is merely matrix-vector multiplication with a gossip matrix $\mathbf{W}$, described below, which is different in the fixed and time-varying cases. Let $\mathcal{L} = \{\mathbf{z} = (z_1, \ldots, z_M)^\top \in (\mathbb{R}^d)^M : z_1 = \ldots = z_M\}$ be the *consensus space*.

**Assumption 2.3** (Fixed network [63]). For a fixed network, communication can be modeled via an undirected connected graph, $\mathcal{G} = (\mathcal{V}, \mathcal{E})$, where $\mathcal{V} = [n]$ are vertices (workers) and $\mathcal{E} = \{(i, j) \,|\, i, j \in \mathcal{V}\}$ are edges. Note that $(i, j) \in \mathcal{E}$ if and only if there exists a communication link between agents $i$ and $j$. The gossip matrix $\mathbf{W}$ satisfies the following three assumptions: 1) $\mathbf{W}$ is symmetric positive semi-definite; 2) $\ker \mathbf{W} \supset \mathcal{L}$; 3) $\mathbf{W}$ is supported on the vertices and edges of the network only: $w_{i,j} \neq 0$ if and only if $i = j$ or $(i, j) \in \mathcal{E}$.

To characterize the matrix $\mathbf{W}$, which captures the properties of the network, we denote $\lambda_{\max}(\mathbf{W}) = 1$ as the maximum eigenvalue of $\mathbf{W}$, $\lambda_{\min}^+(\mathbf{W})$ as the minimum positive eigenvalue of $\mathbf{W}$, and the characteristic number $\chi = \lambda_{\max}(\mathbf{W})/\lambda_{\min}^+(\mathbf{W}) = 1/\lambda_{\min}^+(\mathbf{W})$.

**Assumption 2.4** (Time-varying network [51]). For a time-varying network, at any moment $t$, communication network can be modeled as a directed $B$-connected graph, $\mathcal{G}(t) = (\mathcal{V}, \mathcal{E}(t))$, where $\mathcal{E}(t) = \{(i, j) \,|\, i, j \in \mathcal{V}\}$ are directed edges. $B$-connectedness means that for any time $t$, the graph $\mathcal{G}_B(t)$ with the set of edges $\bigcup_{\tau=t}^{t+B-1} \mathcal{E}(t)$ is connected. To describe the gossip protocol for time-varying case, we define the multi-consensus gossip matrix

$$\mathbf{W}_T(t) = \mathbf{I}_M - \prod_{\tau=t}^{t+T-1} \mathbf{W}(\tau). \tag{6}$$

One can also observe that multiplication with the matrix $\mathbf{W}_T$ requires to perform multiplication with $T$ gossip matrices $\mathbf{W}(t), \ldots, \mathbf{W}(t + T - 1)$, i.e., it requires $T$ decentralized communications. We further assume that the gossip matrices $\mathbf{W}(t)$ (for $\mathcal{G}(t)$) and $\mathbf{W}_B(t)$ satisfy: 1) $\mathbf{W}(t)$ is supported on the nodes and edges of the network: $w_{i,j}(t) \neq 0$ if and only if $i = j$ or $(i, j) \in \mathcal{E}(t)$; 2) $\ker \mathbf{W}(t) \supset \mathcal{L}$; 3) $\mathrm{range}\,\mathbf{W}(t) \subset \{\mathbf{z} \in (\mathbb{R}^d)^M : \sum_{m=1}^M z_m = 0\}$; 4) there exists a characteristic number $\chi \ge 1$ such that $\|\mathbf{W}_B(t)z - z\|^2 \le (1 - \chi^{-1})\|z\|^2$ for all $z \in \mathrm{range}\,\mathbf{W}_B(t)$.

# 3 Lower Bounds

Our lower bounds apply to a specific class of algorithms which are, loosely speaking, allowed to communicate with neighbors, and compute any local first-order information. We now give a formal definition.

**Definition 3.1** (Oracle). Each agent $m$ has its own local memory $\mathcal{M}_m$ with initialization $\mathcal{M}_m = \{0\}$. $\mathcal{M}_m$ is updated as follows. At each iteration, the algorithm either performs local computations or communicates.

• **Local computation:** At each local iteration, device $m$ can sample uniformly and independently batch $S_m$ of any size $b$ from $\{F_{m,i}\}$ and adds to its $\mathcal{M}_m$ a finite number of points $z$, satisfying

$$z \in \mathrm{span}\left\{z', \ \sum_{i_m \in S_m} F_{m,i_m}(z''), \mathrm{prox}_{\rho g}\left(\mathrm{span}\left\{z''', \ \sum_{i_m \in S_m} F_{m,i_m}(z'')\right\}\right)\right\} \tag{7}$$

for $z', z'', z''' \in \mathcal{M}_m$ and $\rho > 0$. Such a call needs $b$ local computations to collect the batch. Batch of size $n$ represents $F_m$;

• **Communication:** Upon communication rounds among the neighboring nodes, and at communication time $t$, $\mathcal{M}_m$ is updated according to

$$\mathcal{M}_m := \mathrm{span}\left\{\bigcup_{(i,m)\in\mathcal{E}(t)}\mathcal{M}_i\right\}. \tag{8}$$

• **Output:** The final global output is calculated as $\hat{z} \in \mathrm{span}\left\{\bigcup_{m=1}^{M}\mathcal{M}_m\right\}$.

The structure of the above definition is typical for distributed lower bounds [63] and for stochastic lower bounds [30]. In particular, Definition 3.1 includes all the approaches for working with stochastic problems, such as SGD or variance reduction techniques (SVRG, SARAH). Note that while our algorithm can invoke the deterministic oracle (full $F_m$) in local computations, in the work on lower bounds in the non-distributed case [29], there is no such a possibility. This narrows the class of algorithms for which results of [29] are valid. In particular, they can not do SVRG-type updates.

**Theorem 3.2** (Lower bound - fixed network). *For any $\overline{L} \geq \mu > 0$ and $\chi \geq 1$, $n \in \mathbb{N}$ and $K, N \in \mathbb{N}$, there exists a decentralized variational inequality (satisfying Assumptions 2.1 and 2.2) on $\mathbb{R}^d$ (where $d$ is sufficiently large) with $z^* \neq 0$ over a fixed network (satisfying Assumption 2.3) with a gossip matrix $\mathbf{W}$ and characteristic number $\chi$, such that for any output $\hat{z}$ of any procedure (Definition 3.1) with $K$ communication rounds and $N$ local computations, it holds that $\mathbb{E}[\|\hat{z} - z^*\|^2]$ is*

$$\Omega\left(\exp\left(-\frac{80}{1+\sqrt{\frac{2L^2}{\mu^2}+1}}\cdot\frac{K}{\sqrt{\chi}}\right)R_0^2\right) \quad and \quad \Omega\left(\exp\left(-\frac{16}{n+\sqrt{\frac{2n\overline{L}^2}{\mu^2}+n^2}}\cdot N\right)R_0^2\right),$$

*where $R_0^2 = \|z^0 - z^*\|^2$ and $L = \frac{\overline{L}}{\sqrt{n}}$.*

**Corollary 3.3.** *In the setting of Theorem 3.2, the number of communication rounds and local computations required to obtain an $\varepsilon$-solution (in expectation) is lower bounded by*

$$\Omega\left(\sqrt{\chi}\left(1+\frac{L}{\mu}\right)\cdot\log\left(\frac{R_0^2}{\varepsilon}\right)\right) \quad and \quad \Omega\left(\left(n+\sqrt{n}\cdot\frac{\overline{L}}{\mu}\right)\cdot\log\left(\frac{R_0^2}{\varepsilon}\right)\right), \quad respectively.$$

**Theorem 3.4** (Lower bound - time varying network). *For any $\overline{L} \geq \mu > 0$ and $\chi \geq 3$, $n \in \mathbb{N}$ and $K, N \in \mathbb{N}$, there exist a decentralized variational inequality (satisfying Assumptions 2.1 and 2.2) on $\mathbb{R}^d$ (where $d$ is sufficiently large) with $z^* \neq 0$ over a time-varying network (satisfying Assumption 2.4) with a sequence of gossip matrices $\mathbf{W}(t)$ and characteristic number $\chi$, such that for any output $\hat{z}$ of any procedure (Definition 3.1) with $K$ communication rounds and $N$ local computations, it holds that $\mathbb{E}[\|\hat{z} - z^*\|^2]$ is*

$$\Omega\left(\exp\left(-\frac{64}{\left(1+\sqrt{\frac{2L^2}{\mu^2}+1}\right)}\cdot\frac{K}{B\chi}\right)R_0^2\right) \quad and \quad \Omega\left(\exp\left(-\frac{16}{n+\sqrt{\frac{2n\overline{L}^2}{\mu^2}+n^2}}\cdot N\right)R_0^2\right),$$

*where $R_0^2 = \|z^0 - z^*\|^2$ and $L = \frac{\overline{L}}{\sqrt{n}}$.*

**Corollary 3.5.** *In the setting of Theorem 3.4, the number of communication rounds and local computations required to obtain an $\varepsilon$-solution (in expectation) is lower bounded by*

$$\Omega\left(B\chi\left(1+\frac{L}{\mu}\right)\cdot\log\left(\frac{R_0^2}{\varepsilon}\right)\right) \quad and \quad \Omega\left(\left(n+\sqrt{n}\cdot\frac{\overline{L}}{\mu}\right)\cdot\log\left(\frac{R_0^2}{\varepsilon}\right)\right), \quad respectively.$$

See proofs for Theorems 3.2 and 3.4 in Appendix C. The proof uses the idea (example of bad functions) of non-distributed deterministic lower bounds from [72]. This idea is further extended to distributed stochastic VIs.

Note that in the time-varying case, the lower bounds for communication differ by the constant $B$ from the estimates that were previously encountered in the literature [12]. This is due to the fact that we consider a more general setup with a $B$-connected graph (see Assumption 2.4), while the existing literature on lower bounds focuses on the simpler $B = 1$ case.

# 4  Optimal Algorithms

Since in the decentralized gossip protocol each local worker $m$ stores its own $z_m$ vector, we consider the problem

$$\text{Find} \quad \mathbf{z}^* \in (\mathbb{R}^d)^M \quad \text{such that} \quad \langle \mathbf{F}(\mathbf{z}^*), \mathbf{z} - \mathbf{z}^* \rangle + \mathbf{g}(\mathbf{z}) - \mathbf{g}(\mathbf{z}^*) \geq 0, \forall \mathbf{z} \in (\mathbb{R}^d)^M, \quad (9)$$

where we use new notation: $\mathbf{z} = (z_1, \ldots, z_M)^\top$ and $\mathbf{z}^* = (z_1^*, \ldots, z_M^*)^\top$. Additionally, here we introduce the lifted operator $\mathbf{F} : (\mathbb{R}^d)^M \to (\mathbb{R}^d)^M$ given as $\mathbf{F}(\mathbf{z}) = (F_1(z_1), \ldots, F_M(z_M))^\top$, and the lifted operator $\mathbf{g} : (\mathbb{R}^d)^M \to \mathbb{R} \cup \{+\infty\}$ defined by $\mathbf{g}(\mathbf{z}) = \frac{1}{M}\sum_{m=1}^M g(z_m)$.

One can note that (9) is a set of $M$ unrelated variational inequalities with their own variables. But the original problem (1) + (5) is a sum of variational inequalities with the same variables: $\sum_{m=1}^M \left[ \langle F(z^*), z - z^* \rangle + \frac{1}{M}g(z) - \frac{1}{M}g(z^*) \right]$. To eliminate this issue and move on to problem (1) + (5), it is easy to get the following modification of (9)

$$\text{Find} \quad \mathbf{z}^* \in \mathcal{L} \quad \text{such that} \quad \langle \mathbf{F}(\mathbf{z}^*), \mathbf{z} - \mathbf{z}^* \rangle + \mathbf{g}(\mathbf{z}) - \mathbf{g}(\mathbf{z}^*) \geq 0, \quad \forall \mathbf{z} \in \mathcal{L}, \quad (10)$$

where $\mathcal{L}$ is the consensus space. Problem (10) is equivalent to (1) + (5). Due to Assumptions 2.1 and 2.2, $\mathbf{F}$ is $L$-Lipschitz continuous, $\overline{L}$-average Lipschitz continuous and $\mu$-strongly monotone.

## 4.1  Fixed networks

We present Algorithm 1 for fixed networks. In Appendix B we give a discussion and intuition. In particular, we give the deterministic variant as well as the non-distributed version of Algorithm 1. The next result gives the iteration complexity of Algorithm 1.

**Theorem 4.1** (Upper bound - fixed network). *Consider the problem* (10) *(or* (1) + (5)*) under Assumptions* 2.1 *and* 2.2 *over a fixed graph* $\mathcal{G}$ *(Assumption* 2.3*) with a gossip matrix* $\mathbf{W}$. *Let* $\{\mathbf{z}^k\}$ *be the sequence generated by Algorithm* 1 *with tuning of* $\eta, \theta, \alpha, \beta, \gamma$ *as described in Appendix* D. *Then, given* $\varepsilon > 0$, *the number of iterations for* $\mathbb{E}[\|\mathbf{z}^k - \mathbf{z}^*\|^2] \leq \varepsilon$ *is*

$$\mathcal{O}\left( \left[ \frac{1}{p} + \chi + \frac{1}{\sqrt{pb}}\frac{\overline{L}}{\mu} + \sqrt{\chi}\frac{L}{\mu} \right] \log \frac{1}{\varepsilon} \right).$$

See the proof in Appendix D. Let us discuss the results of Theorem. First of all, we are interested in how to obtain the complexity of communications and local computations from iterative complexity. At each iteration we require (in average) $\mathcal{O}(b+pn)$ local computations, because we need to store batch $b$ twice and with probability $p$ we update the point $\mathbf{w}^{k+1}$ by $\mathbf{z}^k$, this requires calculating the full $\mathbf{F}$ in the next iteration. Then, as the optimal $p$, one can choose $p \sim \sqrt[b]{n}$. Then with such choice of $p$

---

**Algorithm 1**

1: **Parameters:** Stepsizes $\eta, \theta > 0$, momentums $\alpha, \beta, \gamma$, batchsize $b \in \{1, \ldots, n\}$, probability $p \in (0, 1)$
2: **Initialization:** Choose $\mathbf{z}^0 = \mathbf{w}^0 \in (\mathrm{dom}\,g)^M$, $\mathbf{y}^0 \in \mathcal{L}^\perp$. Put $\mathbf{z}^{-1} = \mathbf{z}^0$, $\mathbf{w}^{-1} = \mathbf{w}^0$, $\mathbf{y}^{-1} = \mathbf{y}^0$
3: **for** $k = 0, 1, 2 \ldots$ **do**
4: $\quad$ Sample $j_{m,1}^k, \ldots, j_{m,b}^k$ independently from $[n]$
5: $\quad S^k = \{j_{m,1}^k, \ldots, j_{m,b}^k\}$
6: $\quad$ Sample $j_{m,1}^{k+1/2}, \ldots, j_{m,b}^{k+1/2}$ independently from $[n]$
7: $\quad S^{k+1/2} = \{j_{m,1}^{k+1/2}, \ldots, j_{m,b}^{k+1/2}\}$
8: $\quad \delta^k = \frac{1}{b}\sum_{j \in S^k}\left( \mathbf{F}_j(\mathbf{z}^k) - \mathbf{F}_j(\mathbf{w}^{k-1}) \right.$
$\qquad\qquad\qquad \left. + \alpha[\mathbf{F}_j(\mathbf{z}^k) - \mathbf{F}_j(\mathbf{z}^{k-1})] \right) + \mathbf{F}(\mathbf{w}^{k-1})$
9: $\quad \Delta^k = \delta^k - (\mathbf{y}^k + \alpha(\mathbf{y}^k - \mathbf{y}^{k-1}))$
10: $\quad \mathbf{z}^{k+1} = \mathrm{prox}_{\eta\mathbf{g}}(\mathbf{z}^k + \gamma(\mathbf{w}^k - \mathbf{z}^k) - \eta\Delta^k)$
11: $\quad \Delta^{k+1/2} = \frac{1}{b}\sum_{j \in S^{k+1/2}}\left( \mathbf{F}_j(\mathbf{z}^{k+1}) - \mathbf{F}_j(\mathbf{w}^k) \right)$
$\qquad\qquad\qquad\qquad + \mathbf{F}(\mathbf{w}^k)$
12: $\quad \mathbf{y}^{k+1} = \mathbf{y}^k - \theta(\mathbf{W} \otimes \mathbf{I}_d)(\mathbf{z}^{k+1} - \beta(\Delta^{k+1/2} - \mathbf{y}^k))$
13: $\quad \mathbf{w}^{k+1} = \begin{cases} \mathbf{z}^k, & \text{with probability } p \\ \mathbf{w}^k, & \text{with probability } 1 - p \end{cases}$
14: **end for**

$\ast\mathbf{F}_j(\mathbf{z}) = (F_{1,j_{1,l}}(z_1), \ldots, F_{M,j_{M,l}}(z_M))^T, l \in \{1, \ldots, b\}$

---

we have the following local and communication complexities

$$\mathcal{O}\left( \left[ n + b\chi + \sqrt{n}\frac{\overline{L}}{\mu} + b\sqrt{\chi}\frac{L}{\mu} \right] \log \frac{1}{\varepsilon} \right) \quad \text{and} \quad \mathcal{O}\left( \left[ \frac{n}{b} + \chi + \frac{\sqrt{n}}{b}\frac{\overline{L}}{\mu} + \sqrt{\chi}\frac{L}{\mu} \right] \log \frac{1}{\varepsilon} \right),$$

respectively (since at each iteration Algorithm 1 performs $\mathcal{O}(1)$ communications).

Hence, with $b = 1$ we have the complexities the same as in Table 1. Depending on $\max\{\sqrt{n}; \sqrt{\chi}\}$, we have the optimality of either local communications or decentralized communications. One can

note that it is enough to take $b \geq \overline{L}\sqrt{n}/L$ and guarantee the optimal communication complexity (see Corollary 3.3), but we have non-optimality in local iterations.

To make the algorithm optimal both in terms of communications and local computations, we need to slightly modify it. One can make it using Chebyshev acceleration (see Algorithm 3 in Appendix A). Following [63], we can construct a polynomial $P$ such that 1) $P(\mathbf{W})$ is a gossip matrix, 2) multiplication by $P(\mathbf{W}) \otimes \mathbf{I}_d$ requires $\sqrt{\chi(\mathbf{W})}$ multiplications by $\mathbf{W}$ (i.e. $\sqrt{\chi(\mathbf{W})}$ communication rounds) 3) $\chi(P(\mathbf{W})) \leq 4$. Then we can modify Algorithm 1 by replacing $\mathbf{W}$ by $P(\mathbf{W})$ and get

**Theorem 4.2** (Upper bound - fixed network). *Consider the problem* (10) *(or* (1) + (5)*) under Assumptions 2.1 and 2.2 over a fixed connected graph $\mathcal{G}$ (Assumption 2.3) with a gossip matrix $\mathbf{W}$. Let $\{\mathbf{z}^k\}$ be the sequence generated by Algorithm 1 with Chebyshev polynomial $P(\mathbf{W})$ as a gossip matrix and with tuning of $\eta, \theta, \alpha, \beta, \gamma$ as described in Appendix D. Then, given $\varepsilon > 0$, the number of iterations for $\mathbb{E}[\|\mathbf{z}^k - \mathbf{z}^*\|^2] \leq \varepsilon$ is*

$$\mathcal{O}\left(\left[\frac{1}{p} + \frac{1}{\sqrt{pb}}\frac{\overline{L}}{\mu} + \frac{L}{\mu}\right]\log\frac{1}{\varepsilon}\right).$$

In this case, the communication complexity of one iteration is $\chi$, and the local complexity (in average) is still $\mathcal{O}(b + pn)$. Then with $p = b/n$ we get the following local and communication complexities

$$\mathcal{O}\left(\left[n + \sqrt{n}\frac{\overline{L}}{\mu} + b\frac{L}{\mu}\right]\log\frac{1}{\varepsilon}\right) \quad \text{and} \quad \mathcal{O}\left(\left[\sqrt{\chi}\frac{n}{b} + \sqrt{\chi}\frac{\sqrt{n}}{b}\frac{\overline{L}}{\mu} + \sqrt{\chi}\frac{L}{\mu}\right]\log\frac{1}{\varepsilon}\right),$$

respectively. To get optimal results from Table 1 we just need to take $b = \overline{L}\sqrt{n}/L$.

In contrast to algorithms of [14, 13] (the closest papers in theoretical convergence), our Algorithm 1 needs multi-consensus/Chebyshev acceleration for both optimal rates, but can work without these additional procedures. Algorithms [14, 13] requires $\mathcal{O}(\sqrt{\chi}\log\varepsilon^{-1})$ iterations for Chebyshev acceleration, which makes the algorithms less practical.

### 4.2 Time-varying networks

We present Algorithm 2 for time-varying networks. It needs to compute $\mathbf{W}_T$ using (6), it requires $T$ communications. In Appendix B we give a discussion and intuition of this algorithm. The next result gives the iteration complexity of Algorithm 2.

**Theorem 4.3** (Upper bound - time varying network). *Consider the problem* (10) *(or* (1) + (5)*) under Assumptions 2.1 and 2.2 over a sequence of time-varying graphs $\mathcal{G}(k)$ (Assumption 2.4) with gossip matrices $\mathbf{W}(k)$. Let $\{\mathbf{z}^k\}$ be the sequence generated by Algorithm 2 with $T \geq B$ and tuning of parameters as described in Appendix E. Let the choice of $T$ guarantees contraction property (Assumption 2.4 point 4) with $\chi(T)$. Then, given $\varepsilon > 0$, the number of iterations for $\mathbb{E}[\|\mathbf{z}^k - \mathbf{z}^*\|^2] \leq \varepsilon$ is*

$$\tilde{\mathcal{O}}\left(\chi^2(T) + \frac{1}{p} + \chi(T)\frac{L}{\mu} + \frac{1}{\sqrt{bp}}\frac{\overline{L}}{\mu}\right).$$

---

**Algorithm 2**

1: **Parameters:** Stepsizes $\eta_z, \eta_y, \eta_x, \theta > 0$, momentums $\alpha, \gamma, \omega, \tau$, parameters $\nu, \beta$, batchsize $b \in \{1, \ldots, n\}$, probability $p \in (0, 1)$
2: **Initialization:** Choose $\mathbf{z}^0 = \mathbf{w}^0 \in (\mathrm{dom}\,g)^M$, $\mathbf{y}^0 \in (\mathbb{R}^d)^M$, $\mathbf{x}^0 \in \mathcal{L}^\perp$. Put $\mathbf{z}^{-1} = \mathbf{z}^0, \mathbf{w}^{-1} = \mathbf{w}^0, \mathbf{y}_f = \mathbf{y}^{-1} = \mathbf{y}^0, \mathbf{x}_f = \mathbf{x}^{-1} = \mathbf{x}^0, m_0 = \mathbf{0}^{dM}$
3: **for** $k = 0, 1, 2, \ldots$ **do**
4:     Sample $j_{m,1}^k, \ldots, j_{m,b}^k$ independently from $[n]$
5:     $S^k = \{j_{m,1}^k, \ldots, j_{m,b}^k\}$
6:     Sample $j_{m,1}^{k+1/2}, \ldots, j_{m,b}^{k+1/2}$ independently from $[n]$
7:     $S^{k+1/2} = \{j_{m,1}^{k+1/2}, \ldots, j_{m,b}^{k+1/2}\}$
8:     $\delta^k = \frac{1}{b}\sum_{j\in S^k}\left(\mathbf{F}_j(\mathbf{z}^k) - \mathbf{F}_j(\mathbf{w}^{k-1})\right.$
          $\left. + \alpha[\mathbf{F}_j(\mathbf{z}^k) - \mathbf{F}_j(\mathbf{z}^{k-1})]\right) + \mathbf{F}(\mathbf{w}^{k-1})$
9:     $\Delta_z^k = \delta^k - \nu\mathbf{z}^k - \mathbf{y}^k - \alpha(\mathbf{y}^k - \mathbf{y}^{k-1})$
10:     $\mathbf{z}^{k+1} = \mathrm{prox}_{\eta_z\mathbf{g}}(\mathbf{z}^k + \omega(\mathbf{w}^k - \mathbf{z}^k) - \eta_z\Delta_z^k)$
11:     $\mathbf{y}_c^k = \tau\mathbf{y}^k + (1 - \tau)\mathbf{y}_f^k$
12:     $\mathbf{x}_c^k = \tau\mathbf{x}^k + (1 - \tau)\mathbf{x}_f^k$
13:     $\Delta_y^k = \nu^{-1}(\mathbf{y}_c^k + \mathbf{x}_c^k) + \mathbf{z}^{k+1} + \gamma(\mathbf{y}^k + \mathbf{x}^k + \nu\mathbf{z}^k)$
14:     $\delta^{k+1/2} = \frac{1}{b}\sum_{j\in S^{k+1/2}}\left(\mathbf{F}_j(\mathbf{z}^{k+1}) - \mathbf{F}_j(\mathbf{w}^k)\right)$
          $+ \mathbf{F}(\mathbf{w}^k)$
15:     $\Delta_x^k = \nu^{-1}(\mathbf{y}_c^k + \mathbf{x}_c^k) + \beta(\mathbf{x}^k + \delta^{k+1/2})$
16:     $\mathbf{y}^{k+1} = \mathbf{y}^k - \eta_y\Delta_y^k$
17:     $\mathbf{x}^{k+1} = \mathbf{x}^k - (\mathbf{W}_T(Tk) \otimes \mathbf{I}_d)(\eta_x\Delta_x^k + m^k)$
18:     $m^{k+1} = \eta_x\Delta_x^k + m^k$
          $- (\mathbf{W}_T(Tk) \otimes \mathbf{I}_d)(\eta_x\Delta_x^k + m^k)$
19:     $\mathbf{y}_f^{k+1} = \mathbf{y}_c^k + \tau(\mathbf{y}^{k+1} - \mathbf{y}^k)$
20:     $\mathbf{x}_f^{k+1} = \mathbf{x}_c^k - \theta(\mathbf{W}_T(Tk) \otimes \mathbf{I}_d)(\mathbf{y}_c^k + \mathbf{x}_c^k)$
21:     $\mathbf{w}^{k+1} = \begin{cases} \mathbf{z}^k, & \text{with probability } p \\ \mathbf{w}^k, & \text{with probability } 1 - p \end{cases}$
22: **end for**

---

Note that an important detail of the method is that $T \geq B$. This limitation is due to the fact that network is $B$-connected. In particular, for $B > 1$ it can happen that in some communications we use empty graphs. Therefore, the requirement $T \geq B$ is natural to guarantee the contraction property (Assumption 2.4, point 4). This means that if $B > 1$ we have to use multi-consensus (6).

But with $B = T = 1$, we can avoid multi-consensus, let us this case first. The same way as in fixed graph case we choose $p = {}^b\!/n$. Then we get the estimates on communications and local calls

$$\mathcal{O}\left(\left[\chi^2 + \frac{n}{b} + \chi\frac{L}{\mu} + \frac{\sqrt{n}}{b}\frac{\overline{L}}{\mu}\right]\log\frac{1}{\varepsilon}\right).$$

If we put $b = 1$, we have the same estimates as in Table 1.

Now we consider general case with any $B$. We use a multi-gossip step and take $T > 1$. In particular, let us choose $T = B \cdot \lceil \chi \ln 2 \rceil$. Than using (6) and point 4 of Assumption 2.4, we can guarantee that $\|\mathbf{W}_T(t)z - z\|^2 \leq \frac{1}{2}\|z\|^2$. Therefore, $\chi(T) = 2$, but we need $T$ communications per iteration. With $p = {}^b\!/n$ and $b = \overline{L}\sqrt{n}/L$, the iteration complexity from Theorem 4.3 can be rewritten as follows

$$\mathcal{O}\left(\left[1 + \frac{\sqrt{n}L}{\overline{L}} + \frac{L}{\mu}\right]\log\frac{1}{\varepsilon}\right).$$

Using that per iteration we make $\mathcal{O}(B \cdot \lceil \chi \ln 2 \rceil)$ communications and $\mathcal{O}\left(\overline{L}\sqrt{n}/L\right)$ local computations, we get

$$\mathcal{O}\left(\left[B\chi + B\chi\frac{\sqrt{n}L}{\overline{L}} + B\chi\frac{L}{\mu}\right]\log\frac{1}{\varepsilon}\right) \text{ commn-s and } \mathcal{O}\left(\left[n + \frac{\overline{L}\sqrt{n}}{L} + \sqrt{n}\frac{\overline{L}}{\mu}\right]\log\frac{1}{\varepsilon}\right) \text{ local calls.}$$

These results are reflected in Table 1.

## 5 Experiments

We now perform several experiments with the goal of corroborating our theoretical results. Note though that we are the first who consider the decentralized stochastic (finite-sum) setting for VIs, and hence there are no competing methods. Therefore, we compare the non-distributed finite sum setting and decentralized deterministic setting separately.

### 5.1 Variance reduction

In this section, we compare the main methods for solving strongly monotone non-distributed stochastic (finite-sum) variational inequalities with non-distributed version of our Algorithm 1.

**Problem.** We first consider bilinear problem:

$$\min_{x\in\triangle^d}\max_{y\in\triangle^d}\frac{1}{n}\sum_{i=1}^{n}x^\top\mathbf{A}_iy, \tag{11}$$

where $\triangle^d$ is the unit simplex in $\mathbb{R}^d$. We use the same experimental setup as in [1], in particular we consider policeman and burglar matrix from [54] and two test matrices from [55].

**Setting.** For comparison, we took methods from Table 3. In particular, we chose EG-Alc-Alg1 and EG-Alc-Alg2 from [1], EG-Car from [17]. The parameters of all methods are selected in two ways: 1) as described in the theory of the corresponding papers, and 2) tuned for the best convergence. We run all methods with different batch sizes. The comparison criterion is the number of epochs (one full gradient = epoch).

**Results.** The plots from Figure 1 show that in the case of a theoretical choice of parameters, our Algorithm 1 is ahead of other methods for any batch size (including $b = 1$). In the case of tuning parameters, the specialized method from

Figure 1: Comparison epoch complexities of Algorithm 1, EG-Alc-Alg1, EG-Alc-Alg2 and EG-Car on (11) with matrix from [54]. Dashed lines give convergence with theoretical parameters, solid lines – with tuned parameters.

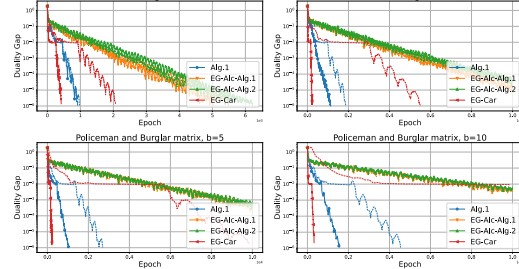

[17] is better than our algorithm. See more experiments with other matrices in Appendix F.1.

## 5.2 Decentralized methods

In this section, we compare the state-of-the-art methods for solving strongly monotone decentralized variational inequalities over fixed and time-varying networks with our Algorithms 1 and 2.

**Problem.** We now consider robust linear regression:

$$\min_{w} \max_{\|r\|\leq e} \frac{1}{N}\sum_{i=1}^{N}(w^{\top}(x_i + r_i) - y_i)^2 + \frac{\lambda}{2}\|w\|^2 - \frac{\beta}{2}\|r\|^2, \tag{12}$$

where $w$ are model weights, $\{(x_i, y_i)\}_{i=1}^{N}$ are pairs of the training data, $r_i$ are noise vectors, and $\lambda$ and $\beta$ are regularization parameters. The noises $r_i$ resist training the model, thereby inducing more robustness and stability.

**Setting.** For comparison, we took methods from Table 1 for decentralized problems over fixed and time-varying networks. In particular, we choose EGD-GT from [48], EGD-Con from [13, 12] and Sliding from [14]. Note that only EGD-Con has a theory for fixed and time-varying networks, despite this, we use all methods in both cases.

For a fair comparison, we consider the deterministic setup, i.e., each worker can compute full gradients. We take datasets from LiBSVM [19] and divided unevenly across $M = 25$ workers. For communication networks we chose the, the ring and the grid topologies. For time-varying networks, the topologies remain the same, but the locations of the vertices in them change randomly. All methods are tuned for the best convergence. The comparison criterion is the number of communication rounds.

### 5.2.1 Fixed networks

**Results.** The plots from Figure 2 show that our Algorithm 1 is ahead of other methods. Among other things, it is ahead of Sliding from [14], which has a fast theoretical communication complexity. However, this happens when the dataset is relatively homogeneous and uniformly divided across the devices. In our setting, this is not the case. See more experiments with other datasets in Appendix F.2.1.

### 5.2.2 Time-varying networks

**Results.** The plots from Figure 3 show that our Algorithm 2 is ahead of other methods in the case of the grid network. In the case of the ring topology, EGD-Con shows the best results. Indeed, such an algorithm in fact implements centralized communications via a decentralized protocol and this approach is not always the fastest, but reliable. See more experiments with other datasets in Appendix F.2.2.

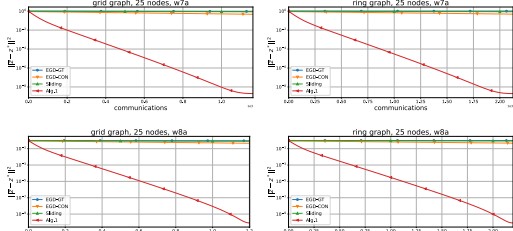

Figure 2: Comparison communication complexities of Algorithm 1, EGD-GT, EGD-Con and Sliding on (12) over fixed networks.

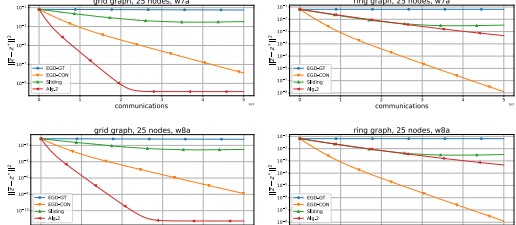

Figure 3: Comparison communication complexities of Algorithm 2, EGD-GT, EGD-Con and Sliding on (12) over time-varying networks.

## Acknowledgments

This work was supported by a grant for research centers in the field of artificial intelligence, provided by the Analytical Center for the Government of the Russian Federation in accordance with the subsidy agreement (agreement identifier 000000D730321P5Q0002) and the agreement with the Moscow Institute of Physics and Technology dated November 1, 2021 No. 70-2021-00138.

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
