# Contents

# Supplementary Materials

## A  Chebyshev acceleration

---

**Algorithm 3** Chebyshev gossip subroutine [63]

---

1: **Input:** $\mathbf{z}, \mathbf{W}$
2: $c_2 = \frac{\chi+1}{\chi-1}$, $a_0 = 1$, $a_1 = c_2$, $c_3 = \frac{2}{\lambda_{\max}(\mathbf{W}) + \lambda_{\min}^+(\mathbf{W})}$.
3: $\mathbf{z}^0 = \mathbf{z}$, $\mathbf{z}^1 = c_2(\mathbf{I} - c_3\mathbf{W})\mathbf{z}$.
4: **for** $k = 1, \ldots, K - 1$ **do**
5:     $a_{k+1} = 2c_2 a_k - a_{k-1}$.
6:     $\mathbf{z}^{k+1} = 2c_2(\mathbf{I} - c_3\mathbf{W})\mathbf{z}^k - \mathbf{z}^{k-1}$.
7: **end for**
8: **Output:** $\mathbf{z}^0 - \frac{\mathbf{z}^K}{a_K}$.

---

## B  Discussion and intuition of Algorithms 1 and 2

The design of Algorithm 1 and Algorithm 2 is based on a combination of two optimization ideas: variance reduction for variational inequalities (Algorithm 4) and optimal algorithms for decentralized distributed optimization on fixed (Algorithm 5) and time-varying (Algorithm 6) networks.

**Variance reduction for variational inequalities.** The variance reduction technique used in the design of Algorithms 1 and 2 is based on the FoRB with variance reduction [1, 2]. However, there are crucial differences with the FoRB algorithm that we now describe. Consider the variance reduced "gradient" estimator of FoRB in the case of batch size being equal to 1 ($b = 1$). It can be written as follows:

$$\delta^k = F_j(z^k) - F_j(w^{k-1}) + F(w^k), \tag{13}$$

where $j$ is sampled form $\{1, \ldots, n\}$ uniformly at random. This step in some sense combines the variance reduction technique and so-called "optimistic" step, which leads to a bad convergence rate $\mathcal{O}\left(n + \sqrt{bn}\frac{L}{\mu}\log\frac{1}{\varepsilon}\right)$ in the single node setting (see Table 3). Our Algorithm 4 (basic for Algorithm 1 and Algorithm 2) is different. They use the following "gradient" estimator (see line 6 of Algorithm 4):

$$\delta^k = \underbrace{F_j(z^k) - F_j(w^{k-1}) + F(w^{k-1})}_{\text{variance reduction}} + \alpha\underbrace{\left(F_j(z^k) - F_j(z^{k-1})\right)}_{\text{"optimistic" step}}. \tag{14}$$

One can observe that our gradient estimator in some sense separates the variance reduction step and the "optimistic" step. This leads to the better convergence rate $\mathcal{O}\left(n + \sqrt{n}\frac{L}{\mu}\log\frac{1}{\varepsilon}\right)$ in the single node setting (see Table 3) and allows to develop the optimal algorithms in the decentralized distributed setting.

**Decentralized distributed optimization.** To provide the optimal algorithms for decentralized stochastic variational inequalities, we use the optimal algorithms for solving decentralized distributed minimization problems for fixed networks [40] and for time-varying networks [39]. These algorithms use Nesterov acceleration [57] which we replace with our "gradient" estimator (14). Another important difference is that we extend our results to the composite case, i.e., $g(z) \neq 0$ in the main problem (1) with the help of the proximal operator $\text{prox}_g(\cdot)$. This is one of the important contributions of our paper. To the best of our knowledge, optimal algorithms in the decentralized distributed setting exist neither for minimization problems nor for the variational inequalities.

**Algorithm 4**

1: **Parameters:** Stepsizes $\eta > 0$, momentums $\alpha, \gamma$, batchsize $b \in \{1, \ldots, n\}$, probability $p \in (0, 1)$
2: **Initialization:** Choose $z^0 = w^0 \in \operatorname{dom} g$. Put $z^{-1} = z^0, w^{-1} = w^0$
3: **for** $k = 0, 1, 2 \ldots$ **do**
4:    Sample $j_1^k, \ldots, j_b^k$ independently from $\{1, \ldots, m\}$ uniformly at random
5:    $S^k = \{j_1^k, \ldots, j_b^k\}$
6:    $\Delta^k = \frac{1}{b} \sum_{j \in S^k} \left( F_j(x^k) - F_j(w^{k-1}) + \alpha(F_j(x^k) - F_j(x^{k-1})) \right) + F(w^{k-1})$
7:    $x^{k+1} = \operatorname{prox}_{\eta g}(x^k + \gamma(w^k - x^k) - \eta \Delta^k)$
8:    $w^{k+1} = \begin{cases} x^{k+1}, & \text{with probability } p \\ w^k, & \text{with probability } 1 - p \end{cases}$
9: **end for**

---

**Algorithm 5**

**Parameters:** Stepsizes $\eta, \theta > 0$, momentums $\alpha, \beta, \gamma$
**Initialization:** Choose $\mathbf{z}^0 = \mathbf{w}^0 \in (\operatorname{dom} g)^M$, $\mathbf{y}^0 \in \mathcal{L}^\perp$. Put $\mathbf{z}^{-1} = \mathbf{z}^0, \mathbf{w}^{-1} = \mathbf{w}^0, \mathbf{y}^{-1} = \mathbf{y}^0$
**for** $k = 0, 1, 2, \ldots$ **do**
   $\Delta^k = \mathbf{F}(x^k) + \alpha(\mathbf{F}(x^k) - \mathbf{F}(x^{k-1})) - (y^k + \alpha(y^k - y^{k-1}))$
   $x^{k+1} = \operatorname{prox}_{\eta \mathbf{g}}(x^k - \eta \Delta^k)$
   $y^{k+1} = y^k - \theta(\mathbf{W} \otimes \mathbf{I}_d)(x^{k+1} - \beta(\mathbf{F}(x^{k+1}) - y^k))$
**end for**

---

**Algorithm 6**

1: **Parameters:** Stepsizes $\eta_z, \eta_y, \eta_x, \theta > 0$, momentums $\alpha, \gamma, \omega, \tau$, parameters $\nu, \beta$
2: **Initialization:** Choose $\mathbf{z}^0 = \mathbf{w}^0 \in (\operatorname{dom} g)^M$, $\mathbf{y}^0 \in (\mathbb{R}^d)^M$, $\mathbf{x}^0 \in \mathcal{L}^\perp$. Put $\mathbf{z}^{-1} = \mathbf{z}^0, \mathbf{w}^{-1} = \mathbf{w}^0, \mathbf{y}_f = \mathbf{y}^{-1} = \mathbf{y}^0, \mathbf{x}_f = \mathbf{x}^{-1} = \mathbf{x}^0, m_0 = \mathbf{0}^{dM}$
3: **for** $k = 0, 1, 2, \ldots$ **do**
4:    $\Delta_z^k = \mathbf{F}(\mathbf{z}^k) + \alpha(\mathbf{F}(\mathbf{z}^k) - \mathbf{F}(\mathbf{z}^{k-1})) - \nu \mathbf{z}^k - \mathbf{y}^k - \alpha(\mathbf{y}^k - \mathbf{y}^{k-1})$
5:    $\mathbf{z}^{k+1} = \operatorname{prox}_{\eta_z \mathbf{g}}(\mathbf{z}^k - \eta_z \Delta_z^k)$
6:    $\mathbf{y}_c^k = \tau \mathbf{y}^k + (1 - \tau)\mathbf{y}_f^k$
7:    $\mathbf{x}_c^k = \tau \mathbf{x}^k + (1 - \tau)\mathbf{x}_f^k$
8:    $\Delta_y^k = \nu^{-1}(\mathbf{y}_c^k + \mathbf{x}_c^k) + \mathbf{z}^{k+1} + \gamma(\mathbf{y}^k + \mathbf{x}^k + \nu \mathbf{z}^k)$
9:    $\Delta_x^k = \nu^{-1}(\mathbf{y}_c^k + \mathbf{x}_c^k) + \beta(\mathbf{x}^k + \mathbf{F}(\mathbf{z}^k))$
10:    $\mathbf{y}^{k+1} = \mathbf{y}^k - \eta_y \Delta_y^k$
11:    $\mathbf{x}^{k+1} = \mathbf{x}^k - (\mathbf{W}_T(Tk) \otimes \mathbf{I}_d)(\eta_x \Delta_x^k + m^k)$
12:    $m^{k+1} = \eta_x \Delta_x^k + m^k - (\mathbf{W}_T(Tk) \otimes \mathbf{I}_d)(\eta_x \Delta_x^k + m^k)$
13:    $\mathbf{y}_f^{k+1} = \mathbf{y}_c^k + \tau(\mathbf{y}^{k+1} - \mathbf{y}^k)$
14:    $\mathbf{x}_f^{k+1} = \mathbf{x}_c^k - \theta(\mathbf{W}_T(Tk) \otimes \mathbf{I}_d)(\mathbf{y}_c^k + \mathbf{x}_c^k)$
15: **end for**

# C  Proof of Theorems 3.2 and 3.4

The idea of obtaining lower bounds is to find some "bad" example in the class of problems. In particular, we consider saddle point problems (as a special case of VIs) and look for the "bad" function among them. The paper [72] gives such an example for non-distributed deterministic problems. To get a distributed bounds, this problem needs to be divided between computing devices, which are connected in some kind of "bad" computing network [63, 39]. Then on each of the devices the function must be further divided in order to obtain a stochastic sum type problem [30, 29].

As stated above, to obtain lower bounds, we consider a particular case of the variational inequality (1)+(5), the saddle point problem:

$$\min_{x \in \mathbb{R}^d} \max_{y \in \mathbb{R}^d} f(x,y) := \sum_{m=1}^{M} f_m(x,y) := \sum_{m=1}^{M} \frac{1}{n} \sum_{i=1}^{n} f_{m,i}(x,y). \tag{15}$$

In this case $F(z) = F(x,y) = [\nabla_x f(x,y), -\nabla_y f(x,y)]$ and $g \equiv 0$. Next we rewrite Assumptions 2.1 and 2.2 for (15).

**Assumption C.1.** Suppose that
• each $f_m$ is $\overline{L}$-average smooth, on $\mathbb{R}^d \times \mathbb{R}^d$, i.e. for all $x_1, x_2 \in \mathbb{R}^d$, $y_1, y_2 \in \mathbb{R}^d$ it holds that

$$\frac{1}{n} \sum_{i=1}^{n} (\|\nabla_x f_{m,i}(x_1,y_1) - \nabla_x f_{m,i}(x_2,y_2)\|^2 + \|\nabla_y f_{m,i}(x_1,y_1) - \nabla_y f_{mi}(x_2,y_2)\|^2)$$
$$\leq \overline{L}^2 \left( \|x_1 - x_2\|^2 + \|y_1 - y_2\|^2 \right); \tag{16}$$

• each $f_m$ is $L$-smooth, on $\mathbb{R}^d \times \mathbb{R}^d$, i.e. for all $x_1, x_2 \in \mathbb{R}^d$, $y_1, y_2 \in \mathbb{R}^d$ it holds that

$$\|\nabla_x f_m(x_1,y_1) - \nabla_x f_m(x_2,y_2)\|^2 + \|\nabla_y f_m(x_1,y_1) - \nabla_y f_m(x_2,y_2)\|^2$$
$$\leq L^2 \left( \|x_1 - x_2\|^2 + \|y_1 - y_2\|^2 \right); \tag{17}$$

• $f$ is $\mu$ - strongly-convex-strongly-concave on $\mathbb{R}^d \times \mathbb{R}^d$, i.e. for all $x_1, x_2 \in \mathbb{R}^d$, $y_1, y_2 \in \mathbb{R}^d$ it holds that

$$\langle \nabla_x f(x_1,y_1) - \nabla_x f(x_2,y_2); x_1 - x_2 \rangle - \langle \nabla_y f(x_1,y_1) - \nabla_y f(x_2,y_2); y_1 - y_2 \rangle$$
$$\geq \mu \left( \|x_1 - x_2\|^2 + \|y_1 - y_2\|^2 \right). \tag{18}$$

Next we rewrite Definition 3.1 for (15). Since $g \equiv 0$, then $\text{prox}_{\rho g}(z) = z$.

**Definition C.2** (Oracle). Each agent $m$ has its own local memories $\mathcal{M}_m^x$ and $\mathcal{M}_m^y$ for the $x$- and $y$-variables, respectively–with initialization $\mathcal{M}_m^x = \mathcal{M}_m^y = \{0\}$. $\mathcal{M}_m^x$ and $\mathcal{M}_m^x$ are updated as follows.

• **Local computation:** At each local iteration device $m$ can sample uniformly and independently batch $S_m$ of any size $b$ from $\{f_{m,i}\}$ and adds to its $\mathcal{M}_m^x$ and $\mathcal{M}_m^y$ a finite number of points $x, y$, satisfying

$$x \in \text{span}\{x', \sum_{i_m \in S_m} \nabla_x f_{m,i_m}(x'', y'')\},$$
$$y \in \text{span}\{y', \sum_{i_m \in S_m} \nabla_y f_{m,i_m}(x'', y'')\}, \tag{19}$$

for given $x', x'' \in \mathcal{M}_m^x$ and $y', y'' \in \mathcal{M}_m^y$. Such call needs $b$ local computations to collect the batch. Batch of the size $n$ is equal to the full $f_m$;

• **Communication:** Based upon communication rounds among neighbouring nodes, at the communication with the number $t$, $\mathcal{M}_m^x$ and $\mathcal{M}_m^y$ are updated according to

$$\mathcal{M}_m^x := \text{span}\left\{ \bigcup_{(i,m) \in \mathcal{E}(t)} \mathcal{M}_i^x \right\}, \quad \mathcal{M}_m^y := \text{span}\left\{ \bigcup_{(i,m) \in \mathcal{E}(t)} \mathcal{M}_i^y \right\}. \tag{20}$$

• **Output:** The final global output is calculated as:

$$\hat{x} \in \text{span}\left\{ \bigcup_{m=1}^{M} \mathcal{M}_m^x \right\}, \quad \hat{y} \in \text{span}\left\{ \bigcup_{m=1}^{M} \mathcal{M}_m^y \right\}.$$

We construct the following bilinearly functions with $\overline{L}, \mu$. Let us consider a communication graph $G$ with vertexes $\{1, \ldots M\}$. Define $B = \{1\}$ and $\bar{B} = \{M\}$, with $|B| = |\bar{B}| = 1$. We then construct the following

bilinear functions on the graph:

$$f_m(x,y) = \begin{cases} f_1(x,y) = \frac{\overline{L}}{4\sqrt{n}}x^T A_1 y + \frac{\mu}{6}\|x\|^2 - \frac{\mu}{6}\|y\|^2 + \frac{\overline{L}^2}{2n\mu}e_1^T y, & m \in B; \\ f_2(x,y) = \frac{\overline{L}}{4\sqrt{n}}x^T A_2 y + \frac{\mu}{6}\|x\|^2 - \frac{\mu}{6}\|y\|^2, & m \in \bar{B}; \\ f_3(x,y) = \frac{1}{M-2}\left(\frac{\mu}{6}\|x\|^2 - \frac{\mu}{6}\|y\|^2\right), & \text{otherwise}; \end{cases} \quad (21)$$

where $e_1 = (1, 0\ldots, 0)$ and

$$A_1 = \begin{pmatrix} 1 & 0 & & & & & & \\ & 1 & -2 & & & & & \\ & & 1 & 0 & & & & \\ & & & 1 & -2 & & & \\ & & & \cdots & \cdots & & & \\ & & & & & 1 & -2 & \\ & & & & & & 1 & 0 \\ & & & & & & & 1 \end{pmatrix}, \quad A_2 = \begin{pmatrix} 1 & -2 & & & & & & \\ & 1 & 0 & & & & & \\ & & 1 & -2 & & & & \\ & & & 1 & 0 & & & \\ & & & \cdots & \cdots & & & \\ & & & & & 1 & 0 & \\ & & & & & & 1 & -2 \\ & & & & & & & 1 \end{pmatrix}.$$

Then we construct functions $f_{m,i}$. Let define $a_{1,q}$ $q$th row of the matrix $A_1$. Then we split function $f_1$ to finite sum $\frac{1}{n}\sum_{i=1}^n f_{1,i}$ in following way:

$$f_{1,i} = \sqrt{n}\cdot\frac{\overline{L}}{4}x^T\left[\sum_{j\equiv(i-1)\bmod n}\left(e_{2j+1}a_{1,2j+1}^T + e_{2j+2}a_{1,2j+2}^T\right)\right]y$$

$$+ \frac{\mu}{6}\|x\|^2 - \frac{\mu}{6}\|y\|^2 + \frac{\overline{L}^2}{2n\mu}e_1^T y \quad (22)$$

Let define $A_{1,i} = \sum_{j\equiv(i-1)\bmod n}\left(e_{2j+1}a_{1,2j+1}^T + e_{2j+2}a_{1,2j+2}^T\right)$. The same way we can construct $f_{2,i}$ with $A_{2,i}$ and $f_{3,i}$ with $A_{3,i} = 0$. Consider the global objective function:

$$f(x,y) = \sum_{m=1}^M f_m(x,y) = |B|\cdot f_1(x,y) + |\bar{B}|\cdot f_2(x,y) + (M - |B| - |\bar{B}|)\cdot f_3(x,y)$$

$$= \frac{\overline{L}}{2\sqrt{n}}x^T A y + \frac{\mu}{2}\|x\|^2 - \frac{\mu}{2}\|y\|^2 + \frac{\overline{L}^2}{2n\mu}e_1^T y, \quad (23)$$

with $A = \frac{1}{2}(A_1 + A_2)$.

**Lemma C.3.** *Problem (21) with $f_m$ from (22) satisfies Assumption C.1 with $\overline{L}, L = \frac{\overline{L}}{\sqrt{n}}, \mu$.*

*Proof.* It is easy to verify that $f$ is $\mu$ - strongly-convex-strongly-concave. Also with $\|A_1\|, \|A_2\| \leq 3$, we get that $f_m(x,y)$ is $L = \frac{\overline{L}}{\sqrt{n}}$ - smooth. Then we need to check that $f_m$ is $\overline{L}$ - average smooth:

$$\frac{1}{n}\sum_{i=1}^n\left[\|\nabla_x f_{1,i}(x_1,y_1) - \nabla_x f_{1,i}(x_2,y_2)\|^2 + \|\nabla_y f_{1,i}(x_1,y_1) - \nabla_y f_{1,i}(x_2,y_2)\|^2\right]$$

$$= \frac{1}{n}\sum_{i=1}^n\left[\left\|\sqrt{n}\cdot\frac{\overline{L}}{4}A_{1,i}(y_1 - y_2) + \frac{\mu}{3}(x_1 - x_2)\right\|^2 + \left\|\sqrt{n}\cdot\frac{\overline{L}}{4}A_{1,i}^T(x_2 - x_1) + \frac{\mu}{3}(y_1 - y_2)\right\|^2\right]$$

$$\leq \frac{1}{n}\sum_{i=1}^n\left[\frac{n\overline{L}^2}{8}\|A_{1,i}(y_1 - y_2)\|^2 + \frac{2\mu^2}{9}\|x_1 - x_2\|^2 + \frac{n\overline{L}^2}{8}\left\|A_{1,i}^T(x_1 - x_2)\right\|^2 + \frac{2\mu^2}{9}\|y_1 - y_2\|^2\right]$$

$$= \frac{\overline{L}^2}{8}\sum_{i=1}^n\left[(y_1 - y_2)^T A_{1,i}^T A_{1,i}(y_1 - y_2) + (x_1 - x_2)^T A_{1,i}A_{1,i}^T(x_1 - x_2)\right]$$

$$+ \frac{2\mu^2}{9}\|x_1 - x_2\|^2 + \frac{2\mu^2}{9}\|y_1 - y_2\|^2$$

$$= \frac{\overline{L}^2}{8} \sum_{i=1}^n (y_1 - y_2)^T \left[ \sum_{j \equiv (i-1) \bmod n} \left( a_{1,2j+1} e_{2j+1}^T + a_{1,2j+2} e_{2j+2}^T \right) \right]$$

$$\left[ \sum_{j \equiv (i-1) \bmod n} \left( e_{2j+1} a_{1,2j+1}^T + e_{2j+2} a_{1,2j+2}^T \right) \right] (y_1 - y_2)$$

$$+ \frac{\overline{L}^2}{8} \sum_{i=1}^n (x_1 - x_2)^T \left[ \sum_{j \equiv (i-1) \bmod n} \left( e_{2j+1} a_{1,2j+1}^T + e_{2j+2} a_{1,2j+2}^T \right) \right]$$

$$\left[ \sum_{j \equiv (i-1) \bmod n} \left( a_{1,2j+1} e_{2j+1}^T + a_{1,2j+2} e_{2j+2}^T \right) \right] (x_1 - x_2)$$

$$+ \frac{2\mu^2}{9} \|x_1 - x_2\|^2 + \frac{2\mu^2}{9} \|y_1 - y_2\|^2$$

$$= \frac{\overline{L}^2}{8} \sum_{i=1}^n (y_1 - y_2)^T \left[ \sum_{j \equiv (i-1) \bmod n} \left( a_{1,2j+1} a_{1,2j+1}^T + a_{1,2j+2} a_{1,2j+2}^T \right) \right] (y_1 - y_2)$$

$$+ \frac{\overline{L}^2}{8} \sum_{i=1}^n (x_1 - x_2)^T \left[ \sum_{j \equiv (i-1) \bmod n} \left( e_{2j+1} e_{2j+1}^T + 5 e_{2j+2} e_{2j+2}^T \right) \right]^T (x_1 - x_2)$$

$$+ \frac{2\mu^2}{9} \|x_1 - x_2\|^2 + \frac{2\mu^2}{9} \|y_1 - y_2\|^2$$

$$\leq \frac{\overline{L}^2}{8} \sum_{j=1}^d (y_1 - y_2)^T \left( a_{1,j} a_{1,j}^T \right) (y_1 - y_2) + \frac{5 \overline{L}^2}{8} \sum_{j=1}^d (x_1 - x_2)^T \left( e_j e_j^T \right) (x_1 - x_2)$$

$$+ \frac{2\mu^2}{9} \|x_1 - x_2\|^2 + \frac{2\mu^2}{9} \|y_1 - y_2\|^2$$

$$\leq \frac{\overline{L}^2}{8} \sum_{j=1}^d (y_1 - y_2)^T \left( a_{1,j} a_{1,j}^T \right) (y_1 - y_2) + \frac{5 \overline{L}^2}{8} \|x_1 - x_2\|^2$$

$$+ \frac{2\mu^2}{9} \|x_1 - x_2\|^2 + \frac{2\mu^2}{9} \|y_1 - y_2\|^2$$

$$\leq \frac{3 \overline{L}^2}{4} \|y_1 - y_2\|^2 + \frac{5 \overline{L}^2}{8} \|x_1 - x_2\|^2 + \frac{2\mu^2}{9} \|x_1 - x_2\|^2 + \frac{2\mu^2}{9} \|y_1 - y_2\|^2.$$

The last inequality follows from $\lambda_{\max} \left( \sum_{j=1}^d \left( a_{1,j} a_{1,j}^T \right) \right) \leq 6$. Finally, with $\mu \leq \overline{L}$ we get

$$\frac{1}{n} \sum_{i=1}^n \left[ \|\nabla_x f_{1,i}(x_1, y_1) - \nabla_x f_{1,i}(x_2, y_2)\|^2 + \|\nabla_y f_{1,i}(x_1, y_1) - \nabla_y f_{1,i}(x_2, y_2)\|^2 \right]$$

$$\leq \overline{L}^2 \left( \|x_1 - x_2\|^2 + \|y_1 - y_2\|^2 \right).$$

$\square$

The next two lemmas give an idea of how quickly we approximate the solution of (23) depending on the number of communications and local iterations. For simplicity, we divide a situation in two parts: one part is devoted to communications (taking into account the fact that we are not limited in the number of local iterations), the second - on the contrary (we concentrate on local computations and assume that communications cost nothing).

**Lemma C.4.** *Let Problem (21) be solved by any method that satisfies Definition C.2. Then after $K$ communication rounds, only the first $\lfloor \frac{K}{l} \rfloor$ coordinates of the global output can be non-zero while the rest of the $d - \lfloor \frac{K}{l} \rfloor$ coordinates are strictly equal to zero. Here $l$ is "distance" between $B$ and $\bar{B}$ (how quickly can we transfer information from $B$ to $\bar{B}$).*

*Proof.* We begin introducing some notation for our proof. Let

$$E_0 := \{0\}, \quad E_K := \text{span}\{e_1, \ldots, e_K\}.$$

Note that, the initialization gives $\mathcal{M}_m^x = E_0, \mathcal{M}_m^y = E_0$.

Suppose that, for some $m$, $\mathcal{M}_m^x = E_K$ and $\mathcal{M}_m^y = E_K$, at some given time. Let us analyze how $\mathcal{M}_m^x, \mathcal{M}_m^y$ can change by performing only local computations.

Firstly, we consider the case when $K$ odd. After one local update, we have the following:

• For machines $m$ which own $f_1$, it holds

$$x \in \text{span}\{e_1 \,,\, x' \,,\, A_1 y'\} = E_K,$$
$$y \in \text{span}\{e_1 \,,\, y' \,,\, A_1^T x'\} = E_K, \tag{24}$$

for given $x' \in \mathcal{M}_m^x$ and $y' \in \mathcal{M}_m^y$. In details, each local iteration uses matrices $A_{1,i}$ (in the stochastic) or $A_1$ (in the deterministic). But here we talks only about communications and do not pay attention to the number of local iterations. Therefore, without loss of generality, we can immediately assume that all local calculations change our output according to (24). Since $A_1$ has a block diagonal structure, after local computations, we have $\mathcal{M}_m^x = E_K$ and $\mathcal{M}_m^y = E_K$. The situation does not change, no matter how many local computations one does.

• For machines $m$ which own $f_2$, it holds

$$x \in \text{span}\{x' \,,\, A_2 y'\} = E_{K+1},$$
$$y \in \text{span}\{y' \,,\, A_2^T x'\} = E_{K+1},$$

for given $x' \in \mathcal{M}_m^x$ and $y' \in \mathcal{M}_m^y$. It means that, after local computations, one has $\mathcal{M}_m^x = E_{K+1}$ and $\mathcal{M}_m^y = E_{K+1}$. Therefore, machines with function $f_2$ can progress by one new non-zero coordinate.

This means that we constantly have to transfer progress from the group of machines with $f_1$ to the group of machines with $f_2$ and back. Initially, all devices have zero coordinates. Further, machines with $f_1$ can receive the first nonzero coordinate (but only the first, the second is not), and the rest of the devices are left with all zeros. Next, we pass the first non-zero coordinate to machines with $f_2$. To do this, $l$ communication rounds are needed. By doing so, they can make the second coordinate non-zero, and then transfer this progress to the machines with $f_1$. Then the process continues in the same way. This completes the proof. □

In the next lemma, we will give an understanding of how local progress towards a solution occurs. For this, we will assume that communications cost nothing.

**Lemma C.5.** *Let Problem (21) be solved by any method that satisfies Definition C.2. Then after $N$ local calls (for each node), in expectation only the first $\left\lfloor \frac{N}{n} \right\rfloor$ coordinates of the global output can be non-zero while the rest of the $d - \left\lfloor \frac{N}{n} \right\rfloor$ coordinates are strictly equal to zero.*

*Proof.* As is clear from the previous lemma, communications make sense if an update ($E_K \to E_{K+1}$) is reached on one of the nodes. Depending on $K$, this happens on the nodes with $f_1$ or $f_2$ (but not simultaneously). The question is how many local calls should be made to get this update. One can understand it by looking at the structure of matrices $A_{1,i}$ and $A_{2,i}$ from (22). Only one of $n$ matrices is suitable for us. For example, in case of $K = 2k$, we need $f_{1,j}$ with $j \equiv k \bmod n$.

Suppose that $s_1, s_2 \ldots$ times call stochastic oracle with batchsize $1, 2 \ldots$. Then $\sum_{j=1}^n j s_j + N$. Due to the fact that the choice of batches is random and uniform, the random variable responsible for the total number of updates during the operation of the algorithm has the sum of binomial distribution with pairs of parameters $\left(s_j; \frac{j}{n}\right)$. It remains only to take the mathematical expectation and the lemma is proved. □

The next lemma is devoted to provide an approximate solution of problem (23), and shows that this approximation is close to a real solution. The proof of the lemma follows closely that of Lemma 3.3 from [72], and is reported for the sake of completeness.

**Lemma C.6** (Lemma 3.3 from [72]). *Let $\alpha = \frac{2n\mu^2}{\overline{L}^2}$ and $q = \frac{1}{2}\left(2 + \alpha - \sqrt{\alpha^2 + 4\alpha}\right) \in (0; 1)$–the smallest root of $q^2 - (2 + \alpha)q + 1 = 0$; and let define*

$$\bar{y}_i^* = \frac{q^i}{1 - q}, \quad i \in [d].$$

*The following bound holds when $\bar{y}^* := [y_1^*, \ldots y_d^*]^\top$ is used to approximate the solution $y^*$:*

$$\|\bar{y}^* - y^*\| \le \frac{q^{d+1}}{\alpha(1 - q)}.$$

*Proof.* Let us write the dual function for (23):

$$h(y) = -\frac{1}{2} y^T \left(\frac{\overline{L}^2}{2n\mu} A^T A + \mu I\right) y + \frac{\overline{L}^2}{2n\mu} e_1^T y,$$

where it is not difficult to check that

$$
AA^T = \begin{pmatrix}
1 & -1 & & & & & & & \\
-1 & 2 & -1 & & & & & & \\
& -1 & 2 & -1 & & & & & \\
& & -1 & 2 & -1 & & & & \\
& & & -1 & 2 & -1 & & & \\
& & & & & \ddots & & & \\
& & & & & & -1 & 2 & -1 \\
& & & & & & & -1 & 2
\end{pmatrix}.
$$

The optimality of dual problem $\nabla h(y^*) = 0$ gives

$$
\left( \frac{\overline{L}^2}{2n\mu} A^T A + \mu I \right) y^* = \frac{\overline{L}^2}{2n\mu} e_1,
$$

or

$$
\left( A^T A + \alpha I \right) y^* = e_1.
$$

Equivalently, we can write

$$
\begin{cases}
(1+\alpha)y_1^* - y_2^* = 1, \\
-y_1^* + (2+\alpha)y_2^* - y_3^* = 0, \\
\cdots \\
-y_{d-2}^* + (2+\alpha)y_{d-1}^* - y_d^* = 0, \\
-y_{d-1}^* + (2+\alpha)y_d^* = 0.
\end{cases}
$$

On the other hand, the approximation $\bar{y}^*$ satisfies the following set of equations:

$$
\begin{cases}
(1+\alpha)\bar{y}_1^* - \bar{y}_2^* = 1, \\
-\bar{y}_1^* + (2+\alpha)\bar{y}_2^* - \bar{y}_3^* = 0, \\
\cdots \\
-\bar{y}_{d-2}^* + (2+\alpha)\bar{y}_{d-1}^* - \bar{y}_d^* = 0, \\
-\bar{y}_{d-1}^* + (2+\alpha)\bar{y}_d^* = \frac{q^{d+1}}{1-q},
\end{cases}
$$

or equivalently

$$
\left( A^T A + \alpha I \right) \bar{y}^* = e_1 + \frac{q^{d+1}}{1-q} e_d.
$$

Therefore, the difference between $\bar{y}^*$ and $y^*$ reads

$$
\bar{y}^* - y^* = \left( A^T A + \alpha I \right)^{-1} \frac{q^{d+1}}{1-q} e_d.
$$

The statement of the lemma follow from the above equality and $\alpha^{-1} I \succeq \left( A^T A + \alpha I \right)^{-1} \succ 0$. $\qquad \square$

The next lemma provides a lower bound for the solution of (23) in the distributed case (21). The proof follows closely that of Lemma 3.4 from [72] and is reported for the sake of completeness.

**Lemma C.7.** *Consider a distributed saddle-point problem with objective function given by* (23). *For any $K, N$, choose any problem size $d \geq \max\left\{ 2\log_q\left(\frac{\alpha}{4\sqrt{2}}\right), 2K, 2N \right\}$, where $\alpha = \frac{2n\mu^2}{\overline{L}^2}$ and $q = $*

$\frac{1}{2}\left(2 + \alpha - \sqrt{\alpha^2 + 4\alpha}\right) \in (0; 1)$. *Then, any output $\hat{x}, \hat{y}$ produced by any method satisfying Definition* C.2 *after $K$ communications and $N$ local calls, is such that*

$$\mathbb{E}\left[\|\hat{x} - x^*\|^2 + \|\hat{y} - y^*\|^2\right] \geq \left(q^{\frac{2K}{l}} + q^{\frac{2N}{n}}\right)\frac{\|y_0 - y^*\|^2}{16}.$$

*Proof.* Let us assume that in output we have $k$ non-zero coordinates. By definition of $\bar{y}^*$, with $q < 1$ and $k \leq \frac{d}{2}$, we have

$$
\begin{aligned}
\|\hat{y} - \bar{y}^*\|^2 &\geq \sqrt{\sum_{j=k+1}^{d}(\bar{y}_j^*)^2} = \frac{q^k}{1-q}\sqrt{q^2 + q^4 + \ldots + q^{2(d-k)}} \\
&\geq \frac{q^k}{\sqrt{2}(1-q)}\sqrt{q^2 + q^4 + \ldots + q^{2d}} = \frac{q^k}{\sqrt{2}}\|\bar{y}^*\|^2 = \frac{q^k}{\sqrt{2}}\|y_0 - \bar{y}^*\|^2.
\end{aligned}
$$

Using Lemma C.6 for $d \geq 2\log_q\left(\frac{\alpha}{4\sqrt{2}}\right)$ we can guarantee that $\bar{y}^* \approx y^*$ (for more detailed proof see [72]) and

$$\|\hat{x} - x^*\|^2 + \|\hat{y} - y^*\|^2 \geq \|\hat{y} - y^*\|^2 \geq \frac{q^{2k}}{16}\|y_0 - y^*\|^2.$$

It remains only to note that $k$ depends on the number of nonzero coordinates from communications $k_c$ and local computations $k_l$. For this we use Lemmas C.4 and C.5:

$$\|\hat{x} - x^*\|^2 + \|\hat{y} - y^*\|^2 \geq \frac{q^{2k}}{16}\|y_0 - y^*\|^2 \geq \frac{q^{2\min(k_c, k_l)}}{16}\|y_0 - y^*\|^2 \geq \frac{q^{k_c + k_l}}{16}\|y_0 - y^*\|^2.$$

By Lemma C.4 we have $k_c \leq \lfloor\frac{K}{l}\rfloor$, where $l$ is "distance" between $B$ and $\bar{B}$. By Lemma C.5 we get that $k_l$ has binomial distribution with parameters $N$ and $\frac{1}{n}$.

$$
\begin{aligned}
\mathbb{E}\left[\|\hat{x} - x^*\|^2 + \|\hat{y} - y^*\|^2\right] &\geq \frac{q^{2\lfloor\frac{K}{l}\rfloor}}{32}\|y_0 - y^*\|^2 + \mathbb{E}\left[\frac{q^{2k_l}}{32}\|y_0 - y^*\|^2\right] \\
&\geq \frac{q^{\frac{2K}{l}}}{32}\|y_0 - y^*\|^2 + \frac{\mathbb{E}\left[q^{2k_l}\right]}{32}\|y_0 - y^*\|^2 \\
&\geq \left(q^{\frac{2K}{l}} + q^{2\mathbb{E}[k_l]}\right)\cdot\frac{1}{32}\|y_0 - y^*\|^2 \\
&\geq \left(q^{\frac{2K}{l}} + q^{\frac{2N}{n}}\right)\cdot\frac{1}{32}\|y_0 - y^*\|^2.
\end{aligned}
$$

Here we use Jensen's inequality. $\qquad\square$

It remains to get an estimate on $l$ ("distance" between $B$ and $\bar{B}$). For the fixed network it is real distance, for time-varying – rate how fast information transmits in the network from $B$ to $\bar{B}$.

## C.1 Fixed network

**Theorem C.8** (Theorem 3.2). *Let $\bar{L} > \mu > 0$, $n \in \mathbb{N}$ (with $\bar{L}/\mu \geq \sqrt{n}$), $\chi \geq 1$ and $K, N \in \mathbb{N}$. There exists a distributed saddle-point problem over fixed network (Assumption 2.3). For which the following statements are true:*

- *a gossip matrix $\mathbf{W}$ has $\chi(\mathbf{W}) = \chi$,*

- $f = \sum_{m=1}^{M}\frac{1}{n}\sum_{i=1}^{n}f_{m.i} : \mathbb{R}^d \times \mathbb{R}^d \to \mathbb{R}$ *is $\mu$ – strongly-convex-strongly-concave,*

- $f_m$ *are $\bar{L}$-average smooth and $L = \frac{\bar{L}}{\sqrt{n}}$ - smooth,*

- *size $d \geq \max\left\{2\log_q\left(\frac{\alpha}{4\sqrt{2}}\right), 2K, 2N\right\}$, where $\alpha = \frac{2n\mu^2}{L^2}$ and $q = \frac{1}{2}\left(2 + \alpha - \sqrt{\alpha^2 + 4\alpha}\right) \in (0; 1)$,*

- *the solution of the problem is non-zero: $x^* \neq 0$, $y^* \neq 0$.*

*Then for any output $\hat{z}$ of any procedure (Definition* C.2*) with $K$ communication rounds and $N$ local computations, one can obtain the following estimate:*

$$\|\hat{z} - z^*\|^2 = \Omega\left(\exp\left(-\frac{80}{1 + \sqrt{\frac{2L^2}{\mu^2} + 1}}\cdot\frac{K}{\sqrt{\chi}}\right)\|y_0 - y^*\|^2\right);$$

$$\|\hat{z} - z^*\|^2 = \Omega\left(\exp\left(-\frac{16}{n + \sqrt{\frac{2n\overline{L}^2}{\mu^2} + n^2}} \cdot N\right)\|y_0 - y^*\|^2\right).$$

*Proof.* Applying Lemma C.7, we have

$$\left(\frac{1}{q}\right)^{\frac{2K}{l}} \geq \frac{\|y_0 - y^*\|^2}{32\mathbb{E}\left[\|\hat{x} - x^*\|^2 + \|\hat{y} - y^*\|^2\right]} \quad \text{and} \quad \left(\frac{1}{q}\right)^{\frac{2N}{n}} \geq \frac{\|y_0 - y^*\|^2}{32\mathbb{E}\left[\|\hat{x} - x^*\|^2 + \|\hat{y} - y^*\|^2\right]}.$$

Taking the logarithm on both sides, we get

$$\frac{2K}{l} \geq \ln\left(\frac{\|y_0 - y^*\|^2}{32\mathbb{E}\left[\|\hat{x} - x^*\|^2 + \|\hat{y} - y^*\|^2\right]}\right)\frac{1}{\ln(q^{-1})}.$$

Next, we work with

$$\frac{1}{\ln(q^{-1})} = \frac{1}{\ln(1 + (1-q)/q))} \geq \frac{q}{1-q} = \frac{1 + \frac{n\mu^2}{\overline{L}^2} - \sqrt{\frac{2n\mu^2}{\overline{L}^2} + \left(\frac{n\mu^2}{\overline{L}^2}\right)^2}}{\sqrt{\frac{2n\mu^2}{\overline{L}^2} + \left(\frac{n\mu^2}{\overline{L}^2}\right)^2} - \frac{n\mu^2}{\overline{L}^2}}$$

$$\geq \frac{\sqrt{\frac{2n\mu^2}{\overline{L}^2} + \left(\frac{n\mu^2}{\overline{L}^2}\right)^2} + \frac{n\mu^2}{\overline{L}^2}}{\frac{8n\mu^2}{\overline{L}^2}} = \frac{1}{8}\left(1 + \sqrt{\frac{2\overline{L}^2}{n\mu^2} + 1}\right).$$

One can then obtain

$$\frac{2K}{l} \geq \ln\left(\frac{\|y_0 - y^*\|^2}{32\mathbb{E}\left[\|\hat{x} - x^*\|^2 + \|\hat{y} - y^*\|^2\right]}\right) \cdot \frac{1}{8}\left(1 + \sqrt{\frac{2\overline{L}^2}{n\mu^2} + 1}\right),$$

and

$$\exp\left(\frac{1}{1 + \sqrt{\frac{2\overline{L}^2}{n\mu^2} + 1}}\frac{16K}{l}\right) \geq \frac{\|y_0 - y^*\|^2}{32\mathbb{E}\left[\|\hat{x} - x^*\|^2 + \|\hat{y} - y^*\|^2\right]}.$$

The next proof follow similar steps as in the proof of Theorem 2 from [63]. Let $\gamma_M = \frac{1 - \cos\frac{\pi}{M}}{1 + \cos\frac{\pi}{M}}$ be a decreasing sequence of positive numbers. Since $\gamma_2 = 1$ and $\lim_m \gamma_M = 0$, there exists $M \geq 2$ such that $\gamma_M \geq \chi^{-1} > \gamma_{M+1}$.

• If $M \geq 3$, let us consider linear graph of size $M$ with vertexes $v_1, \ldots v_M$, and weighted with $w_{1,2} = 1 - a$ and $w_{i,i+1} = 1$ for $i \geq 2$. Then we applied Lemmas 1 and 3 and get:

$$\|\hat{x} - x^*\|^2 + \|\hat{y} - y^*\|^2 \geq q^{\frac{2K}{l}}\frac{\|y_0 - y^*\|^2}{32}.$$

If $\mathbf{W}_a$ is the normalized Laplacian of the weighted graph $\mathcal{G}$, one can note that with $a = 0$, $\chi^{-1}(W_a) = \gamma_M$, with $a = 1 - \chi^{-1}(\mathbf{W}_a) = 0$. Hence, there exists $a \in (0; 1]$ such that $\chi^{-1}(\mathbf{W}_a) = \chi^{-1}$. Then $\chi^{-1} \geq \gamma_{M+1} \geq \frac{2}{(M+1)^2}$, and $M \geq \sqrt{2\chi} - 1 \geq \frac{\sqrt{\chi}}{4}$. Finally, $l = M - 1 \geq \frac{15M}{16} - 1 \geq \frac{15}{16}\left(\sqrt{2\chi} - 1\right) - 1 \geq \frac{\sqrt{\chi}}{5}$ since $\chi^{-1} \leq \gamma_3 = \frac{1}{3}$. Hence,

$$\exp\left(80\frac{K}{\sqrt{\chi}} \cdot \frac{1}{1 + \sqrt{\frac{2\overline{L}^2}{n\mu^2} + 1}}\right) \geq \frac{\|y_0 - y^*\|^2}{32(\|\hat{x} - x^*\|^2 + \|\hat{y} - y^*\|^2)}. \tag{25}$$

• If $M = 2$, we construct a totally connected network with 3 nodes with weight $w_{1,3} = a \in [0; 1]$. Let $W_a$ is the normalized Laplacian. If $a = 0$, then the network is a linear graph and $\chi^{-1}(\mathbf{W}_a) = \gamma_3 = \frac{1}{3}$. Hence, there exists $a \in [0; 1]$ such that $\chi^{-1}(\mathbf{W}_a) = \chi^{-1}$. Finally, $B = \{v_1\}$, $\bar{B} = \{v_3\}$ and $l \geq 1 \geq \frac{1}{2\sqrt{\chi^{-1}}}$. Whence it follows that in this case (25) is also valid.

The same way we can work with (but without considering graph):

$$\left(\frac{1}{q}\right)^{\frac{2N}{n}} \geq \frac{\|y_0 - y^*\|^2}{32\mathbb{E}\left[\|\hat{x} - x^*\|^2 + \|\hat{y} - y^*\|^2\right]}.$$

$\square$

## C.2 Time-varying network

**Theorem C.9** (Theorem 3.4). *Let $\overline{L} > \mu > 0$, $n \in \mathbb{N}$ (with $\overline{L}/\mu \geq \sqrt{n}$), $\hat{\chi} \geq 3$ and $K, N \in \mathbb{N}$. There exists a distributed saddle-point problem over time-varying network (Assumption 2.4). For which the following statements are true:*

- *Assumption 2.4 holds with $\chi = \hat{\chi}$,*

- $f = \sum\limits_{m=1}^{M} \frac{1}{n} \sum\limits_{i=1}^{n} f_{m.i} : \mathbb{R}^d \times \mathbb{R}^d \to \mathbb{R}$ *is $\mu$ – strongly-convex-strongly-concave,*

- $f_m$ *are $\overline{L}$-average smooth and $L = \frac{\overline{L}}{\sqrt{n}}$ - smooth,*

- *size $d \geq \max\left\{ 2\log_q\left(\frac{\alpha}{4\sqrt{2}}\right), 2K, 2N \right\}$, where $\alpha = \frac{2n\mu^2}{L^2}$ and $q = \frac{1}{2}\left(2 + \alpha - \sqrt{\alpha^2 + 4\alpha}\right) \in (0;1)$,*

- *the solution of the problem is non-zero: $x^* \neq 0$, $y^* \neq 0$.*

*Then for any output $\hat{z}$ of any procedure (Definition C.2) with $K$ communication rounds and $N$ local computations, one can obtain the following estimate:*

$$\|\hat{z} - z^*\|^2 = \Omega\left(\exp\left(-\frac{64}{1 + \sqrt{\frac{2L^2}{\mu^2} + 1}} \cdot \frac{K}{B\hat{\chi}}\right) \|y_0 - y^*\|^2\right);$$

$$\|\hat{z} - z^*\|^2 = \Omega\left(\exp\left(-\frac{16}{n + \sqrt{\frac{2n\overline{L}^2}{\mu^2} + n^2}} \cdot N\right) \|y_0 - y^*\|^2\right).$$

*Proof.* The same way as in the previous Theorem we can obtain

$$\exp\left(\frac{1}{1 + \sqrt{\frac{2\overline{L}^2}{n\mu^2} + 1}} \frac{16K}{l}\right) \geq \frac{\|y_0 - y^*\|^2}{32\mathbb{E}\left[\|\hat{x} - x^*\|^2 + \|\hat{y} - y^*\|^2\right]}.$$

Following [39], we can consider the next sequences of graphs. Let us choose $M = \lfloor\hat{\chi}\rfloor$. Each communication $t$ such that $t \neq 8\tau$ (for any $\tau \in \mathbb{N}$) we consider empty network. Each communication $t$ such that $t = 8\tau$ (for some $\tau \in \mathbb{N}$) we construct star graph with vertex $[(\tau - 1) \mod (M - 2)] + 2$ in the center of this star. It means that in the 8th communication vertex 2 is in the center; in the 16th communication vertex 3 is in the center etc. One can note that only vertexes with $f_3$ are in the center and they change sequentially. As matrices $\mathbf{W}(t)$ we consider the normalized Laplacians. Then it holds $\chi = M = \lfloor\hat{\chi}\rfloor$. It holds that Assumption 2.4 is valid with $\hat{\chi}$.

It is left to estimate $l$. Suppose we need to transfer information from $B$ to $\bar{B}$. At best, the following will happen:

- vertex $j$ in the center, it means that we can transfer information to $j$;

- $B - 1$ empty graphs – "empty" communications;

- vertex $j + 1$ in the center, it means that we can transfer information to $j + 1$;

...

- vertex $j - 1$ in the center, it means that we can transfer information to $j - 1$ (still there is no information in $\bar{B}$);

- $B - 1$ empty graphs – "empty" communications;

- vertex $j$ in the center, now we can transfer from $j$ to $\bar{B}$.

In this (the best variant) we spend $M - 1 + (B - 1)(M - 2)$ communication rounds. It means that $l \geq M - 1 + (B - 1)(M - 2) \geq B(M - 2) = B(\lfloor\hat{\chi}\rfloor - 2) \geq \frac{B\hat{\chi}}{4}$ (for $\hat{\chi} \geq 3$). Then we get

$$\exp\left(\frac{1}{1 + \sqrt{\frac{2\overline{L}^2}{n\mu^2} + 1}} \frac{64K}{B\hat{\chi}}\right) \geq \frac{\|y_0 - y^*\|^2}{32\mathbb{E}\left[\|\hat{x} - x^*\|^2 + \|\hat{y} - y^*\|^2\right]}.$$

$\square$

# D   Proof of Theorem 4.1

We start the proof from the following lemma on $\delta^k$ and $\Delta^{k+1/2}$ from Algorithm 1.

**Lemma D.1.** *The following inequality holds:*

$$\mathbb{E}_k\left[\left\|\delta^k - \mathbb{E}_k\left[\delta^k\right]\right\|^2\right] \le \frac{2\overline{L}^2}{b}\mathbb{E}\left[\left\|\mathbf{z}^k - \mathbf{w}^{k-1}\right\|^2 + \alpha^2\left\|\mathbf{z}^k - \mathbf{z}^{k-1}\right\|^2\right]. \tag{26}$$

$$\mathbb{E}_k\left[\left\|\Delta^{k+1/2} - \mathbf{F}(\mathbf{z}^*)\right\|^2\right] \le \frac{2\overline{L}^2}{b}\mathbb{E}\left[\left\|\mathbf{z}^{k+1} - \mathbf{w}^k\right\|^2\right] + 2L^2\mathbb{E}\left[\left\|\mathbf{z}^{k+1} - \mathbf{z}^*\right\|^2\right]. \tag{27}$$

*where $\mathbb{E}_k\left[\delta^k\right]$ is equal to*

$$\mathbb{E}_k\left[\delta^k\right] = F(\mathbf{z}^k) + \alpha(F(\mathbf{z}^k) - F(\mathbf{z}^{k-1})). \tag{28}$$

*Proof.* Due to the fact that the batch $S^k$ is generated uniformly and independently for all workers, we can make sure that (28) is correct. Then using definition of $\Delta^k$ from Algorithm 1, we get

$$\mathbb{E}_k\left[\left\|\delta^k - \mathbb{E}_k\left[\delta^k\right]\right\|^2\right] \le 2\mathbb{E}_k\left[\left\|\frac{1}{b}\sum_{j\in S^k}\left(\mathbf{F}_j(\mathbf{z}^k) - \mathbf{F}_j(\mathbf{w}^{k-1})\right) - \left(\mathbf{F}(\mathbf{z}^k) - \mathbf{F}(\mathbf{w}^{k-1})\right)\right\|^2\right]$$
$$+ 2\mathbb{E}_k\left[\left\|\frac{\alpha}{b}\sum_{j\in S^k}\left(\mathbf{F}_j(\mathbf{z}^k) - \mathbf{F}_j(\mathbf{z}^{k-1})\right) - \left(\mathbf{F}(\mathbf{z}^k) - \mathbf{F}(\mathbf{z}^{k-1})\right)\right\|^2\right]$$

By randomness and independence of indexes in $S^k$, we obtain

$$\mathbb{E}_k\left[\left\|\delta^k - \mathbb{E}_k\left[\delta^k\right]\right\|^2\right] = \frac{2}{b^2}\mathbb{E}_k\left[\sum_{j\in S^k}\left\|\left(\mathbf{F}_j(\mathbf{z}^k) - \mathbf{F}_j(\mathbf{w}^{k-1})\right) - \left(\mathbf{F}(\mathbf{z}^k) - \mathbf{F}(\mathbf{w}^{k-1})\right)\right\|^2\right]$$
$$+ \frac{2\alpha^2}{b^2}\mathbb{E}_k\left[\sum_{j\in S^k}\left\|\left(\mathbf{F}_j(\mathbf{z}^k) - \mathbf{F}_j(\mathbf{z}^{k-1})\right) - \left(\mathbf{F}(\mathbf{z}^k) - \mathbf{F}(\mathbf{z}^{k-1})\right)\right\|^2\right]$$

The property of the second moment $\mathbb{E}\|\xi - \mathbb{E}\xi\|^2 = \mathbb{E}\|\xi\|^2 - \|\mathbb{E}\xi\|^2$ gives

$$\mathbb{E}_k\left[\left\|\delta^k - \mathbb{E}_k\left[\delta^k\right]\right\|^2\right] \le \frac{2}{b^2}\mathbb{E}_k\left[\sum_{j\in S^k}\left\|\mathbf{F}_j(\mathbf{z}^k) - \mathbf{F}_j(\mathbf{w}^{k-1})\right\|^2\right]$$
$$+ \frac{2\alpha^2}{b^2}\mathbb{E}_k\left[\sum_{j\in S^k}\left\|\mathbf{F}_j(\mathbf{z}^k) - F_j(\mathbf{z}^{k-1})\right\|^2\right]$$

Again from the fact that $S^k$ is generated uniformly and independently for all samples and workers, we can obtain that for each worker $m$ indexes $j_{m,1}^k, \ldots j_{m,b}^k$ have the same uniform distribution, that means

$$\mathbb{E}_k\left[\left\|\delta^k - \mathbb{E}_k\left[\delta^k\right]\right\|^2\right] \le \frac{2}{b^2}\mathbb{E}_k\left[\sum_{j\in S^k}\left\|\mathbf{F}_j(\mathbf{z}^k) - \mathbf{F}_j(\mathbf{w}^{k-1})\right\|^2\right]$$
$$+ \frac{2\alpha^2}{b^2}\mathbb{E}_k\left[\sum_{j\in S^k}\left\|\mathbf{F}_j(\mathbf{z}^k) - F_j(\mathbf{z}^{k-1})\right\|^2\right]$$
$$= \frac{2}{mb}\sum_{j=1}^m\left(\left\|\mathbf{F}_j(\mathbf{z}^k) - \mathbf{F}_j(\mathbf{w}^{k-1})\right\|^2 + \alpha^2\left\|\mathbf{F}_j(\mathbf{z}^k) - \mathbf{F}_j(\mathbf{z}^{k-1})\right\|^2\right).$$

Using Assumption 2.1, we get

$$\mathbb{E}_k\left[\left\|\delta^k - \mathbb{E}_k\left[\delta^k\right]\right\|^2\right] \le \frac{2\overline{L}^2}{b}\left(\left\|\mathbf{z}^k - \mathbf{w}^{k-1}\right\|^2 + \alpha^2\left\|\mathbf{z}^k - \mathbf{z}^{k-1}\right\|^2\right).$$

This concludes the proof of (26).

The proof chain for (27) is very similar.

$$\mathbb{E}\left[\left\|\Delta^{k+1/2} - \mathbf{F}(\mathbf{z}^*)\right\|^2\right] \le 2\mathbb{E}\left[\left\|\Delta^{k+1/2} - \mathbf{F}(\mathbf{z}^{k+1})\right\|^2\right] + 2\mathbb{E}\left[\left\|\mathbf{F}(\mathbf{z}^{k+1}) - \mathbf{F}(\mathbf{z}^*)\right\|^2\right]$$

$$= \mathbb{E}\left[\frac{2}{b^2}\mathbb{E}_{k+1/2}\sum_{j\in S^{k+1/2}}\left\|\left(\mathbf{F}_j(\mathbf{z}^{k+1})-\mathbf{F}_j(\mathbf{w}^k)\right)-\left(\mathbf{F}(\mathbf{z}^{k+1})-\mathbf{F}(\mathbf{w}^k)\right)\right\|^2\right]$$
$$+2L^2\mathbb{E}\left[\left\|\mathbf{z}^{k+1}-\mathbf{z}^*\right\|^2\right]$$
$$\leq \mathbb{E}\left[\frac{2}{b^2}\mathbb{E}_{k+1/2}\sum_{j\in S^{k+1/2}}\left\|\mathbf{F}_j(\mathbf{z}^{k+1})-\mathbf{F}_j(\mathbf{w}^k)\right\|^2\right]+2L^2\mathbb{E}\left[\left\|\mathbf{z}^{k+1}-\mathbf{z}^*\right\|^2\right]$$
$$\leq \frac{2\overline{L}^2}{b}\mathbb{E}\left[\left\|\mathbf{z}^{k+1}-\mathbf{w}^k\right\|^2\right]+2L^2\mathbb{E}\left[\left\|\mathbf{z}^{k+1}-\mathbf{z}^*\right\|^2\right].$$

Here, we used Assumption 2.1. $\qquad\square$

Before proving the main lemma of this section, let us introduce an auxiliary notation. Throughout the proof, we denote $\mathbf{W}^\dagger : \mathrm{range}\,\mathbf{W}\to\mathrm{range}\,\mathbf{W}$ the inverse of the map $\mathbf{W}:\mathrm{range}\,\mathbf{W}\to\mathrm{range}\,\mathbf{W}$. And we denote $\|\mathbf{y}\|^2_{(\mathbf{W}^\dagger\otimes\mathbf{I}_d)}=\langle(\mathbf{W}^\dagger\otimes\mathbf{I}_d)y;y\rangle$.

We define the following Lyapunov function:

$$\Psi^k = \left(\frac{1}{\eta}+\frac{3\mu}{2}\right)\left\|\mathbf{z}^{k+1}-\mathbf{z}^*\right\|^2+\frac{1}{\theta}\left\|\mathbf{y}^{k+1}-\mathbf{y}^*\right\|^2_{(\mathbf{W}^\dagger\otimes\mathbf{I}_d)}$$
$$+2\langle\mathbf{F}(\mathbf{z}^k)-\mathbf{F}(\mathbf{z}^{k+1})-(\mathbf{y}^k-\mathbf{y}^{k+1}),\mathbf{z}^{k+1}-\mathbf{z}^*\rangle+\frac{1}{2\theta}\left\|\mathbf{y}^{k+1}-\mathbf{y}^k\right\|^2 \qquad (29)$$
$$+\frac{1}{8\eta}\left\|\mathbf{z}^{k+1}-\mathbf{z}^k\right\|^2+\frac{\gamma+\frac{1}{2}\eta\mu}{p\eta}\left\|\mathbf{w}^{k+1}-\mathbf{z}^*\right\|^2+\frac{\gamma}{2\eta}\left\|\mathbf{w}^k-\mathbf{z}^{k+1}\right\|^2.$$

Here we also use

$$\mathbf{y}^* = \mathbf{P}\mathbf{F}(\mathbf{z}^*) \qquad (30)$$

with $\mathbf{P}\in\mathbb{R}^{nd\times nd}$, an orthogonal projection matrix onto the subspace $\mathcal{L}^\perp$, given as

$$\mathbf{P}=(\mathbf{I}_M-\tfrac{1}{M}\mathbf{1}_M\mathbf{1}_M^\top)\otimes\mathbf{I}_d,$$

where $\mathbf{1}_M=(1,\dots,1)^\top\in\mathbb{R}^M$.

**Lemma D.2.** *Consider the problem* (10) *(or* (1) *+* (5)*) under Assumptions* 2.1 *and* 2.2 *over a fixed connected graph* $\mathcal{G}$ *with a gossip matrix* $\mathbf{W}$*. Let* $\{\mathbf{z}^k\}$ *be the sequence generated by Algorithm* 1 *with parameters*

$$\gamma\leq\frac{1}{8},\quad \eta\leq\min\left\{\frac{\sqrt{\alpha\gamma b}}{\sqrt{8}\cdot\overline{L}};\frac{1}{16L}\right\},\quad \beta\leq\min\left\{\frac{\mu}{4L^2};\frac{b\gamma}{4\eta\overline{L}^2}\right\},\quad \theta\leq\min\left\{\frac{1}{2\beta};\frac{1}{16\eta}\right\},$$
$$\alpha=\max\left[\left(1-\frac{\mu\eta}{4}\right);\left(1-\beta\theta\chi^{-1}\right);\left(1-\frac{p\eta\mu}{2\gamma+\eta\mu}\right)\right].$$

*Then, after $k$ iterations we get*

$$\mathbb{E}\left[\frac{1}{2\eta}\|\mathbf{z}^k-\mathbf{z}^*\|^2\right]\leq\max\left[\left(1-\frac{\mu\eta}{4}\right);\left(1-\beta\theta\chi^{-1}\right);\left(1-\frac{p\eta\mu}{2\gamma+\eta\mu}\right)\right]^k\cdot\Psi^0.$$

*Proof.* **Part 1.** We start the proof from considering update of $\mathbf{y}^{k+1}$ in Algorithm 1.

$$\frac{1}{\theta}\left\|\mathbf{y}^{k+1}-\mathbf{y}^*\right\|^2_{(\mathbf{W}^\dagger\otimes\mathbf{I}_d)}=\frac{1}{\theta}\left\|\mathbf{y}^k-\mathbf{y}^*\right\|^2_{(\mathbf{W}^\dagger\otimes\mathbf{I}_d)}-\frac{1}{\theta}\left\|\mathbf{y}^{k+1}-\mathbf{y}^k\right\|^2_{(\mathbf{W}^\dagger\otimes\mathbf{I}_d)}$$
$$-2\langle(\mathbf{W}\otimes\mathbf{I}_d)(\mathbf{z}^{k+1}-\beta(\Delta^{k+1/2}-\mathbf{y}^k)),(\mathbf{W}^\dagger\otimes\mathbf{I}_d)(\mathbf{y}^{k+1}-\mathbf{y}^*)\rangle.$$

Next, we use the fact that $(\mathbf{W}\otimes\mathbf{I}_d)(\mathbf{W}^\dagger\otimes\mathbf{I}_d)=\mathbf{P}$ and obtain

$$\frac{1}{\theta}\left\|\mathbf{y}^{k+1}-\mathbf{y}^*\right\|^2_{(\mathbf{W}^\dagger\otimes\mathbf{I}_d)}=\frac{1}{\theta}\left\|\mathbf{y}^k-\mathbf{y}^*\right\|^2_{(\mathbf{W}^\dagger\otimes\mathbf{I}_d)}-\frac{1}{\theta}\left\|\mathbf{y}^{k+1}-\mathbf{y}^k\right\|^2_{(\mathbf{W}^\dagger\otimes\mathbf{I}_d)}$$
$$-2\langle\mathbf{P}(\mathbf{z}^{k+1}-\beta(\Delta^{k+1/2}-\mathbf{y}^k)),\mathbf{y}^{k+1}-\mathbf{y}^*\rangle.$$

One can observe, that $\mathbf{z}^*\in\mathcal{L}$ and then $\mathbf{P}\mathbf{z}^*=0$. Additionally, using update $\mathbf{y}^{k+1}$, we can note that $\mathbf{y}^k\in\mathcal{L}^\perp$ for all $k=0,1,2,\dots$. Hence,

$$\frac{1}{\theta}\left\|\mathbf{y}^{k+1}-\mathbf{y}^*\right\|^2_{(\mathbf{W}^\dagger\otimes\mathbf{I}_d)}=\frac{1}{\theta}\left\|\mathbf{y}^k-\mathbf{y}^*\right\|^2_{(\mathbf{W}^\dagger\otimes\mathbf{I}_d)}-\frac{1}{\theta}\left\|\mathbf{y}^{k+1}-\mathbf{y}^k\right\|^2_{(\mathbf{W}^\dagger\otimes\mathbf{I}_d)}$$

$$- 2\langle \mathbf{P}(\mathbf{z}^{k+1} - \mathbf{z}^*), \mathbf{y}^{k+1} - \mathbf{y}^* \rangle + 2\beta \langle \mathbf{P}(\Delta^{k+1/2} - \mathbf{F}(\mathbf{z}^*)), \mathbf{y}^{k+1} - \mathbf{y}^* \rangle$$
$$- 2\beta \langle \mathbf{P}\mathbf{y}^k - \mathbf{y}^*, \mathbf{y}^{k+1} - \mathbf{y}^* \rangle$$
$$= \frac{1}{\theta} \left\| \mathbf{y}^k - \mathbf{y}^* \right\|_{(\mathbf{W}^\dagger \otimes \mathbf{I}_d)}^2 - \frac{1}{\theta} \left\| \mathbf{y}^{k+1} - \mathbf{y}^k \right\|_{(\mathbf{W}^\dagger \otimes \mathbf{I}_d)}^2$$
$$- 2\langle \mathbf{P}(\mathbf{z}^{k+1} - \mathbf{z}^*), \mathbf{y}^{k+1} - \mathbf{y}^* \rangle + 2\beta \langle \mathbf{P}(\Delta^{k+1/2} - \mathbf{F}(\mathbf{z}^*)), \mathbf{y}^{k+1} - \mathbf{y}^* \rangle$$
$$- 2\beta \langle \mathbf{P}(\mathbf{y}^k - \mathbf{y}^*), \mathbf{y}^{k+1} - \mathbf{y}^* \rangle$$
$$= \frac{1}{\theta} \left\| \mathbf{y}^k - \mathbf{y}^* \right\|_{(\mathbf{W}^\dagger \otimes \mathbf{I}_d)}^2 - \frac{1}{\theta} \left\| \mathbf{y}^{k+1} - \mathbf{y}^k \right\|_{(\mathbf{W}^\dagger \otimes \mathbf{I}_d)}^2$$
$$- 2\langle \mathbf{z}^{k+1} - \mathbf{z}^*, \mathbf{y}^{k+1} - \mathbf{y}^* \rangle + 2\beta \langle \Delta^{k+1/2} - \mathbf{F}(\mathbf{z}^*), \mathbf{y}^{k+1} - \mathbf{y}^* \rangle$$
$$- 2\beta \langle \mathbf{y}^k - \mathbf{y}^*, \mathbf{y}^{k+1} - \mathbf{y}^* \rangle$$

By the simple fact $\|a + b\|^2 = \|a\|^2 + 2\langle a; b \rangle + \|b\|^2$, we get

$$\frac{1}{\theta} \left\| \mathbf{y}^{k+1} - \mathbf{y}^* \right\|_{(\mathbf{W}^\dagger \otimes \mathbf{I}_d)}^2 \leq \frac{1}{\theta} \left\| \mathbf{y}^k - \mathbf{y}^* \right\|_{(\mathbf{W}^\dagger \otimes \mathbf{I}_d)}^2 - \frac{1}{\theta} \left\| \mathbf{y}^{k+1} - \mathbf{y}^k \right\|_{(\mathbf{W}^\dagger \otimes \mathbf{I}_d)}^2$$
$$- 2\langle \mathbf{z}^{k+1} - \mathbf{z}^*, \mathbf{y}^{k+1} - \mathbf{y}^* \rangle + \beta \left\| \Delta^{k+1/2} - \mathbf{F}(\mathbf{z}^*) \right\|^2 + \beta \left\| \mathbf{y}^{k+1} - \mathbf{y}^* \right\|^2$$
$$- \beta \left\| \mathbf{y}^k - \mathbf{y}^* \right\|^2 - \beta \left\| \mathbf{y}^{k+1} - \mathbf{y}^* \right\|^2 + \beta \left\| \mathbf{y}^{k+1} - \mathbf{y}^k \right\|^2$$
$$= \frac{1}{\theta} \left\| \mathbf{y}^k - \mathbf{y}^* \right\|_{(\mathbf{W}^\dagger \otimes \mathbf{I}_d)}^2 - \frac{1}{\theta} \left\| \mathbf{y}^{k+1} - \mathbf{y}^k \right\|_{(\mathbf{W}^\dagger \otimes \mathbf{I}_d)}^2$$
$$- 2\langle \mathbf{z}^{k+1} - \mathbf{z}^*, \mathbf{y}^{k+1} - \mathbf{y}^* \rangle + \beta \left\| \Delta^{k+1/2} - \mathbf{F}(\mathbf{z}^*) \right\|^2 - \beta \left\| \mathbf{y}^k - \mathbf{y}^* \right\|^2$$
$$+ \beta \left\| \mathbf{y}^{k+1} - \mathbf{y}^k \right\|^2 .$$

Using the fact, that $\lambda_{\min}^+(\mathbf{W}) = \chi^{-1}$ and $\lambda_{\max}(\mathbf{W}) = 1$, we can note $-\lambda_{\min}^+(\mathbf{W}^\dagger) \leq -1$ and $\chi^{-1}\lambda_{\max}(\mathbf{W}^\dagger) \leq 1$ and then get

$$\frac{1}{\theta} \left\| \mathbf{y}^{k+1} - \mathbf{y}^* \right\|_{\mathbf{W}^\dagger \otimes \mathbf{I}_d}^2 \leq \frac{1}{\theta} \left\| \mathbf{y}^k - \mathbf{y}^* \right\|_{\mathbf{W}^\dagger \otimes \mathbf{I}_d}^2 - \frac{1}{\theta} \left\| \mathbf{y}^{k+1} - \mathbf{y}^k \right\|^2 - 2\langle \mathbf{z}^{k+1} - \mathbf{z}^*, \mathbf{y}^{k+1} - \mathbf{y}^* \rangle$$
$$+ \beta \left\| \Delta^{k+1/2} - \mathbf{F}(\mathbf{z}^*) \right\|^2 - \beta\chi^{-1} \left\| \mathbf{y}^k - \mathbf{y}^* \right\|_{\mathbf{W}^\dagger \otimes \mathbf{I}_d}^2 + \beta \left\| \mathbf{y}^{k+1} - \mathbf{y}^k \right\|^2$$
$$= \left( \frac{1}{\theta} - \beta\chi^{-1} \right) \left\| \mathbf{y}^k - \mathbf{y}^* \right\|_{(\mathbf{W}^\dagger \otimes \mathbf{I}_d)}^2 - \left( \frac{1}{\theta} - \beta \right) \left\| \mathbf{y}^{k+1} - \mathbf{y}^k \right\|^2$$
$$- 2\langle \mathbf{z}^{k+1} - \mathbf{z}^*, \mathbf{y}^{k+1} - \mathbf{y}^* \rangle + \beta \left\| \Delta^{k+1/2} - \mathbf{F}(\mathbf{z}^*) \right\|^2 .$$

Then we take a full expectation and use (27):

$$\mathbb{E}\left[ \frac{1}{\theta} \left\| \mathbf{y}^{k+1} - \mathbf{y}^* \right\|_{(\mathbf{W}^\dagger \otimes \mathbf{I}_d)}^2 \right] \leq \left( \frac{1}{\theta} - \beta\chi^{-1} \right) \mathbb{E}\left[ \left\| \mathbf{y}^k - \mathbf{y}^* \right\|_{(\mathbf{W}^\dagger \otimes \mathbf{I}_d)}^2 \right] - \left( \frac{1}{\theta} - \beta \right) \mathbb{E}\left[ \left\| \mathbf{y}^{k+1} - \mathbf{y}^k \right\|^2 \right]$$
$$- 2\mathbb{E}\left[ \langle \mathbf{z}^{k+1} - \mathbf{z}^*, \mathbf{y}^{k+1} - \mathbf{y}^* \rangle \right] + 2\beta L^2 \mathbb{E}\left[ \left\| \mathbf{z}^{k+1} - \mathbf{z}^* \right\|^2 \right]$$
$$+ \frac{2\beta \overline{L}^2}{b} \mathbb{E}\left[ \left\| \mathbf{z}^{k+1} - \mathbf{w}^k \right\|^2 \right] .$$
$$\tag{31}$$

Here we stop and put aside (31). We will not return to it later.

**Part 2.** Now we work with

$$\frac{1}{\eta} \left\| \mathbf{z}^{k+1} - \mathbf{z}^* \right\|^2 = \frac{1}{\eta} \left\| \mathbf{z}^k - \mathbf{z}^* \right\|^2 + \frac{2}{\eta} \langle \mathbf{z}^{k+1} - \mathbf{z}^k, \mathbf{z}^{k+1} - \mathbf{z}^* \rangle - \frac{1}{\eta} \left\| \mathbf{z}^{k+1} - \mathbf{z}^k \right\|^2$$
$$= \frac{1}{\eta} \left\| \mathbf{z}^k - \mathbf{z}^* \right\|^2 + \frac{2\gamma}{\eta} \langle \mathbf{w}^k - \mathbf{z}^k, \mathbf{z}^{k+1} - \mathbf{z}^* \rangle - 2\langle \Delta^k - (\mathbf{F}(\mathbf{z}^*) - \mathbf{y}^*), \mathbf{z}^{k+1} - \mathbf{z}^* \rangle$$
$$- \frac{1}{\eta} \left\| \mathbf{z}^{k+1} - \mathbf{z}^k \right\|^2$$
$$- 2\langle \frac{1}{\eta}(\mathbf{z}^k + \gamma(\mathbf{w}^k - \mathbf{z}^k) - \eta\Delta^k - \mathbf{z}^{k+1}) + (\mathbf{F}(\mathbf{z}^*) - \mathbf{y}^*), \mathbf{z}^{k+1} - \mathbf{z}^* \rangle .$$

Optimality condition for (1)+(5) it follows, that

$$-F(z^*) \in \partial g(z^*).$$

Let us define $\Delta^* \in (\mathbb{R}^d)^M$ as

$$\Delta^* = \tfrac{1}{M}[F(z^*), \dots F(z^*)]^T.$$

It is to note that $-\Delta^* \in M\partial \mathbf{g}(\mathbf{z}^*)$. On the other hand

$$\Delta^* = (\mathbf{1}_M \mathbf{1}_M^\top \otimes \mathbf{I}_d)\mathbf{F}(\mathbf{z}^*) = M(\mathbf{F}(\mathbf{z}^*) - \mathbf{P}\mathbf{F}(\mathbf{z}^*)) = M(\mathbf{F}(\mathbf{z}^*) - \mathbf{y}^*))$$

This means that $-(\mathbf{F}(\mathbf{z}^*) - \mathbf{y}^*) \in \partial \mathbf{g}(\mathbf{z}^*)$. From update for $\mathbf{z}^{k+1}$ of Algorithm 1 it follows, that

$$\mathbf{z}^k + \gamma(\mathbf{w}^k - \mathbf{z}^k) - \eta\Delta^k - \mathbf{z}^{k+1} \in \partial(\eta\mathbf{g})(\mathbf{z}^{k+1}).$$

Hence, from monotonicity of $\partial \mathbf{g}(\cdot)$ we get

$$\frac{1}{\eta}\mathbb{E}\left[\left\|\mathbf{z}^{k+1} - \mathbf{z}^*\right\|^2\right] \le \frac{1}{\eta}\mathbb{E}\left[\left\|\mathbf{z}^k - \mathbf{z}^*\right\|^2\right] + \frac{2\gamma}{\eta}\mathbb{E}\left[\langle\mathbf{w}^k - \mathbf{z}^k, \mathbf{z}^{k+1} - \mathbf{z}^*\rangle\right]$$

$$- 2\mathbb{E}\left[\langle\Delta^k - (\mathbf{F}(\mathbf{z}^*) - \mathbf{y}^*), \mathbf{z}^{k+1} - \mathbf{z}^*\rangle\right] - \frac{1}{\eta}\mathbb{E}\left[\left\|\mathbf{z}^{k+1} - \mathbf{z}^k\right\|^2\right]$$

$$= \frac{1}{\eta}\mathbb{E}\left[\left\|\mathbf{z}^k - \mathbf{z}^*\right\|^2\right] + \frac{2\gamma}{\eta}\mathbb{E}\left[\langle\mathbf{w}^k - \mathbf{z}^*, \mathbf{z}^{k+1} - \mathbf{z}^*\rangle\right]$$

$$- \frac{2\gamma}{\eta}\mathbb{E}\left[\langle\mathbf{z}^k - \mathbf{z}^*, \mathbf{z}^{k+1} - \mathbf{z}^*\rangle\right] - 2\mathbb{E}\left[\langle\Delta^k - (\mathbf{F}(\mathbf{z}^*) - \mathbf{y}^*), \mathbf{z}^{k+1} - \mathbf{z}^*\rangle\right]$$

$$- \frac{1}{\eta}\mathbb{E}\left[\left\|\mathbf{z}^{k+1} - \mathbf{z}^k\right\|^2\right]$$

$$= \frac{1}{\eta}\mathbb{E}\left[\left\|\mathbf{z}^k - \mathbf{z}^*\right\|^2\right] + \frac{\gamma}{\eta}\mathbb{E}\left[\left\|\mathbf{w}^k - \mathbf{z}^*\right\|^2 + \left\|\mathbf{z}^{k+1} - \mathbf{z}^*\right\|^2 - \left\|\mathbf{z}^{k+1} - \mathbf{w}^k\right\|^2\right]$$

$$- \frac{\gamma}{\eta}\mathbb{E}\left[\left\|\mathbf{z}^{k+1} - \mathbf{z}^*\right\|^2 + \left\|\mathbf{z}^k - \mathbf{z}^*\right\|^2 - \left\|\mathbf{z}^{k+1} - \mathbf{z}^k\right\|^2\right]$$

$$- 2\mathbb{E}\left[\langle\Delta^k - (\mathbf{F}(\mathbf{z}^*) - \mathbf{y}^*), \mathbf{z}^{k+1} - \mathbf{z}^*\rangle\right] - \frac{1}{\eta}\mathbb{E}\left[\left\|\mathbf{z}^{k+1} - \mathbf{z}^k\right\|^2\right]$$

$$= \frac{1}{\eta}\mathbb{E}\left[\left\|\mathbf{z}^k - \mathbf{z}^*\right\|^2\right] + \frac{\gamma}{\eta}\mathbb{E}\left[\left\|\mathbf{w}^k - \mathbf{z}^*\right\|^2\right] - \frac{\gamma}{\eta}\mathbb{E}\left[\left\|\mathbf{z}^k - \mathbf{z}^*\right\|^2\right]$$

$$- \frac{\gamma}{\eta}\mathbb{E}\left[\left\|\mathbf{w}^k - \mathbf{z}^{k+1}\right\|^2\right] - 2\mathbb{E}\left[\langle\Delta^k - (\mathbf{F}(\mathbf{z}^*) - \mathbf{y}^*), \mathbf{z}^{k+1} - \mathbf{z}^*\rangle\right]$$

$$- \frac{1 - \gamma}{\eta}\mathbb{E}\left[\left\|\mathbf{z}^{k+1} - \mathbf{z}^k\right\|^2\right].$$

In previous we also use the simple fact $\|a + b\|^2 = \|a\|^2 + 2\langle a; b\rangle + \|b\|^2$ twice. Small rearrangement gives

$$\frac{1}{\eta}\mathbb{E}\left[\left\|\mathbf{z}^{k+1} - \mathbf{z}^*\right\|^2\right] \le \frac{1}{\eta}\mathbb{E}\left[\left\|\mathbf{z}^k - \mathbf{z}^*\right\|^2\right] + \frac{\gamma}{\eta}\mathbb{E}\left[\left\|\mathbf{w}^k - \mathbf{z}^*\right\|^2\right] - \frac{\gamma}{\eta}\mathbb{E}\left[\left\|\mathbf{z}^k - \mathbf{z}^*\right\|^2\right]$$

$$- \frac{\gamma}{\eta}\mathbb{E}\left[\left\|\mathbf{w}^k - \mathbf{z}^{k+1}\right\|^2\right] - \frac{1 - \gamma}{\eta}\mathbb{E}\left[\left\|\mathbf{z}^{k+1} - \mathbf{z}^k\right\|^2\right]$$

$$- 2\mathbb{E}\left[\langle\mathbb{E}_k\left[\delta^k\right] - (\mathbf{y}^k + \alpha(\mathbf{y}^k - \mathbf{y}^{k-1})) - (\mathbf{F}(\mathbf{z}^*) - \mathbf{y}^*), \mathbf{z}^{k+1} - \mathbf{z}^*\rangle\right]$$

$$- 2\mathbb{E}\left[\langle\delta^k - \mathbb{E}_k\left[\delta^k\right], \mathbf{z}^{k+1} - \mathbf{z}^k\rangle\right] - 2\mathbb{E}\left[\langle\delta^k - \mathbb{E}_k\left[\delta^k\right], \mathbf{z}^k - \mathbf{z}^*\rangle\right].$$

Using the tower property of expectation we can obtain the following:

$$\mathbb{E}\left[\langle\mathbb{E}_k\left[\delta^k\right] - \delta^k, \mathbf{z}^k - \mathbf{z}^*\rangle\right] = \mathbb{E}\left[\mathbb{E}_k\left[\langle\mathbb{E}_k\left[\delta^k\right] - \delta^k, \mathbf{z}^k - \mathbf{z}^*\rangle\right]\right]$$

$$= \mathbb{E}\left[\langle\mathbb{E}_k\left[\mathbb{E}_k\left[\delta^k\right] - \delta^k\right], \mathbf{z}^k - \mathbf{z}^*\rangle\right]$$

$$= \mathbb{E}\left[\langle\mathbb{E}_k\left[\delta^k\right] - \mathbb{E}_k\left[\delta^k\right], \mathbf{z}^k - \mathbf{z}^*\rangle\right] = 0.$$

Hence, with (28)

$$\frac{1}{\eta}\mathbb{E}\left[\left\|\mathbf{z}^{k+1} - \mathbf{z}^*\right\|^2\right] \le \frac{1}{\eta}\mathbb{E}\left[\left\|\mathbf{z}^k - \mathbf{z}^*\right\|^2\right] + \frac{\gamma}{\eta}\mathbb{E}\left[\left\|\mathbf{z}^k - \mathbf{z}*\right\|^2\right] - \frac{\gamma}{\eta}\mathbb{E}\left[\left\|\mathbf{z}^k - \mathbf{z}^*\right\|^2\right]$$

$$- \frac{\gamma}{\eta} \mathbb{E}\left[\left\|\mathbf{w}^k - \mathbf{z}^{k+1}\right\|^2\right] - \frac{1-\gamma}{\eta} \mathbb{E}\left[\left\|\mathbf{z}^{k+1} - \mathbf{z}^k\right\|^2\right]$$

$$- 2\mathbb{E}\left[\langle \mathbf{F}(\mathbf{z}^k) + \alpha(\mathbf{F}(\mathbf{z}^k) - \mathbf{F}(\mathbf{z}^{k-1})) - \mathbf{F}(\mathbf{z}^*) - (\mathbf{y}^k + \alpha(\mathbf{y}^k - \mathbf{y}^{k-1}) - \mathbf{y}^*), \mathbf{z}^{k+1} - \mathbf{z}^*\rangle\right]$$

$$+ 2\mathbb{E}\left[\langle \mathbb{E}_k\left[\delta^k\right] - \delta^k, \mathbf{z}^{k+1} - \mathbf{z}^k\rangle\right].$$

Using the Young's inequality we get

$$\frac{1}{\eta} \mathbb{E}\left[\left\|\mathbf{z}^{k+1} - \mathbf{z}^*\right\|^2\right] \leq \frac{1}{\eta} \mathbb{E}\left[\left\|\mathbf{z}^k - \mathbf{z}^*\right\|^2\right] + \frac{\gamma}{\eta} \mathbb{E}\left[\left\|\mathbf{w}^k - \mathbf{z}^*\right\|^2\right] - \frac{\gamma}{\eta} \mathbb{E}\left[\left\|\mathbf{z}^k - \mathbf{z}^*\right\|^2\right]$$

$$- \frac{\gamma}{\eta} \mathbb{E}\left[\left\|\mathbf{w}^k - \mathbf{z}^{k+1}\right\|^2\right] - \frac{1-\gamma}{\eta} \mathbb{E}\left[\left\|\mathbf{z}^{k+1} - \mathbf{z}^k\right\|^2\right]$$

$$- 2\mathbb{E}\left[\langle \mathbf{F}(\mathbf{z}^k) + \alpha(\mathbf{F}(\mathbf{z}^k) - \mathbf{F}(\mathbf{z}^{k-1})) - \mathbf{F}(\mathbf{z}^*) - (\mathbf{y}^k + \alpha(\mathbf{y}^k - \mathbf{y}^{k-1}) - \mathbf{y}^*), \mathbf{z}^{k+1} - \mathbf{z}^*\rangle\right]$$

$$+ 2\eta \mathbb{E}\left[\left\|\mathbb{E}_k\left[\delta^k\right] - \delta^k\right\|^2\right] + \frac{1}{2\eta} \mathbb{E}\left[\left\|\mathbf{z}^{k+1} - \mathbf{z}^k\right\|^2\right]$$

$$= \frac{1}{\eta} \mathbb{E}\left[\left\|\mathbf{z}^k - \mathbf{z}^*\right\|^2\right] + \frac{\gamma}{\eta} \mathbb{E}\left[\left\|\mathbf{w}^k - \mathbf{z}^*\right\|^2\right] - \frac{\gamma}{\eta} \mathbb{E}\left[\left\|\mathbf{z}^k - \mathbf{z}^*\right\|^2\right]$$

$$- \frac{\gamma}{\eta} \mathbb{E}\left[\left\|\mathbf{w}^k - \mathbf{z}^{k+1}\right\|^2\right] - \frac{1/2 - \gamma}{\eta} \mathbb{E}\left[\left\|\mathbf{z}^{k+1} - \mathbf{z}^k\right\|^2\right] + 2\eta \mathbb{E}\left[\left\|\mathbb{E}_k\left[\delta^k\right] - \delta^k\right\|^2\right]$$

$$- 2\mathbb{E}\left[\langle \mathbf{F}(\mathbf{z}^k) + \alpha(\mathbf{F}(\mathbf{z}^k) - \mathbf{F}(\mathbf{z}^{k-1})) - \mathbf{F}(\mathbf{z}^*) - (\mathbf{y}^k + \alpha(\mathbf{y}^k - \mathbf{y}^{k-1}) - \mathbf{y}^*), \mathbf{z}^{k+1} - \mathbf{z}^*\rangle\right].$$

By (26) we get

$$\frac{1}{\eta} \mathbb{E}\left[\left\|\mathbf{z}^{k+1} - \mathbf{z}^*\right\|^2\right] \leq \frac{1}{\eta} \mathbb{E}\left[\left\|\mathbf{z}^k - \mathbf{z}^*\right\|^2\right] + \frac{\gamma}{\eta} \mathbb{E}\left[\left\|\mathbf{w}^k - \mathbf{z}^*\right\|^2\right] - \frac{\gamma}{\eta} \mathbb{E}\left[\left\|\mathbf{z}^k - \mathbf{z}^*\right\|^2\right]$$

$$- \frac{\gamma}{\eta} \mathbb{E}\left[\left\|\mathbf{w}^k - \mathbf{z}^{k+1}\right\|^2\right] - \frac{1/2 - \gamma}{\eta} \mathbb{E}\left[\left\|\mathbf{z}^{k+1} - \mathbf{z}^k\right\|^2\right]$$

$$+ \frac{4\overline{L}^2 \eta}{b} \mathbb{E}\left[\left\|\mathbf{z}^k - \mathbf{w}^{k-1}\right\|^2\right] + \frac{4\alpha^2 \overline{L}^2 \eta}{b} \mathbb{E}\left[\left\|\mathbf{z}^k - \mathbf{z}^{k-1}\right\|^2\right]$$

$$- 2\mathbb{E}\left[\langle \mathbf{F}(\mathbf{z}^k) + \alpha(\mathbf{F}(\mathbf{z}^k) - \mathbf{F}(\mathbf{z}^{k-1})) - \mathbf{F}(\mathbf{z}^*) - (\mathbf{y}^k + \alpha(\mathbf{y}^k - \mathbf{y}^{k-1}) - \mathbf{y}^*), \mathbf{z}^{k+1} - \mathbf{z}^*\rangle\right]$$

$$= \frac{1}{\eta} \mathbb{E}\left[\left\|\mathbf{z}^k - \mathbf{z}^*\right\|^2\right] - 2\mathbb{E}\left[\langle \mathbf{F}(\mathbf{z}^{k+1}) - \mathbf{F}(\mathbf{z}^*), \mathbf{z}^{k+1} - \mathbf{z}^*\rangle\right] + \frac{\gamma}{\eta} \mathbb{E}\left[\left\|\mathbf{w}^k - \mathbf{z}^*\right\|^2\right]$$

$$- \frac{\gamma}{\eta} \mathbb{E}\left[\left\|\mathbf{z}^k - \mathbf{z}^*\right\|^2\right] - \frac{\gamma}{\eta} \mathbb{E}\left[\left\|\mathbf{w}^k - \mathbf{z}^{k+1}\right\|^2\right] - \frac{1/2 - \gamma}{\eta} \mathbb{E}\left[\left\|\mathbf{z}^{k+1} - \mathbf{z}^k\right\|^2\right]$$

$$+ \frac{4\overline{L}^2 \eta}{b} \mathbb{E}\left[\left\|\mathbf{z}^k - \mathbf{w}^{k-1}\right\|^2\right] + \frac{4\alpha^2 \overline{L}^2 \eta}{b} \mathbb{E}\left[\left\|\mathbf{z}^k - \mathbf{z}^{k-1}\right\|^2\right]$$

$$- 2\mathbb{E}\left[\langle \mathbf{F}(\mathbf{z}^k) - \mathbf{F}(\mathbf{z}^{k+1}) + \alpha(\mathbf{F}(\mathbf{z}^k) - \mathbf{F}(\mathbf{z}^{k-1})) - (\mathbf{y}^k + \alpha(\mathbf{y}^k - \mathbf{y}^{k-1}) - \mathbf{y}^*), \mathbf{z}^{k+1} - \mathbf{z}^*\rangle\right].$$

Assumption 2.2 about $\mu$-strong monotonicity of $F$ gives

$$\frac{1}{\eta} \mathbb{E}\left[\left\|\mathbf{z}^{k+1} - \mathbf{z}^*\right\|^2\right]$$

$$\leq \frac{1}{\eta} \mathbb{E}\left[\left\|\mathbf{z}^k - \mathbf{z}^*\right\|^2\right] - 2\mu \mathbb{E}\left[\left\|\mathbf{z}^{k+1} - \mathbf{z}^*\right\|^2\right] + \frac{\gamma}{\eta} \mathbb{E}\left[\left\|\mathbf{w}^k - \mathbf{z}^*\right\|^2\right] - \frac{\gamma}{\eta} \mathbb{E}\left[\left\|\mathbf{z}^k - \mathbf{z}^*\right\|^2\right]$$

$$- \frac{\gamma}{\eta} \mathbb{E}\left[\left\|\mathbf{w}^k - \mathbf{z}^{k+1}\right\|^2\right] - \frac{1/2 - \gamma}{\eta} \mathbb{E}\left[\left\|\mathbf{z}^{k+1} - \mathbf{z}^k\right\|^2\right]$$

$$+ \frac{4\overline{L}^2 \eta}{b} \mathbb{E}\left[\left\|\mathbf{z}^k - \mathbf{w}^{k-1}\right\|^2\right] + \frac{4\alpha^2 \overline{L}^2 \eta}{b} \mathbb{E}\left[\left\|\mathbf{z}^k - \mathbf{z}^{k-1}\right\|^2\right]$$

$$- 2\mathbb{E}\left[\langle \mathbf{F}(\mathbf{z}^k) - \mathbf{F}(\mathbf{z}^{k+1}) + \alpha(\mathbf{F}(\mathbf{z}^k) - \mathbf{F}(\mathbf{z}^{k-1})) - (\mathbf{y}^k + \alpha(\mathbf{y}^k - \mathbf{y}^{k-1}) - \mathbf{y}^*), \mathbf{z}^{k+1} - \mathbf{z}^*\rangle\right]$$

$$= \frac{1}{\eta} \mathbb{E}\left[\left\|\mathbf{z}^k - \mathbf{z}^*\right\|^2\right] - 2\mu \mathbb{E}\left[\left\|\mathbf{z}^{k+1} - \mathbf{z}^*\right\|^2\right] + \frac{\gamma}{\eta} \mathbb{E}\left[\left\|\mathbf{w}^k - \mathbf{z}^*\right\|^2\right] - \frac{\gamma}{\eta} \mathbb{E}\left[\left\|\mathbf{z}^k - \mathbf{z}^*\right\|^2\right]$$

$$- \frac{\gamma}{\eta} \mathbb{E}\left[\left\|\mathbf{w}^k - \mathbf{z}^{k+1}\right\|^2\right] - \frac{1/2 - \gamma}{\eta} \mathbb{E}\left[\left\|\mathbf{z}^{k+1} - \mathbf{z}^k\right\|^2\right]$$

$$+ \frac{4\overline{L}^2\eta}{b}\mathbb{E}\left[\left\|\mathbf{z}^k - \mathbf{w}^{k-1}\right\|^2\right] + \frac{4\alpha^2\overline{L}^2\eta}{b}\mathbb{E}\left[\left\|\mathbf{z}^k - \mathbf{z}^{k-1}\right\|^2\right]$$

$$- 2\mathbb{E}\left[\langle\mathbf{F}(\mathbf{z}^k) - \mathbf{F}(\mathbf{z}^{k+1}) - (\mathbf{y}^k - \mathbf{y}^{k+1}), \mathbf{z}^{k+1} - \mathbf{z}^*\rangle\right] + 2\mathbb{E}\left[\langle\mathbf{y}^{k+1} - \mathbf{y}^*, \mathbf{z}^{k+1} - \mathbf{z}^*\rangle\right]$$

$$- 2\alpha\mathbb{E}\left[\langle\mathbf{F}(\mathbf{z}^k) - \mathbf{F}(\mathbf{z}^{k-1}) - (\mathbf{y}^k - \mathbf{y}^{k-1}), \mathbf{z}^k - \mathbf{z}^*\rangle\right]$$

$$- 2\alpha\mathbb{E}\left[\langle\mathbf{F}(\mathbf{z}^k) - \mathbf{F}(\mathbf{z}^{k-1}) - (\mathbf{y}^k - \mathbf{y}^{k-1}), \mathbf{z}^{k+1} - \mathbf{z}^k\rangle\right].$$

**Part 3.** Now we are ready to consider (31). We sum up previous expression and (31). After small rearrangement, we have

$$\frac{1}{\eta}\mathbb{E}\left[\left\|\mathbf{z}^{k+1} - \mathbf{z}^*\right\|^2\right] + \frac{1}{\theta}\left\|\mathbf{y}^{k+1} - \mathbf{y}^*\right\|^2_{(\mathbf{W}^\dagger\otimes\mathbf{I}_d)} + 2\mathbb{E}\left[\langle\mathbf{F}(\mathbf{z}^k) - \mathbf{F}(\mathbf{z}^{k+1}) - (\mathbf{y}^k - \mathbf{y}^{k+1}), \mathbf{z}^{k+1} - \mathbf{z}^*\rangle\right]$$

$$\leq \frac{1}{\eta}\mathbb{E}\left[\left\|\mathbf{z}^k - \mathbf{z}^*\right\|^2\right] - (2\mu - 2\beta L^2)\mathbb{E}\left[\left\|\mathbf{z}^{k+1} - \mathbf{z}^*\right\|^2\right] + \left(1 - \beta\theta\chi^{-1}\right)\cdot\frac{1}{\theta}\left\|\mathbf{y}^k - \mathbf{y}^*\right\|^2_{(\mathbf{W}^\dagger\otimes\mathbf{I}_d)}$$

$$- 2\alpha\mathbb{E}\left[\langle\mathbf{F}(\mathbf{z}^k) - \mathbf{F}(\mathbf{z}^{k-1}) - (\mathbf{y}^k - \mathbf{y}^{k-1}), \mathbf{z}^k - \mathbf{z}^*\rangle\right] + \frac{\gamma}{\eta}\mathbb{E}\left[\left\|\mathbf{w}^k - \mathbf{z}^*\right\|^2\right] - \frac{\gamma}{\eta}\mathbb{E}\left[\left\|\mathbf{z}^k - \mathbf{z}^*\right\|^2\right]$$

$$- \left(\frac{\gamma}{\eta} - \frac{2\beta\overline{L}^2}{b}\right)\mathbb{E}\left[\left\|\mathbf{w}^k - \mathbf{z}^{k+1}\right\|^2\right] - \frac{1/2 - \gamma}{\eta}\mathbb{E}\left[\left\|\mathbf{z}^{k+1} - \mathbf{z}^k\right\|^2\right] + \frac{4\overline{L}^2\eta}{b}\mathbb{E}\left[\left\|\mathbf{z}^k - \mathbf{w}^{k-1}\right\|^2\right]$$

$$+ \frac{4\alpha^2\overline{L}^2\eta}{b}\mathbb{E}\left[\left\|\mathbf{z}^k - \mathbf{z}^{k-1}\right\|^2\right] - 2\alpha\mathbb{E}\left[\langle\mathbf{F}(\mathbf{z}^k) - \mathbf{F}(\mathbf{z}^{k-1}), \mathbf{z}^{k+1} - \mathbf{z}^k\rangle\right]$$

$$+ 2\alpha\mathbb{E}\left[\langle\mathbf{y}^k - \mathbf{y}^{k-1}, \mathbf{z}^{k+1} - \mathbf{z}^k\rangle\right] - \left(\frac{1}{\theta} - \beta\right)\mathbb{E}\left[\left\|\mathbf{y}^{k+1} - \mathbf{y}^k\right\|^2\right].$$

With $\theta \leq \frac{1}{2\beta}$ (or $\beta \leq \frac{1}{2\theta}$) and Young's inequality we get

$$\frac{1}{\eta}\mathbb{E}\left[\left\|\mathbf{z}^{k+1} - \mathbf{z}^*\right\|^2\right] + \frac{1}{\theta}\left\|\mathbf{y}^{k+1} - \mathbf{y}^*\right\|^2_{(\mathbf{W}^\dagger\otimes\mathbf{I}_d)} + 2\mathbb{E}\left[\langle\mathbf{F}(\mathbf{z}^k) - \mathbf{F}(\mathbf{z}^{k+1}) - (\mathbf{y}^k - \mathbf{y}^{k+1}), \mathbf{z}^{k+1} - \mathbf{z}^*\rangle\right]$$

$$\leq \frac{1}{\eta}\mathbb{E}\left[\left\|\mathbf{z}^k - \mathbf{z}^*\right\|^2\right] - (2\mu - 2\beta L^2)\mathbb{E}\left[\left\|\mathbf{z}^{k+1} - \mathbf{z}^*\right\|^2\right] + \left(1 - \beta\theta\chi^{-1}\right)\cdot\frac{1}{\theta}\left\|\mathbf{y}^k - \mathbf{y}^*\right\|^2_{(\mathbf{W}^\dagger\otimes\mathbf{I}_d)}$$

$$- 2\alpha\mathbb{E}\left[\langle\mathbf{F}(\mathbf{z}^k) - \mathbf{F}(\mathbf{z}^{k-1}) - (\mathbf{y}^k - \mathbf{y}^{k-1}), \mathbf{z}^k - \mathbf{z}^*\rangle\right] + \frac{\gamma}{\eta}\mathbb{E}\left[\left\|\mathbf{w}^k - \mathbf{z}^*\right\|^2\right] - \frac{\gamma}{\eta}\mathbb{E}\left[\left\|\mathbf{z}^k - \mathbf{z}^*\right\|^2\right]$$

$$- \left(\frac{\gamma}{\eta} - \frac{2\beta\overline{L}^2}{b}\right)\mathbb{E}\left[\left\|\mathbf{w}^k - \mathbf{z}^{k+1}\right\|^2\right] - \frac{1/2 - \gamma}{\eta}\mathbb{E}\left[\left\|\mathbf{z}^{k+1} - \mathbf{z}^k\right\|^2\right] + \frac{4\overline{L}^2\eta}{b}\mathbb{E}\left[\left\|\mathbf{z}^k - \mathbf{w}^{k-1}\right\|^2\right]$$

$$+ \frac{4\alpha^2\overline{L}^2\eta}{b}\mathbb{E}\left[\left\|\mathbf{z}^k - \mathbf{z}^{k-1}\right\|^2\right] + \frac{1}{8\eta}\mathbb{E}\left[\left\|\mathbf{z}^{k+1} - \mathbf{z}^k\right\|^2\right] + 8\eta\alpha^2\mathbb{E}\left[\left\|\mathbf{F}(\mathbf{z}^k) - \mathbf{F}(\mathbf{z}^{k-1})\right\|^2\right]$$

$$+ 2\theta\alpha\mathbb{E}\left[\left\|\mathbf{z}^{k+1} - \mathbf{z}^k\right\|^2\right] + \frac{\alpha}{2\theta}\mathbb{E}\left[\left\|\mathbf{y}^k - \mathbf{y}^{k-1}\right\|^2\right] - \frac{1}{2\theta}\mathbb{E}\left[\left\|\mathbf{y}^{k+1} - \mathbf{y}^k\right\|^2\right].$$

By choosing $\theta \leq \frac{1}{16\eta}$ and $\alpha \leq 1$

$$\frac{1}{\eta}\mathbb{E}\left[\left\|\mathbf{z}^{k+1} - \mathbf{z}^*\right\|^2\right] + \frac{1}{\theta}\left\|\mathbf{y}^{k+1} - \mathbf{y}^*\right\|^2_{(\mathbf{W}^\dagger\otimes\mathbf{I}_d)} + 2\mathbb{E}\left[\langle\mathbf{F}(\mathbf{z}^k) - \mathbf{F}(\mathbf{z}^{k+1}) - (\mathbf{y}^k - \mathbf{y}^{k+1}), \mathbf{z}^{k+1} - \mathbf{z}^*\rangle\right]$$

$$\leq \frac{1}{\eta}\mathbb{E}\left[\left\|\mathbf{z}^k - \mathbf{z}^*\right\|^2\right] - (2\mu - 2\beta L^2)\mathbb{E}\left[\left\|\mathbf{z}^{k+1} - \mathbf{z}^*\right\|^2\right] + \left(1 - \beta\theta\chi^{-1}\right)\cdot\frac{1}{\theta}\left\|\mathbf{y}^k - \mathbf{y}^*\right\|^2_{(\mathbf{W}^\dagger\otimes\mathbf{I}_d)}$$

$$- 2\alpha\mathbb{E}\left[\langle\mathbf{F}(\mathbf{z}^k) - \mathbf{F}(\mathbf{z}^{k-1}) - (\mathbf{y}^k - \mathbf{y}^{k-1}), \mathbf{z}^k - \mathbf{z}^*\rangle\right] + \frac{\gamma}{\eta}\mathbb{E}\left[\left\|\mathbf{w}^k - \mathbf{z}^*\right\|^2\right] - \frac{\gamma}{\eta}\mathbb{E}\left[\left\|\mathbf{z}^k - \mathbf{z}^*\right\|^2\right]$$

$$- \left(\frac{\gamma}{\eta} - \frac{2\beta\overline{L}^2}{b}\right)\mathbb{E}\left[\left\|\mathbf{w}^k - \mathbf{z}^{k+1}\right\|^2\right] - \frac{1/4 - \gamma}{\eta}\mathbb{E}\left[\left\|\mathbf{z}^{k+1} - \mathbf{z}^k\right\|^2\right] + \frac{4\overline{L}^2\eta}{b}\mathbb{E}\left[\left\|\mathbf{z}^k - \mathbf{w}^{k-1}\right\|^2\right]$$

$$+ \frac{4\alpha^2\overline{L}^2\eta}{b}\mathbb{E}\left[\left\|\mathbf{z}^k - \mathbf{z}^{k-1}\right\|^2\right] + 8\eta\alpha^2\mathbb{E}\left[\left\|\mathbf{F}(\mathbf{z}^k) - \mathbf{F}(\mathbf{z}^{k-1})\right\|^2\right] + \frac{\alpha}{2\theta}\mathbb{E}\left[\left\|\mathbf{y}^k - \mathbf{y}^{k-1}\right\|^2\right]$$

$$- \frac{1}{2\theta}\mathbb{E}\left[\left\|\mathbf{y}^{k+1} - \mathbf{y}^k\right\|^2\right].$$

With $L$-Lipshitzness of $\mathbf{F}$ (Assumption 2.1) and $\gamma \leq \frac{1}{8}$, we have

$$\frac{1}{\eta}\mathbb{E}\left[\left\|\mathbf{z}^{k+1} - \mathbf{z}^*\right\|^2\right] + \frac{1}{\theta}\left\|\mathbf{y}^{k+1} - \mathbf{y}^*\right\|^2_{(\mathbf{W}^\dagger\otimes\mathbf{I}_d)} + 2\mathbb{E}\left[\langle\mathbf{F}(\mathbf{z}^k) - \mathbf{F}(\mathbf{z}^{k+1}) - (\mathbf{y}^k - \mathbf{y}^{k+1}), \mathbf{z}^{k+1} - \mathbf{z}^*\rangle\right]$$

$$\leq \frac{1}{\eta}\mathbb{E}\left[\left\|\mathbf{z}^k - \mathbf{z}^*\right\|^2\right] - (2\mu - 2\beta L^2)\mathbb{E}\left[\left\|\mathbf{z}^{k+1} - \mathbf{z}^*\right\|^2\right] + \left(1 - \beta\theta\chi^{-1}\right)\cdot\frac{1}{\theta}\left\|\mathbf{y}^k - \mathbf{y}^*\right\|^2_{(\mathbf{W}^\dagger\otimes\mathbf{I}_d)}$$

$$- 2\alpha\mathbb{E}\left[\langle\mathbf{F}(\mathbf{z}^k) - \mathbf{F}(\mathbf{z}^{k-1}) - (\mathbf{y}^k - \mathbf{y}^{k-1}), \mathbf{z}^k - \mathbf{z}^*\rangle\right] + \frac{\gamma}{\eta}\mathbb{E}\left[\left\|\mathbf{w}^k - \mathbf{z}^*\right\|^2\right] - \frac{\gamma}{\eta}\mathbb{E}\left[\left\|\mathbf{z}^k - \mathbf{z}^*\right\|^2\right]$$

$$- \left(\frac{\gamma}{\eta} - \frac{2\beta\overline{L}^2}{b}\right)\mathbb{E}\left[\left\|\mathbf{w}^k - \mathbf{z}^{k+1}\right\|^2\right] - \frac{1}{8\eta}\mathbb{E}\left[\left\|\mathbf{z}^{k+1} - \mathbf{z}^k\right\|^2\right] + \frac{4\overline{L}^2\eta}{b}\mathbb{E}\left[\left\|\mathbf{z}^k - \mathbf{w}^{k-1}\right\|^2\right]$$

$$+ \left(\frac{32\alpha^2\overline{L}^2\eta^2}{b} + 64\eta^2\alpha^2 L^2\right)\cdot\frac{1}{8\eta}\mathbb{E}\left[\left\|\mathbf{z}^k - \mathbf{z}^{k-1}\right\|^2\right] + \frac{\alpha}{2\theta}\mathbb{E}\left[\left\|\mathbf{y}^k - \mathbf{y}^{k-1}\right\|^2\right]$$

$$- \frac{1}{2\theta}\mathbb{E}\left[\left\|\mathbf{y}^{k+1} - \mathbf{y}^k\right\|^2\right].$$

Small rearrangement gives

$$\frac{1}{\eta}\mathbb{E}\left[\left\|\mathbf{z}^{k+1} - \mathbf{z}^*\right\|^2\right] + \frac{1}{\theta}\left\|\mathbf{y}^{k+1} - \mathbf{y}^*\right\|^2_{(\mathbf{W}^\dagger\otimes\mathbf{I}_d)} + 2\mathbb{E}\left[\langle\mathbf{F}(\mathbf{z}^k) - \mathbf{F}(\mathbf{z}^{k+1}) - (\mathbf{y}^k - \mathbf{y}^{k+1}), \mathbf{z}^{k+1} - \mathbf{z}^*\rangle\right]$$

$$+ \frac{1}{2\theta}\mathbb{E}\left[\left\|\mathbf{y}^{k+1} - \mathbf{y}^k\right\|^2\right] + \frac{1}{8\eta}\mathbb{E}\left[\left\|\mathbf{z}^{k+1} - \mathbf{z}^k\right\|^2\right]$$

$$\leq \frac{1}{\eta}\mathbb{E}\left[\left\|\mathbf{z}^k - \mathbf{z}^*\right\|^2\right] - (2\mu - 2\beta L^2)\mathbb{E}\left[\left\|\mathbf{z}^{k+1} - \mathbf{z}^*\right\|^2\right] + \left(1 - \beta\theta\chi^{-1}\right)\cdot\frac{1}{\theta}\left\|\mathbf{y}^k - \mathbf{y}^*\right\|^2_{(\mathbf{W}^\dagger\otimes\mathbf{I}_d)}$$

$$- 2\alpha\mathbb{E}\left[\langle\mathbf{F}(\mathbf{z}^k) - \mathbf{F}(\mathbf{z}^{k-1}) - (\mathbf{y}^k - \mathbf{y}^{k-1}), \mathbf{z}^k - \mathbf{z}^*\rangle\right] + \frac{\gamma}{\eta}\mathbb{E}\left[\left\|\mathbf{w}^k - \mathbf{z}^*\right\|^2\right] - \frac{\gamma}{\eta}\mathbb{E}\left[\left\|\mathbf{z}^k - \mathbf{z}^*\right\|^2\right]$$

$$- \left(\frac{\gamma}{\eta} - \frac{2\beta\overline{L}^2}{b}\right)\mathbb{E}\left[\left\|\mathbf{w}^k - \mathbf{z}^{k+1}\right\|^2\right] + \frac{4\overline{L}^2\eta}{b}\mathbb{E}\left[\left\|\mathbf{z}^k - \mathbf{w}^{k-1}\right\|^2\right]$$

$$+ \left(\frac{32\alpha^2\overline{L}^2\eta^2}{b} + 64\eta^2\alpha^2 L^2\right)\cdot\frac{1}{8\eta}\mathbb{E}\left[\left\|\mathbf{z}^k - \mathbf{z}^{k-1}\right\|^2\right] + \frac{\alpha}{2\theta}\mathbb{E}\left[\left\|\mathbf{y}^k - \mathbf{y}^{k-1}\right\|^2\right].$$

With our choice of $\eta \leq \min\left\{\frac{\sqrt{\alpha\gamma b}}{\sqrt{8}\cdot\overline{L}}, \frac{1}{16L}\right\}$ and $\alpha < 1$, we get

$$\frac{1}{\eta}\mathbb{E}\left[\left\|\mathbf{z}^{k+1} - \mathbf{z}^*\right\|^2\right] + \frac{1}{\theta}\left\|\mathbf{y}^{k+1} - \mathbf{y}^*\right\|^2_{(\mathbf{W}^\dagger\otimes\mathbf{I}_d)} + 2\mathbb{E}\left[\langle\mathbf{F}(\mathbf{z}^k) - \mathbf{F}(\mathbf{z}^{k+1}) - (\mathbf{y}^k - \mathbf{y}^{k+1}), \mathbf{z}^{k+1} - \mathbf{z}^*\rangle\right]$$

$$+ \frac{1}{2\theta}\left\|\mathbf{y}^{k+1} - \mathbf{y}^k\right\|^2 + \frac{1}{8\eta}\mathbb{E}\left[\left\|\mathbf{z}^{k+1} - \mathbf{z}^k\right\|^2\right]$$

$$\leq \frac{1}{\eta}\mathbb{E}\left[\left\|\mathbf{z}^k - \mathbf{z}^*\right\|^2\right] - (2\mu - 2\beta L^2)\mathbb{E}\left[\left\|\mathbf{z}^{k+1} - \mathbf{z}^*\right\|^2\right] + \left(1 - \beta\theta\chi^{-1}\right)\cdot\frac{1}{\theta}\left\|\mathbf{y}^k - \mathbf{y}^*\right\|^2_{(\mathbf{W}^\dagger\otimes\mathbf{I}_d)}$$

$$- \alpha\cdot 2\mathbb{E}\left[\langle\mathbf{F}(\mathbf{z}^k) - \mathbf{F}(\mathbf{z}^{k-1}) - (\mathbf{y}^k - \mathbf{y}^{k-1}), \mathbf{z}^k - \mathbf{z}^*\rangle\right] + \alpha\cdot\frac{1}{2\theta}\mathbb{E}\left[\left\|\mathbf{y}^k - \mathbf{y}^{k-1}\right\|^2\right]$$

$$+ \alpha\cdot\frac{1}{8\eta}\mathbb{E}\left[\left\|\mathbf{z}^k - \mathbf{z}^{k-1}\right\|^2\right]$$

$$+ \frac{\gamma}{\eta}\mathbb{E}\left[\left\|\mathbf{w}^k - \mathbf{z}^*\right\|^2\right] - \frac{\gamma}{\eta}\mathbb{E}\left[\left\|\mathbf{z}^k - \mathbf{z}^*\right\|^2\right]$$

$$- \left(\frac{\gamma}{\eta} - \frac{2\beta\overline{L}^2}{b}\right)\mathbb{E}\left[\left\|\mathbf{w}^k - \mathbf{z}^{k+1}\right\|^2\right] + \frac{4\overline{L}^2\eta}{b}\mathbb{E}\left[\left\|\mathbf{z}^k - \mathbf{w}^{k-1}\right\|^2\right].$$

Now, we add $\frac{\mu}{2}\mathbb{E}\left[\left\|\mathbf{z}^{k+1} - \mathbf{z}^*\right\|^2\right] + \frac{\gamma + \frac{1}{2}\eta\mu}{p\eta}\mathbb{E}\left[\left\|\mathbf{w}^{k+1} - \mathbf{z}^*\right\|^2\right]$ to both sides and use update for $w^{k+1}$ of Algorithm [1]

$$\frac{\gamma + \frac{1}{2}\eta\mu}{p\eta}\mathbb{E}\left[\mathbb{E}_{w^{k+1}}\left\|\mathbf{w}^{k+1} - \mathbf{z}^*\right\|^2\right] = \frac{\gamma + \frac{1}{2}\eta\mu}{\eta}\mathbb{E}\left[\left\|\mathbf{z}^k - \mathbf{z}^*\right\|^2\right] + \frac{(\gamma + \frac{1}{2}\eta\mu)(1 - p)}{\eta p}\mathbb{E}\left[\left\|\mathbf{w}^k - \mathbf{z}^*\right\|^2\right],$$

and get

$$\left(\frac{1}{\eta} + \frac{\mu}{2}\right)\mathbb{E}\left[\left\|\mathbf{z}^{k+1} - \mathbf{z}^*\right\|^2\right] + \frac{1}{\theta}\left\|\mathbf{y}^{k+1} - \mathbf{y}^*\right\|^2_{(\mathbf{W}^\dagger\otimes\mathbf{I}_d)}$$

$$+ 2\mathbb{E}\left[\langle\mathbf{F}(\mathbf{z}^k) - \mathbf{F}(\mathbf{z}^{k+1}) - (\mathbf{y}^k - \mathbf{y}^{k+1}), \mathbf{z}^{k+1} - \mathbf{z}^*\rangle\right] + \frac{\gamma + \frac{1}{2}\eta\mu}{p\eta}\mathbb{E}\left[\left\|\mathbf{w}^{k+1} - \mathbf{z}^*\right\|^2\right]$$

$$+ \frac{1}{2\theta}\left\|\mathbf{y}^{k+1}-\mathbf{y}^k\right\|^2 + \frac{1}{8\eta}\mathbb{E}\left[\left\|\mathbf{z}^{k+1}-\mathbf{z}^k\right\|^2\right] + \left(\frac{\gamma}{\eta}-\frac{2\beta\overline{L}^2}{b}\right)\mathbb{E}\left[\left\|\mathbf{w}^k-\mathbf{z}^{k+1}\right\|^2\right]$$

$$\leq \left(\frac{1}{\eta}+\frac{\mu}{2}\right)\mathbb{E}\left[\left\|\mathbf{z}^k-\mathbf{z}^*\right\|^2\right] - \left(\frac{3}{2}\mu-2\beta L^2\right)\mathbb{E}\left[\left\|\mathbf{z}^{k+1}-\mathbf{z}^*\right\|^2\right]$$

$$+ \left(1-\beta\theta\chi^{-1}\right)\cdot\frac{1}{\theta}\left\|\mathbf{y}^k-\mathbf{y}^*\right\|^2_{(\mathbf{W}^\dagger\otimes\mathbf{I}_d)} - \alpha\cdot 2\mathbb{E}\left[\langle\mathbf{F}(\mathbf{z}^k)-\mathbf{F}(\mathbf{z}^{k-1})-(\mathbf{y}^k-\mathbf{y}^{k-1}),\mathbf{z}^k-\mathbf{z}^*\rangle\right]$$

$$+ \alpha\cdot\frac{1}{2\theta}\mathbb{E}\left[\left\|\mathbf{y}^k-\mathbf{y}^{k-1}\right\|^2\right] + \alpha\cdot\frac{1}{8\eta}\mathbb{E}\left[\left\|\mathbf{z}^k-\mathbf{z}^{k-1}\right\|^2\right]$$

$$+ \left(1-p+\frac{p\gamma}{\gamma+\frac{1}{2}\eta\mu}\right)\frac{(\gamma+\frac{1}{2}\eta\mu)}{\eta p}\mathbb{E}\left[\left\|\mathbf{w}^k-\mathbf{z}^*\right\|^2\right] + \frac{4\overline{L}^2\eta}{b}\mathbb{E}\left[\left\|\mathbf{z}^k-\mathbf{w}^{k-1}\right\|^2\right].$$

Note that $\eta \leq \min\left\{\frac{\sqrt{\alpha\gamma b}}{\sqrt{8}\cdot\overline{L}},\frac{1}{16L}\right\}$, $\beta \leq \min\left\{\frac{\mu}{4L^2};\frac{b\gamma}{4\eta\overline{L}^2}\right\}$, then we get that

$$\left(\frac{\gamma}{\eta}-\frac{2\beta\overline{L}^2}{b}\right)\geq\frac{\gamma}{2\eta};\quad -\left(\frac{3}{2}\mu-2\beta L^2\right)\leq-\mu;\quad \frac{4\overline{L}^2\eta}{b}\leq\alpha\cdot\frac{\gamma}{2\eta}.$$

Hence, it holds

$$\left(\frac{1}{\eta}+\frac{3\mu}{2}\right)\mathbb{E}\left[\left\|\mathbf{z}^{k+1}-\mathbf{z}^*\right\|^2\right] + \frac{1}{\theta}\left\|\mathbf{y}^{k+1}-\mathbf{y}^*\right\|^2_{(\mathbf{W}^\dagger\otimes\mathbf{I}_d)}$$

$$+ 2\mathbb{E}\left[\langle\mathbf{F}(\mathbf{z}^k)-\mathbf{F}(\mathbf{z}^{k+1})-(\mathbf{y}^k-\mathbf{y}^{k+1}),\mathbf{z}^{k+1}-\mathbf{z}^*\rangle\right] + \frac{\gamma+\frac{1}{2}\eta\mu}{p\eta}\mathbb{E}\left[\left\|\mathbf{w}^{k+1}-\mathbf{z}^*\right\|^2\right]$$

$$+ \frac{1}{2\theta}\left\|\mathbf{y}^{k+1}-\mathbf{y}^k\right\|^2 + \frac{1}{8\eta}\mathbb{E}\left[\left\|\mathbf{z}^{k+1}-\mathbf{z}^k\right\|^2\right] + \frac{\gamma}{2\eta}\mathbb{E}\left[\left\|\mathbf{w}^k-\mathbf{z}^{k+1}\right\|^2\right]$$

$$\leq \left(\frac{1}{\eta}+\frac{\mu}{2}\right)\mathbb{E}\left[\left\|\mathbf{z}^k-\mathbf{z}^*\right\|^2\right] + \left(1-\beta\theta\chi^{-1}\right)\cdot\frac{1}{\theta}\left\|\mathbf{y}^k-\mathbf{y}^*\right\|^2_{(\mathbf{W}^\dagger\otimes\mathbf{I}_d)}$$

$$- \alpha\cdot 2\mathbb{E}\left[\langle\mathbf{F}(\mathbf{z}^k)-\mathbf{F}(\mathbf{z}^{k-1})-(\mathbf{y}^k-\mathbf{y}^{k-1}),\mathbf{z}^k-\mathbf{z}^*\rangle\right]$$

$$+ \alpha\cdot\frac{1}{2\theta}\mathbb{E}\left[\left\|\mathbf{y}^k-\mathbf{y}^{k-1}\right\|^2\right] + \alpha\cdot\frac{1}{8\eta}\mathbb{E}\left[\left\|\mathbf{z}^k-\mathbf{z}^{k-1}\right\|^2\right]$$

$$+ \left(1-\frac{p\eta\mu}{2\gamma+\eta\mu}\right)\cdot\frac{\gamma+\frac{1}{2}\eta\mu}{p\eta}\mathbb{E}\left[\left\|\mathbf{w}^k-\mathbf{z}^*\right\|^2\right] + \alpha\cdot\frac{\gamma}{2\eta}\mathbb{E}\left[\left\|\mathbf{x}^k-\mathbf{w}^{k-1}\right\|^2\right].$$

With our $\eta$ we have $\eta\mu \leq 1$ and then

$$\left(\frac{1}{\eta}+\frac{\mu}{2}\right)\leq\left(\frac{1}{\eta}+\frac{3\mu}{2}\right)\left(1-\frac{\mu\eta}{4}\right).$$

It gives

$$\left(\frac{1}{\eta}+\frac{3\mu}{2}\right)\mathbb{E}\left[\left\|\mathbf{z}^{k+1}-\mathbf{z}^*\right\|^2\right] + \frac{1}{\theta}\left\|\mathbf{y}^{k+1}-\mathbf{y}^*\right\|^2_{(\mathbf{W}^\dagger\otimes\mathbf{I}_d)}$$

$$+ 2\mathbb{E}\left[\langle\mathbf{F}(\mathbf{z}^k)-\mathbf{F}(\mathbf{z}^{k+1})-(\mathbf{y}^k-\mathbf{y}^{k+1}),\mathbf{z}^{k+1}-\mathbf{z}^*\rangle\right] + \frac{\gamma+\frac{1}{2}\eta\mu}{p\eta}\mathbb{E}\left[\left\|\mathbf{w}^{k+1}-\mathbf{z}^*\right\|^2\right]$$

$$+ \frac{1}{2\theta}\left\|\mathbf{y}^{k+1}-\mathbf{y}^k\right\|^2 + \frac{1}{8\eta}\mathbb{E}\left[\left\|\mathbf{z}^{k+1}-\mathbf{z}^k\right\|^2\right] + \frac{\gamma}{2\eta}\mathbb{E}\left[\left\|\mathbf{w}^k-\mathbf{z}^{k+1}\right\|^2\right]$$

$$\leq \left(1-\frac{\mu\eta}{4}\right)\cdot\left(\frac{1}{\eta}+\frac{3\mu}{2}\right)\mathbb{E}\left[\left\|\mathbf{z}^k-\mathbf{z}^*\right\|^2\right] + \left(1-\beta\theta\chi^{-1}\right)\cdot\frac{1}{\theta}\left\|\mathbf{y}^k-\mathbf{y}^*\right\|^2_{(\mathbf{W}^\dagger\otimes\mathbf{I}_d)}$$

$$- \alpha\cdot 2\mathbb{E}\left[\langle\mathbf{F}(\mathbf{z}^k)-\mathbf{F}(\mathbf{z}^{k-1})-(\mathbf{y}^k-\mathbf{y}^{k-1}),\mathbf{z}^k-\mathbf{z}^*\rangle\right]$$

$$+ \alpha\cdot\frac{1}{2\theta}\mathbb{E}\left[\left\|\mathbf{y}^k-\mathbf{y}^{k-1}\right\|^2\right] + \alpha\cdot\frac{1}{8\eta}\mathbb{E}\left[\left\|\mathbf{z}^k-\mathbf{z}^{k-1}\right\|^2\right]$$

$$+ \left(1-\frac{p\eta\mu}{2\gamma+\eta\mu}\right)\cdot\frac{\gamma+\frac{1}{2}\eta\mu}{p\eta}\mathbb{E}\left[\left\|\mathbf{w}^k-\mathbf{z}^*\right\|^2\right] + \alpha\cdot\frac{\gamma}{2\eta}\mathbb{E}\left[\left\|\mathbf{x}^k-\mathbf{w}^{k-1}\right\|^2\right].$$

Definition (29) of the Lyapunov function and the choice $\alpha = \max\left[\left(1-\frac{\mu\eta}{4}\right);\left(1-\beta\theta\chi^{-1}\right);\left(1-\frac{p\eta\mu}{2\gamma+\eta\mu}\right)\right]$ move us to

$$\mathbb{E}\Psi^{k+1} \leq \max\left[\left(1-\frac{\mu\eta}{4}\right);\left(1-\beta\theta\chi^{-1}\right);\left(1-\frac{p\eta\mu}{2\gamma+\eta\mu}\right)\right]\mathbb{E}\Psi^k.$$

It remains to show, that

$$\Psi^k \geq \frac{1}{2\eta} \left\| \mathbf{z}^k - \mathbf{z}^* \right\|^2.$$

$$\Psi^k \geq \frac{1}{\eta} \left\| \mathbf{z}^k - \mathbf{z}^* \right\|^2 + \frac{1}{8\eta} \left\| \mathbf{z}^k - \mathbf{z}^{k-1} \right\|^2 + \frac{1}{2\theta} \left\| \mathbf{y}^k - \mathbf{y}^{k-1} \right\|^2$$
$$+ 2\langle \mathbf{F}(\mathbf{z}^k) - \mathbf{F}(\mathbf{z}^{k-1}) - (\mathbf{y}^k - \mathbf{y}^{k-1}), \mathbf{z}^* - \mathbf{z}^k \rangle$$
$$\geq \frac{1}{\eta} \left\| \mathbf{z}^k - \mathbf{z}^* \right\|^2 + \frac{1}{8\eta} \left\| \mathbf{z}^k - \mathbf{z}^{k-1} \right\|^2 + \frac{1}{2\theta} \left\| \mathbf{y}^k - \mathbf{y}^{k-1} \right\|^2$$
$$- \frac{1}{2\theta} \left\| \mathbf{y}^k - \mathbf{y}^{k-1} \right\|^2 - 2\theta \left\| \mathbf{z}^k - \mathbf{z}^* \right\|^2 - \frac{1}{8\eta L^2} \left\| \mathbf{F}(\mathbf{z}^k) - \mathbf{F}(\mathbf{z}^{k-1}) \right\|^2 - 8\eta L^2 \left\| \mathbf{z}^k - \mathbf{z}^* \right\|^2.$$

Using $L$-Lipschitzness of $\mathbf{F}$ we get

$$\Psi^k \geq \frac{1}{2\eta} \left( 2 - 8\eta^2 L^2 - 4\theta\eta \right) \left\| \mathbf{z}^k - \mathbf{z}^* \right\|^2.$$

With $\eta \leq \frac{1}{16L}$ and $\theta \leq \frac{1}{16\eta}$ we get

$$\Psi^k \geq \frac{1}{2\eta} \left\| \mathbf{z}^k - \mathbf{z}^* \right\|^2.$$

$\square$

**Theorem D.3** (Theorem 4.1). *Consider the problem (10) (or (1) + (5)) under Assumptions 2.1 and 2.2 over a fixed connected graph $\mathcal{G}$ with a gossip matrix $\mathbf{W}$. Let $\{\mathbf{z}^k\}$ be the sequence generated by Algorithm 1 with parameters*

$$\gamma = p \leq \frac{1}{8}, \quad \eta = \min \left\{ \frac{\sqrt{\gamma b}}{4\overline{L}}, \frac{1}{16L\sqrt{\chi}} \right\}, \quad \beta = \min \left\{ \frac{\mu}{4L^2}; \frac{b\gamma}{4\eta\overline{L}^2} \right\}, \quad \theta = \min \left\{ \frac{1}{2\beta}; \frac{1}{16\eta} \right\},$$

$$\alpha = \max \left[ \left( 1 - \frac{\mu\eta}{4} \right); \left( 1 - \beta\theta\chi^{-1} \right); \left( 1 - \frac{p\eta\mu}{2\gamma + \eta\mu} \right) \right].$$

*Then, given $\varepsilon > 0$, the number of iterations for $\mathbb{E}[\|\mathbf{z}^k - \mathbf{z}^*\|^2] \leq \varepsilon$ is*

$$O\left( \left[ \frac{1}{p} + \chi + \frac{1}{\sqrt{pb}} \frac{\overline{L}}{\mu} + \sqrt{\chi} \frac{L}{\mu} \right] \log \frac{1}{\varepsilon} \right).$$

*Proof.* It is easy to check that $\alpha \geq \frac{1}{2}$, then also one can verify that the choice of $\gamma$, $\eta$, $\beta$, $\theta$, $\alpha$ satisfies the conditions of Lemma D.2. We can get that the iteration complexity of Algorithm 1:

$$O\left( \left[ 1 + \frac{1}{\eta\mu} + \frac{1}{\beta\theta\chi^{-1}} + \frac{\gamma + \eta\mu}{p\eta\mu} \right] \log \frac{1}{\varepsilon} \right) = O\left( \left[ \frac{1}{p} + \frac{1}{\eta\mu} + \frac{1}{\beta\theta\chi^{-1}} \right] \log \frac{1}{\varepsilon} \right)$$

$$= O\left( \left[ \frac{1}{p} + \chi + \frac{1}{\eta\mu} + \frac{\chi\eta}{\beta} \right] \log \frac{1}{\varepsilon} \right)$$

$$= O\left( \left[ \frac{1}{p} + \chi + \frac{1}{\eta\mu} + \frac{\chi\eta^2\overline{L}^2}{bp} + \frac{\chi\eta L^2}{\mu} \right] \log \frac{1}{\varepsilon} \right)$$

$$= O\left( \left[ \frac{1}{p} + \chi + \frac{1}{\sqrt{pb}} \frac{\overline{L}}{\mu} + \sqrt{\chi} \frac{L}{\mu} \right] \log \frac{1}{\varepsilon} \right).$$

$\square$

# E Proof of Theorem 4.3

We start the proof from the following lemma on $\delta^k$ and $\delta^{k+1/2}$ from Algorithm 2.

**Lemma E.1.** *The following inequality holds:*

$$\mathbb{E}_k\left[\left\|\delta^k - \mathbb{E}_k\left[\delta^k\right]\right\|^2\right] \leq \frac{2\overline{L}^2}{b}\mathbb{E}\left[\left\|\mathbf{z}^k - \mathbf{w}^{k-1}\right\|^2 + \alpha^2\left\|\mathbf{z}^k - \mathbf{z}^{k-1}\right\|^2\right]. \tag{32}$$

$$\mathbb{E}_k\left[\left\|\delta^{k+1/2} - \mathbf{F}(\mathbf{z}^*)\right\|^2\right] \leq \frac{2\overline{L}^2}{b}\mathbb{E}\left[\left\|\mathbf{z}^{k+1} - \mathbf{w}^k\right\|^2\right] + 2L^2\mathbb{E}\left[\left\|\mathbf{z}^{k+1} - \mathbf{z}^*\right\|^2\right]. \tag{33}$$

*where $\mathbb{E}_k\left[\delta^k\right]$ is equal to*

$$\mathbb{E}_k\left[\delta^k\right] = F(\mathbf{z}^k) + \alpha(F(\mathbf{z}^k) - F(\mathbf{z}^{k-1})). \tag{34}$$

*Proof.* The proof is the same as the proof for Lemma D.1. $\qquad\square$

Before proving the main lemma of this section, let us introduce an auxiliary notation. Let $\hat{z}^k$ be defined for all $k = 0, 1, 2, \ldots$ as follows:

$$\hat{\mathbf{x}}^k = \mathbf{x}^k - \mathbf{P}m^k, \tag{35}$$

and the Lyapunov function $\Psi^k$ be denoted by

$$\begin{aligned}
\Psi^k &= \frac{1}{\eta_y}\left\|\mathbf{y}^k - \mathbf{y}^*\right\|^2 + \frac{1}{\eta_x}\left\|\hat{\mathbf{x}}^k - \mathbf{x}^*\right\|^2 + \frac{2}{\eta_x}\left\|m^k\right\|_{\mathbf{P}}^2 + \frac{1}{2\eta_y}\left\|\mathbf{y}^k - \mathbf{y}^{k-1}\right\|^2 \\
&\quad + \frac{\nu^{-1}}{\tau}\left\|\mathbf{y}_f^k + \mathbf{x}_f^k - \mathbf{y}^* - \mathbf{x}^*\right\|^2 - 2\langle\mathbf{F}(\mathbf{z}^k) - \mathbf{F}(\mathbf{z}^{k-1}) - (\mathbf{y}^k - \mathbf{y}^{k-1}), \mathbf{z}^k - \mathbf{z}^*\rangle \\
&\quad + \frac{1}{4\eta_z}\left\|\mathbf{z}^k - \mathbf{z}^{k-1}\right\|^2 + \left(1 + \frac{3\mu\eta_z}{2}\right)\cdot\frac{1}{\eta_z}\left\|\mathbf{z}^k - \mathbf{z}^*\right\|^2 + \frac{\omega}{2\eta_z}\mathbb{E}\left[\left\|\mathbf{z}^k - \mathbf{w}^{k-1}\right\|^2\right].
\end{aligned} \tag{36}$$

Here we also use the next notation

$$\mathbf{y}^* = \mathbf{PF}(\mathbf{z}^*) - \nu\mathbf{z}^*, \tag{37}$$

and:

$$\mathbf{x}^* = -\mathbf{PF}(\mathbf{z}^*). \tag{38}$$

**Lemma E.2.** *Consider the problem* (10) *(or* (1) + (5)*) under Assumptions* 2.1 *and* 2.2 *over a sequence of time-varying graphs $\mathcal{G}(k)$ with gossip matrices $\mathbf{W}(k)$. Let $\{\mathbf{z}^k\}$ be the sequence generated by Algorithm 2 with*

$$T \geq B, \quad \omega = p \leq \frac{1}{16}; \quad \theta = \frac{1}{2}; \quad \beta = 5\gamma; \quad \nu \leq \frac{\mu}{4}; \quad \tau \in (0; 1);$$

$$\eta_z \leq \min\left\{\frac{1}{8L}, \frac{1}{32\eta_y}, \frac{\sqrt{\alpha b\omega}}{8\overline{L}}\right\}; \quad \eta_y \leq \min\left\{\frac{1}{4\gamma}, \frac{\nu}{8\tau}\right\}; \quad \eta_x \leq \min\left\{\frac{1}{900\chi(T)\gamma}, \frac{\nu}{36\tau\chi^2(T)}\right\};$$

$$\gamma \leq \min\left\{\frac{\mu}{16L^2}; \frac{b\omega}{24\overline{L}^2\eta_z}\right\},$$

$$\alpha = \max\left[\left(1 - \frac{p\eta_z\nu}{p + \eta_z\nu}\right); 1 - \eta_y\gamma; 1 - \eta_x\gamma; 1 - \frac{\mu\eta_z}{8}; 1 - \tau; 1 - \frac{1}{4\chi(T)}\right].$$

*Let the choice of $T$ guarantees contraction property (Assumption* 2.4 *point 4) with $\chi(T)$. Then, after $k$ iterations we get*

$$\mathbb{E}\left[\frac{1}{4\eta_z}\left\|\mathbf{z}^k - \mathbf{z}^*\right\|^2\right]$$

$$\leq \max\left[\left(1 - \frac{p\eta_z\nu}{\omega + \eta_z\nu}\right); 1 - \eta_y\gamma; 1 - \eta_x\gamma, \left(1 - \frac{\mu\eta_z}{8}\right); 1 - \tau; \left(1 - \frac{1}{4\chi(T)}\right)\right]^k \cdot \Psi^0.$$

*Proof.* **Part 1.** We start from

$$\begin{aligned}
\frac{1}{\eta_z}\left\|\mathbf{z}^{k+1} - \mathbf{z}^*\right\|^2 &= \frac{1}{\eta_z}\left\|\mathbf{z}^k - \mathbf{z}^*\right\|^2 + \frac{2}{\eta_z}\langle\mathbf{z}^{k+1} - \mathbf{z}^k, \mathbf{z}^{k+1} - \mathbf{z}^*\rangle - \frac{1}{\eta_z}\left\|\mathbf{z}^{k+1} - \mathbf{z}^k\right\|^2 \\
&= \frac{1}{\eta_z}\left\|\mathbf{z}^k - \mathbf{z}^*\right\|^2 + \frac{2\omega}{\eta_z}\langle\mathbf{w}^k - \mathbf{z}^k, \mathbf{z}^{k+1} - \mathbf{z}^*\rangle \\
&\quad - 2\langle\Delta_z^k - (\mathbf{F}(\mathbf{z}^*) - \mathbf{y}^* - \nu\mathbf{z}^*), \mathbf{z}^{k+1} - \mathbf{z}^*\rangle - \frac{1}{\eta_z}\left\|\mathbf{z}^{k+1} - \mathbf{z}^k\right\|^2
\end{aligned}$$

$$-2\langle \tfrac{1}{\eta_z}(\mathbf{z}^k + \omega(\mathbf{w}^k - \mathbf{z}^k) - \eta_z\Delta_z^k - \mathbf{z}^{k+1}) - (-(\mathbf{F}(\mathbf{z}^*) - \mathbf{y}^* - \nu\mathbf{z}^*)), \mathbf{z}^{k+1} - \mathbf{z}^*\rangle.$$

Optimality condition for (1)+(5) it follows, that

$$-F(z^*) \in \partial g(z^*).$$

Let us define $\Delta^* \in (\mathbb{R}^d)^M$ as

$$\Delta^* = \tfrac{1}{M}[F(z^*), \dots F(z^*)]^\top.$$

It is to note that $-\Delta^* \in M\partial \mathbf{g}(\mathbf{z}^*)$. On the other hand with notation (37), we get

$$\Delta^* = (\mathbf{1}_M\mathbf{1}_M^\top \otimes \mathbf{I}_d)\mathbf{F}(\mathbf{z}^*) = M(\mathbf{F}(\mathbf{z}^*) - \mathbf{P}\mathbf{F}(\mathbf{z}^*)) = M(\mathbf{F}(\mathbf{z}^*) - \mathbf{y}^* - \nu\mathbf{z}^*)$$

It means that $-(\mathbf{F}(\mathbf{z}^*) - \mathbf{y}^* - \nu\mathbf{z}^*) \in \partial \mathbf{g}(\mathbf{z}^*)$. From update for $\mathbf{z}^{k+1}$ of Algorithm 2 it follows, that $\tfrac{1}{\eta_z}(\mathbf{z}^k + \omega(\mathbf{w}^k - \mathbf{z}^k) - \mathbf{z}^{k+1} - \eta_z\Delta_z^k) \in \partial \mathbf{g}(\mathbf{z}^{k+1})$. This together with $-(\mathbf{F}(\mathbf{z}^*) - \mathbf{y}^* - \nu\mathbf{z}^*) \in \partial \mathbf{g}(\mathbf{z}^*)$ and monotonicity of $\partial \mathbf{g}(\cdot)$ implies

$$\frac{1}{\eta_z}\mathbb{E}\left[\left\|\mathbf{z}^{k+1} - \mathbf{z}^*\right\|^2\right] \le \frac{1}{\eta_z}\mathbb{E}\left[\left\|\mathbf{z}^k - \mathbf{z}^*\right\|^2\right] + \frac{2\omega}{\eta_z}\mathbb{E}\left[\langle\mathbf{w}^k - \mathbf{z}^k, \mathbf{z}^{k+1} - \mathbf{z}^*\rangle\right]$$

$$- 2\mathbb{E}\left[\langle\Delta_z^k - (\mathbf{F}(\mathbf{z}^*) - \mathbf{y}^* - \nu\mathbf{z}^*), \mathbf{z}^{k+1} - \mathbf{z}^*\rangle\right] - \frac{1}{\eta_z}\mathbb{E}\left[\left\|\mathbf{z}^{k+1} - \mathbf{z}^k\right\|^2\right]$$

$$= \frac{1}{\eta_z}\mathbb{E}\left[\left\|\mathbf{z}^k - \mathbf{z}^*\right\|^2\right] + \frac{2\omega}{\eta_z}\mathbb{E}\left[\langle\mathbf{w}^k - \mathbf{z}^*, \mathbf{z}^{k+1} - \mathbf{z}^*\rangle\right]$$

$$- \frac{2\omega}{\eta_z}\mathbb{E}\left[\langle\mathbf{z}^k - \mathbf{z}^*, \mathbf{z}^{k+1} - \mathbf{z}^*\rangle\right] - 2\mathbb{E}\left[\langle\Delta_z^k - (\mathbf{F}(\mathbf{z}^*) - \mathbf{y}^* - \nu\mathbf{z}^*), \mathbf{z}^{k+1} - \mathbf{z}^*\rangle\right]$$

$$- \frac{1}{\eta_z}\mathbb{E}\left[\left\|\mathbf{z}^{k+1} - \mathbf{z}^k\right\|^2\right]$$

$$= \frac{1}{\eta_z}\mathbb{E}\left[\left\|\mathbf{z}^k - \mathbf{z}^*\right\|^2\right] + \frac{\omega}{\eta_z}\mathbb{E}\left[\left\|\mathbf{w}^k - \mathbf{z}^*\right\|^2 + \left\|\mathbf{z}^{k+1} - \mathbf{z}^*\right\|^2 - \left\|\mathbf{z}^{k+1} - \mathbf{w}^k\right\|^2\right]$$

$$- \frac{\omega}{\eta_z}\mathbb{E}\left[\left\|\mathbf{z}^{k+1} - \mathbf{z}^*\right\|^2 + \left\|\mathbf{z}^k - \mathbf{z}^*\right\|^2 - \left\|\mathbf{z}^{k+1} - \mathbf{z}^k\right\|^2\right]$$

$$- 2\mathbb{E}\left[\langle\Delta_z^k - (\mathbf{F}(\mathbf{z}^*) - \mathbf{y}^* - \nu\mathbf{z}^*), \mathbf{z}^{k+1} - \mathbf{z}^*\rangle\right] - \frac{1}{\eta_z}\mathbb{E}\left[\left\|\mathbf{z}^{k+1} - \mathbf{z}^k\right\|^2\right]$$

$$= \frac{1}{\eta_z}\mathbb{E}\left[\left\|\mathbf{z}^k - \mathbf{z}^*\right\|^2\right] + \frac{\omega}{\eta_z}\mathbb{E}\left[\left\|\mathbf{w}^k - \mathbf{z}^*\right\|^2\right] - \frac{\omega}{\eta_z}\mathbb{E}\left[\left\|\mathbf{z}^{k+1} - \mathbf{w}^k\right\|^2\right]$$

$$- \frac{\omega}{\eta_z}\mathbb{E}\left[\left\|\mathbf{z}^k - \mathbf{z}^*\right\|^2\right] - 2\mathbb{E}\left[\langle\Delta_z^k - (\mathbf{F}(\mathbf{z}^*) - \mathbf{y}^* - \nu\mathbf{z}^*), \mathbf{z}^{k+1} - \mathbf{z}^*\rangle\right]$$

$$- \frac{1-\omega}{\eta_z}\mathbb{E}\left[\left\|\mathbf{z}^{k+1} - \mathbf{z}^k\right\|^2\right].$$

In previous we also use the simple fact $\|a + b\|^2 = \|a\|^2 + 2\langle a; b\rangle + \|b\|^2$ twice. Definition $\Delta_z^k$ and small rearrangement gives

$$\frac{1}{\eta_z}\mathbb{E}\left[\left\|\mathbf{z}^{k+1} - \mathbf{z}^*\right\|^2\right] \le \frac{1}{\eta_z}\mathbb{E}\left[\left\|\mathbf{z}^k - \mathbf{z}^*\right\|^2\right] + \frac{\omega}{\eta_z}\mathbb{E}\left[\left\|\mathbf{w}^k - \mathbf{z}^*\right\|^2\right] - \frac{\omega}{\eta_z}\mathbb{E}\left[\left\|\mathbf{z}^{k+1} - \mathbf{w}^k\right\|^2\right]$$

$$- \frac{\omega}{\eta_z}\mathbb{E}\left[\left\|\mathbf{z}^k - \mathbf{z}^*\right\|^2\right] - \frac{1-\omega}{\eta_z}\mathbb{E}\left[\left\|\mathbf{z}^{k+1} - \mathbf{z}^k\right\|^2\right]$$

$$- 2\mathbb{E}\left[\langle\mathbb{E}\left[\delta^k\right] - \nu\mathbf{z}^k - y^k - \alpha(\mathbf{y}^k - \mathbf{y}^{k-1}) - (\mathbf{F}(\mathbf{z}^*) - \mathbf{y}^* - \nu\mathbf{z}^*), \mathbf{z}^{k+1} - \mathbf{z}^*\rangle\right]$$

$$+ 2\mathbb{E}\left[\langle\mathbb{E}\left[\delta^k\right] - \delta^k, \mathbf{z}^{k+1} - \mathbf{z}^k\rangle\right] + 2\mathbb{E}\left[\langle\mathbb{E}\left[\delta^k\right] - \delta^k, \mathbf{z}^k - \mathbf{z}^*\rangle\right].$$

Using the tower property of expectation we can obtain the following:

$$\mathbb{E}\left[\langle\mathbb{E}_k\left[\delta^k\right] - \delta^k, \mathbf{z}^k - \mathbf{z}^*\rangle\right] = \mathbb{E}\left[\mathbb{E}_k\left[\langle\mathbb{E}_k\left[\delta^k\right] - \delta^k, \mathbf{z}^k - \mathbf{z}^*\rangle\right]\right]$$

$$= \mathbb{E}\left[\langle\mathbb{E}_k\left[\mathbb{E}_k\left[\delta^k\right] - \delta^k\right], \mathbf{z}^k - \mathbf{z}^*\rangle\right]$$

$$= \mathbb{E}\left[\langle\mathbb{E}_k\left[\delta^k\right] - \mathbb{E}_k\left[\delta^k\right], \mathbf{z}^k - \mathbf{z}^*\rangle\right] = 0.$$

Then we have the following

$$\frac{1}{\eta_z}\mathbb{E}\left[\left\|\mathbf{z}^{k+1} - \mathbf{z}^*\right\|^2\right] \le \frac{1}{\eta_z}\mathbb{E}\left[\left\|\mathbf{z}^k - \mathbf{z}^*\right\|^2\right] + \frac{\omega}{\eta_z}\mathbb{E}\left[\left\|\mathbf{w}^k - \mathbf{z}^*\right\|^2\right] - \frac{\omega}{\eta_z}\mathbb{E}\left[\left\|\mathbf{z}^{k+1} - \mathbf{w}^k\right\|^2\right]$$

$$- \frac{\omega}{\eta_z} \mathbb{E}\left[\left\|\mathbf{z}^k - \mathbf{z}^*\right\|^2\right] - \frac{1-\omega}{\eta_z} \mathbb{E}\left[\left\|\mathbf{z}^{k+1} - \mathbf{z}^k\right\|^2\right]$$

$$- 2\mathbb{E}\left[\langle \mathbb{E}\left[\delta^k\right] - \nu \mathbf{z}^k - \mathbf{y}^k - \alpha(\mathbf{y}^k - \mathbf{y}^{k-1}) - (\mathbf{F}(\mathbf{z}^*) - \mathbf{y}^* - \nu \mathbf{z}^*), \mathbf{z}^{k+1} - \mathbf{z}^*\rangle\right]$$

$$+ 2\mathbb{E}\left[\langle \mathbb{E}\left[\delta^k\right] - \delta^k, \mathbf{z}^{k+1} - \mathbf{z}^k\rangle\right].$$

Using the Young's inequality we get

$$\frac{1}{\eta_z}\mathbb{E}\left[\left\|\mathbf{z}^{k+1} - \mathbf{z}^*\right\|^2\right] \leq \frac{1}{\eta_z}\mathbb{E}\left[\left\|\mathbf{z}^k - \mathbf{z}^*\right\|^2\right] + \frac{\omega}{\eta_z}\mathbb{E}\left[\left\|w^k - \mathbf{z}^*\right\|^2\right] - \frac{\omega}{\eta_z}\mathbb{E}\left[\left\|\mathbf{z}^{k+1} - \mathbf{w}^k\right\|^2\right]$$

$$- \frac{\omega}{\eta_z}\mathbb{E}\left[\left\|\mathbf{z}^k - \mathbf{z}^*\right\|^2\right] - \frac{1-\omega}{\eta_z}\mathbb{E}\left[\left\|\mathbf{z}^{k+1} - \mathbf{z}^k\right\|^2\right]$$

$$- 2\mathbb{E}\left[\langle \mathbb{E}\left[\delta^k\right] - \nu \mathbf{z}^k - \mathbf{y}^k - \alpha(\mathbf{y}^k - \mathbf{y}^{k-1}) - (\mathbf{F}(\mathbf{z}^*) - \mathbf{y}^* - \nu \mathbf{z}^*), \mathbf{z}^{k+1} - \mathbf{z}^*\rangle\right]$$

$$+ 2\eta_z\mathbb{E}\left[\left\|\mathbb{E}\left[\delta^k\right] - \delta^k\right\|^2\right] + \frac{1}{2\eta_z}\mathbb{E}\left[\left\|\mathbf{z}^{k+1} - \mathbf{z}^k\right\|^2\right]$$

$$= \frac{1}{\eta_z}\mathbb{E}\left[\left\|\mathbf{z}^k - \mathbf{z}^*\right\|^2\right] + \frac{\omega}{\eta_z}\mathbb{E}\left[\left\|\mathbf{w}^k - \mathbf{z}^*\right\|^2\right] - \frac{\omega}{\eta_z}\mathbb{E}\left[\left\|\mathbf{z}^{k+1} - \mathbf{w}^k\right\|^2\right]$$

$$- \frac{\omega}{\eta_z}\mathbb{E}\left[\left\|\mathbf{z}^k - \mathbf{z}^*\right\|^2\right] - \frac{1/2-\omega}{\eta_z}\mathbb{E}\left[\left\|\mathbf{z}^{k+1} - \mathbf{z}^k\right\|^2\right]$$

$$- 2\mathbb{E}\left[\langle \mathbb{E}\left[\delta^k\right] - \nu \mathbf{z}^k - \mathbf{y}^k - \alpha(\mathbf{y}^k - \mathbf{y}^{k-1}) - (\mathbf{F}(\mathbf{z}^*) - \mathbf{y}^* - \nu \mathbf{z}^*), \mathbf{z}^{k+1} - \mathbf{z}^*\rangle\right]$$

$$+ 2\eta_z\mathbb{E}\left[\left\|\mathbb{E}\left[\delta^k\right] - \delta^k\right\|^2\right].$$

With (32) and (34) we have

$$\frac{1}{\eta_z}\mathbb{E}\left[\left\|\mathbf{z}^{k+1} - \mathbf{z}^*\right\|^2\right] \leq \frac{1}{\eta_z}\mathbb{E}\left[\left\|\mathbf{z}^k - \mathbf{z}^*\right\|^2\right] + \frac{\omega}{\eta_z}\mathbb{E}\left[\left\|w^k - \mathbf{z}^*\right\|^2\right] - \frac{\omega}{\eta_z}\mathbb{E}\left[\left\|\mathbf{z}^{k+1} - \mathbf{w}^k\right\|^2\right]$$

$$- \frac{\omega}{\eta_z}\mathbb{E}\left[\left\|\mathbf{z}^k - \mathbf{z}^*\right\|^2\right] - \frac{1/2-\omega}{\eta_z}\mathbb{E}\left[\left\|\mathbf{z}^{k+1} - \mathbf{z}^k\right\|^2\right]$$

$$- 2\mathbb{E}\left[\langle \mathbf{F}(\mathbf{z}^k) + \alpha(\mathbf{F}(\mathbf{z}^k) - \mathbf{F}(\mathbf{z}^{k-1})) - \nu \mathbf{z}^k - \mathbf{y}^k, \mathbf{z}^{k+1} - \mathbf{z}^*\rangle\right]$$

$$- 2\mathbb{E}\left[\langle -\alpha(\mathbf{y}^k - \mathbf{y}^{k-1}) - (\mathbf{F}(\mathbf{z}^*) - \mathbf{y}^* - \nu \mathbf{z}^*), \mathbf{z}^{k+1} - \mathbf{z}^*\rangle\right]$$

$$+ \frac{4\eta_z \overline{L}^2}{b}\mathbb{E}\left[\left\|\mathbf{z}^k - \mathbf{w}^{k-1}\right\|^2\right] + \frac{4\eta_z \alpha^2 \overline{L}^2}{b}\mathbb{E}\left[\left\|\mathbf{z}^k - \mathbf{z}^{k-1}\right\|^2\right].$$

Small rearrangement gives

$$\frac{1}{\eta_z}\mathbb{E}\left[\left\|\mathbf{z}^{k+1} - \mathbf{z}^*\right\|^2\right] \leq \frac{1}{\eta_z}\mathbb{E}\left[\left\|\mathbf{z}^k - \mathbf{z}^*\right\|^2\right] + \frac{\omega}{\eta_z}\mathbb{E}\left[\left\|\mathbf{w}^k - \mathbf{z}^*\right\|^2\right] - \frac{\omega}{\eta_z}\mathbb{E}\left[\left\|\mathbf{z}^{k+1} - \mathbf{w}^k\right\|^2\right]$$

$$- \frac{\omega}{\eta_z}\mathbb{E}\left[\left\|\mathbf{z}^k - \mathbf{z}^*\right\|^2\right] - \frac{1/2-\omega}{\eta_z}\mathbb{E}\left[\left\|\mathbf{z}^{k+1} - \mathbf{z}^k\right\|^2\right]$$

$$- 2\mathbb{E}\left[\langle \mathbf{F}(\mathbf{z}^{k+1}) - \mathbf{F}(\mathbf{z}^*), \mathbf{z}^{k+1} - \mathbf{z}^*\rangle\right] + 2\nu\mathbb{E}\left[\langle \mathbf{z}^k - \mathbf{z}^*, \mathbf{z}^{k+1} - \mathbf{z}^*\rangle\right]$$

$$+ 2\mathbb{E}\left[\langle \mathbf{y}^{k+1} - \mathbf{y}^*, \mathbf{z}^{k+1} - \mathbf{z}^*\rangle\right]$$

$$+ 2\mathbb{E}\left[\langle \mathbf{F}(\mathbf{z}^{k+1}) - \mathbf{F}(\mathbf{z}^k) - (\mathbf{y}^{k+1} - \mathbf{y}^k), \mathbf{z}^{k+1} - \mathbf{z}^*\rangle\right]$$

$$- 2\alpha\mathbb{E}\left[\langle \mathbf{F}(\mathbf{z}^k) - \mathbf{F}(\mathbf{z}^{k-1}) - (\mathbf{y}^k - \mathbf{y}^{k-1}), \mathbf{z}^k - \mathbf{z}^*\rangle\right]$$

$$- 2\alpha\mathbb{E}\left[\langle \mathbf{F}(\mathbf{z}^k) - \mathbf{F}(\mathbf{z}^{k-1}) - (\mathbf{y}^k - \mathbf{y}^{k-1}), \mathbf{z}^{k+1} - \mathbf{z}^k\rangle\right]$$

$$+ \frac{4\eta_z \overline{L}^2}{b}\mathbb{E}\left[\left\|\mathbf{z}^k - \mathbf{w}^{k-1}\right\|^2\right] + \frac{4\eta_z \alpha^2 \overline{L}^2}{b}\mathbb{E}\left[\left\|\mathbf{z}^k - \mathbf{z}^{k-1}\right\|^2\right].$$

Using Assumption 2.2 of $\mu$-strong monotonicity and Young's inequality we get

$$\frac{1}{\eta_z}\mathbb{E}\left[\left\|\mathbf{z}^{k+1} - \mathbf{z}^*\right\|^2\right] \leq \frac{1}{\eta_z}\mathbb{E}\left[\left\|\mathbf{z}^k - \mathbf{z}^*\right\|^2\right] + \frac{\omega}{\eta_z}\mathbb{E}\left[\left\|\mathbf{w}^k - \mathbf{z}^*\right\|^2\right] - \frac{\omega}{\eta_z}\mathbb{E}\left[\left\|\mathbf{z}^{k+1} - \mathbf{w}^k\right\|^2\right]$$

$$- \frac{\omega}{\eta_z} \mathbb{E}\left[\left\|\mathbf{z}^k - \mathbf{z}^*\right\|^2\right] - \frac{1/2 - \omega}{\eta_z} \mathbb{E}\left[\left\|\mathbf{z}^{k+1} - \mathbf{z}^k\right\|^2\right]$$

$$- 2\mu \mathbb{E}\left[\left\|\mathbf{z}^{k+1} - \mathbf{z}^*\right\|^2\right] + 2\nu \mathbb{E}\left[\langle \mathbf{z}^k - \mathbf{z}^*, \mathbf{z}^{k+1} - \mathbf{z}^*\rangle\right]$$

$$+ 2\mathbb{E}\left[\langle \mathbf{y}^{k+1} - \mathbf{y}^*, \mathbf{z}^{k+1} - \mathbf{z}^*\rangle\right]$$

$$+ 2\mathbb{E}\left[\langle \mathbf{F}(\mathbf{z}^{k+1}) - \mathbf{F}(\mathbf{z}^k) - (\mathbf{y}^{k+1} - \mathbf{y}^k), \mathbf{z}^{k+1} - \mathbf{z}^*\rangle\right]$$

$$- 2\alpha \mathbb{E}\left[\langle \mathbf{F}(\mathbf{z}^k) - \mathbf{F}(\mathbf{z}^{k-1}) - (\mathbf{y}^k - \mathbf{y}^{k-1}), \mathbf{z}^k - \mathbf{z}^*\rangle\right]$$

$$- 2\alpha \mathbb{E}\left[\langle \mathbf{F}(\mathbf{z}^k) - \mathbf{F}(\mathbf{z}^{k-1}) - (\mathbf{y}^k - \mathbf{y}^{k-1}), \mathbf{z}^{k+1} - \mathbf{z}^k\rangle\right]$$

$$+ \frac{4\eta_z \overline{L}^2}{b} \mathbb{E}\left[\left\|\mathbf{z}^k - \mathbf{w}^{k-1}\right\|^2\right] + \frac{4\eta_z \alpha^2 \overline{L}^2}{b} \mathbb{E}\left[\left\|\mathbf{z}^k - \mathbf{z}^{k-1}\right\|^2\right]$$

$$\leq \frac{1}{\eta_z} \mathbb{E}\left[\left\|\mathbf{z}^k - \mathbf{z}^*\right\|^2\right] + \frac{\omega}{\eta_z} \mathbb{E}\left[\left\|w^k - \mathbf{z}^*\right\|^2\right] - \frac{\omega}{\eta_z} \mathbb{E}\left[\left\|\mathbf{z}^{k+1} - \mathbf{w}^k\right\|^2\right]$$

$$- \frac{\omega}{\eta_z} \mathbb{E}\left[\left\|\mathbf{z}^k - \mathbf{z}^*\right\|^2\right] - \frac{1/2 - \omega}{\eta_z} \mathbb{E}\left[\left\|\mathbf{z}^{k+1} - \mathbf{z}^k\right\|^2\right]$$

$$- 2\mu \mathbb{E}\left[\left\|\mathbf{z}^{k+1} - \mathbf{z}^*\right\|^2\right] + \nu \mathbb{E}\left[\left\|\mathbf{z}^k - \mathbf{z}^*\right\|^2\right] + \nu \mathbb{E}\left[\left\|\mathbf{z}^{k+1} - \mathbf{z}^*\right\|^2\right]$$

$$+ 2\mathbb{E}\left[\langle \mathbf{y}^{k+1} - \mathbf{y}^*, \mathbf{z}^{k+1} - \mathbf{z}^*\rangle\right]$$

$$+ 2\mathbb{E}\left[\langle \mathbf{F}(\mathbf{z}^{k+1}) - \mathbf{F}(\mathbf{z}^k) - (\mathbf{y}^{k+1} - \mathbf{y}^k), \mathbf{z}^{k+1} - \mathbf{z}^*\rangle\right]$$

$$- 2\alpha \mathbb{E}\left[\langle \mathbf{F}(\mathbf{z}^k) - \mathbf{F}(\mathbf{z}^{k-1}) - (\mathbf{y}^k - \mathbf{y}^{k-1}), \mathbf{z}^k - \mathbf{z}^*\rangle\right]$$

$$+ 2\alpha \mathbb{E}\left[\|\mathbf{F}(\mathbf{z}^k) - \mathbf{F}(\mathbf{z}^{k-1})\|\|\mathbf{z}^{k+1} - \mathbf{z}^k\|\right] + 2\alpha \mathbb{E}\left[\|\mathbf{y}^k - \mathbf{y}^{k-1}\|\|\mathbf{z}^{k+1} - \mathbf{z}^k\|\right]$$

$$+ \frac{4\eta_z \overline{L}^2}{b} \mathbb{E}\left[\left\|\mathbf{z}^k - \mathbf{w}^{k-1}\right\|^2\right] + \frac{4\eta_z \alpha^2 \overline{L}^2}{b} \mathbb{E}\left[\left\|\mathbf{z}^k - \mathbf{z}^{k-1}\right\|^2\right].$$

With $L$-Lipschitzness of $\mathbf{F}(\cdot)$ (Assumption 2.1) we have

$$\frac{1}{\eta_z} \mathbb{E}\left[\left\|\mathbf{z}^{k+1} - \mathbf{z}^*\right\|^2\right] \leq \frac{1}{\eta_z} \mathbb{E}\left[\left\|\mathbf{z}^k - \mathbf{z}^*\right\|^2\right] + \frac{\omega}{\eta_z} \mathbb{E}\left[\left\|\mathbf{w}^k - \mathbf{z}^*\right\|^2\right] - \frac{\omega}{\eta_z} \mathbb{E}\left[\left\|\mathbf{z}^{k+1} - \mathbf{w}^k\right\|^2\right]$$

$$- \left(\frac{\omega}{\eta_z} - \nu\right) \mathbb{E}\left[\left\|\mathbf{z}^k - \mathbf{z}^*\right\|^2\right] - \frac{1/2 - \omega}{\eta_z} \mathbb{E}\left[\left\|\mathbf{z}^{k+1} - \mathbf{z}^k\right\|^2\right]$$

$$- (2\mu - \nu)\mathbb{E}\left[\left\|\mathbf{z}^{k+1} - \mathbf{z}^*\right\|^2\right] + 2\mathbb{E}\left[\langle y^{k+1} - y^*, \mathbf{z}^{k+1} - \mathbf{z}^*\rangle\right]$$

$$+ 2\mathbb{E}\left[\langle \mathbf{F}(\mathbf{z}^{k+1}) - \mathbf{F}(\mathbf{z}^k) - (\mathbf{y}^{k+1} - \mathbf{y}^k), \mathbf{z}^{k+1} - \mathbf{z}^*\rangle\right]$$

$$- 2\alpha \mathbb{E}\left[\langle \mathbf{F}(\mathbf{z}^k) - \mathbf{F}(\mathbf{z}^{k-1}) - (\mathbf{y}^k - \mathbf{y}^{k-1}), \mathbf{z}^k - \mathbf{z}^*\rangle\right]$$

$$+ 2\alpha L \mathbb{E}\left[\|\mathbf{z}^k - \mathbf{z}^{k-1}\|\|\mathbf{z}^{k+1} - \mathbf{z}^k\|\right] + 2\alpha \mathbb{E}\left[\|\mathbf{y}^k - \mathbf{y}^{k-1}\|\|\mathbf{z}^{k+1} - \mathbf{z}^k\|\right]$$

$$+ \frac{4\eta_z \overline{L}^2}{b} \mathbb{E}\left[\left\|\mathbf{z}^k - \mathbf{w}^{k-1}\right\|^2\right] + \frac{4\eta_z \alpha^2 \overline{L}^2}{b} \mathbb{E}\left[\left\|\mathbf{z}^k - \mathbf{z}^{k-1}\right\|^2\right]$$

Young's inequality we get

$$\frac{1}{\eta_z} \mathbb{E}\left[\left\|\mathbf{z}^{k+1} - \mathbf{z}^*\right\|^2\right] \leq \frac{1}{\eta_z} \mathbb{E}\left[\left\|\mathbf{z}^k - \mathbf{z}^*\right\|^2\right] + \frac{\omega}{\eta_z} \mathbb{E}\left[\left\|\mathbf{w}^k - \mathbf{z}^*\right\|^2\right] - \frac{\omega}{\eta_z} \mathbb{E}\left[\left\|\mathbf{z}^{k+1} - \mathbf{w}^k\right\|^2\right]$$

$$- \left(\frac{\omega}{\eta_z} - \nu\right) \mathbb{E}\left[\left\|\mathbf{z}^k - \mathbf{z}^*\right\|^2\right] - \frac{1/2 - \omega}{\eta_z} \mathbb{E}\left[\left\|\mathbf{z}^{k+1} - \mathbf{z}^k\right\|^2\right]$$

$$- (2\mu - \nu)\mathbb{E}\left[\left\|\mathbf{z}^{k+1} - \mathbf{z}^*\right\|^2\right] + 2\mathbb{E}\left[\langle y^{k+1} - y^*, \mathbf{z}^{k+1} - \mathbf{z}^*\rangle\right]$$

$$+ 2\mathbb{E}\left[\langle \mathbf{F}(\mathbf{z}^{k+1}) - \mathbf{F}(\mathbf{z}^k) - (y^{k+1} - y^k), \mathbf{z}^{k+1} - \mathbf{z}^*\rangle\right]$$

$$- 2\alpha \mathbb{E}\left[\langle \mathbf{F}(\mathbf{z}^k) - \mathbf{F}(\mathbf{z}^{k-1}) - (y^k - y^{k-1}), \mathbf{z}^k - \mathbf{z}^*\rangle\right]$$

$$
+ \alpha L \mathbb{E}\left[\left\|\mathbf{z}^k - \mathbf{z}^{k-1}\right\|^2\right] + \alpha L \mathbb{E}\left[\left\|\mathbf{z}^{k+1} - \mathbf{z}^k\right\|^2\right]
$$

$$
+ \frac{\alpha}{2\eta_y} \mathbb{E}\left[\left\|\mathbf{y}^k - \mathbf{y}^{k-1}\right\|^2\right] + 2\alpha\eta_y \mathbb{E}\left[\left\|\mathbf{z}^{k+1} - \mathbf{z}^k\right\|^2\right]
$$

$$
+ \frac{4\eta_z \overline{L}^2}{b} \mathbb{E}\left[\left\|\mathbf{z}^k - \mathbf{w}^{k-1}\right\|^2\right] + \frac{4\eta_z \alpha^2 \overline{L}^2}{b} \mathbb{E}\left[\left\|\mathbf{z}^k - \mathbf{z}^{k-1}\right\|^2\right].
$$

Using the assumption on $\eta_z$ we get that $L \leq \frac{1}{8\eta_z}$ and $\eta_y \leq \frac{1}{32\eta_z}$

$$
\begin{aligned}
\frac{1}{\eta_z} \mathbb{E}\left[\left\|\mathbf{z}^{k+1} - \mathbf{z}^*\right\|^2\right] &\leq \frac{1}{\eta_z} \mathbb{E}\left[\left\|\mathbf{z}^k - \mathbf{z}^*\right\|^2\right] + \frac{\omega}{\eta_z} \mathbb{E}\left[\left\|\mathbf{w}^k - \mathbf{z}^*\right\|^2\right] - \frac{\omega}{\eta_z} \mathbb{E}\left[\left\|\mathbf{z}^{k+1} - \mathbf{w}^k\right\|^2\right] \\
&\quad - \left(\frac{\omega}{\eta_z} - \nu\right) \mathbb{E}\left[\left\|\mathbf{z}^k - \mathbf{z}^*\right\|^2\right] - \frac{1/2 - \omega}{\eta_z} \mathbb{E}\left[\left\|\mathbf{z}^{k+1} - \mathbf{z}^k\right\|^2\right] \\
&\quad - (2\mu - \nu)\mathbb{E}\left[\left\|\mathbf{z}^{k+1} - \mathbf{z}^*\right\|^2\right] + 2\mathbb{E}\left[\langle \mathbf{y}^{k+1} - \mathbf{y}^*, \mathbf{z}^{k+1} - \mathbf{z}^*\rangle\right] \\
&\quad + 2\mathbb{E}\left[\langle \mathbf{F}(\mathbf{z}^{k+1}) - \mathbf{F}(\mathbf{z}^k) - (\mathbf{y}^{k+1} - \mathbf{y}^k), \mathbf{z}^{k+1} - \mathbf{z}^*\rangle\right] \\
&\quad - 2\alpha\mathbb{E}\left[\langle \mathbf{F}(\mathbf{z}^k) - \mathbf{F}(\mathbf{z}^{k-1}) - (\mathbf{y}^k - \mathbf{y}^{k-1}), \mathbf{z}^k - \mathbf{z}^*\rangle\right] \\
&\quad + \frac{\alpha}{8\eta_z} \mathbb{E}\left[\left\|\mathbf{z}^k - \mathbf{z}^{k-1}\right\|^2\right] + \frac{\alpha}{8\eta_z} \mathbb{E}\left[\left\|\mathbf{z}^{k+1} - \mathbf{z}^k\right\|^2\right] \\
&\quad + \frac{\alpha}{2\eta_y} \mathbb{E}\left[\left\|\mathbf{y}^k - \mathbf{y}^{k-1}\right\|^2\right] + \frac{\alpha}{16\eta_z} \mathbb{E}\left[\left\|\mathbf{z}^{k+1} - \mathbf{z}^k\right\|^2\right] \\
&\quad + \frac{4\eta_z \overline{L}^2}{b} \mathbb{E}\left[\left\|\mathbf{z}^k - \mathbf{w}^{k-1}\right\|^2\right] + \frac{4\eta_z \alpha^2 \overline{L}^2}{b} \mathbb{E}\left[\left\|\mathbf{z}^k - \mathbf{z}^{k-1}\right\|^2\right].
\end{aligned}
$$

With $\alpha \leq 1$ and $\eta_z \leq \frac{\sqrt{\alpha b}}{8\overline{L}}$ we obtain

$$
\begin{aligned}
\frac{1}{\eta_z} \mathbb{E}\left[\left\|\mathbf{z}^{k+1} - \mathbf{z}^*\right\|^2\right] &\leq \frac{1}{\eta_z} \mathbb{E}\left[\left\|\mathbf{z}^k - \mathbf{z}^*\right\|^2\right] + \frac{\omega}{\eta_z} \mathbb{E}\left[\left\|\mathbf{w}^k - \mathbf{z}^*\right\|^2\right] - \frac{\omega}{\eta_z} \mathbb{E}\left[\left\|\mathbf{z}^{k+1} - \mathbf{w}^k\right\|^2\right] \\
&\quad - \left(\frac{\omega}{\eta_z} - \nu\right) \mathbb{E}\left[\left\|\mathbf{z}^k - \mathbf{z}^*\right\|^2\right] - \frac{1/2 - \omega}{\eta_z} \mathbb{E}\left[\left\|\mathbf{z}^{k+1} - \mathbf{z}^k\right\|^2\right] \\
&\quad - (2\mu - \nu)\mathbb{E}\left[\left\|\mathbf{z}^{k+1} - \mathbf{z}^*\right\|^2\right] + 2\mathbb{E}\left[\langle \mathbf{y}^{k+1} - \mathbf{y}^*, \mathbf{z}^{k+1} - \mathbf{z}^*\rangle\right] \\
&\quad + 2\mathbb{E}\left[\langle \mathbf{F}(\mathbf{z}^{k+1}) - \mathbf{F}(\mathbf{z}^k) - (\mathbf{y}^{k+1} - \mathbf{y}^k), \mathbf{z}^{k+1} - \mathbf{z}^*\rangle\right] \\
&\quad - 2\alpha\mathbb{E}\left[\langle \mathbf{F}(\mathbf{z}^k) - \mathbf{F}(\mathbf{z}^{k-1}) - (\mathbf{y}^k - \mathbf{y}^{k-1}), \mathbf{z}^k - \mathbf{z}^*\rangle\right] \\
&\quad + \frac{\alpha}{8\eta_z} \mathbb{E}\left[\left\|\mathbf{z}^k - \mathbf{z}^{k-1}\right\|^2\right] + \frac{1}{8\eta_z} \mathbb{E}\left[\left\|\mathbf{z}^{k+1} - \mathbf{z}^k\right\|^2\right] \\
&\quad + \frac{\alpha}{2\eta_y} \mathbb{E}\left[\left\|\mathbf{y}^k - \mathbf{y}^{k-1}\right\|^2\right] + \frac{1}{16\eta_z} \mathbb{E}\left[\left\|\mathbf{z}^{k+1} - \mathbf{z}^k\right\|^2\right] \\
&\quad + \frac{4\eta_z \overline{L}^2}{b} \mathbb{E}\left[\left\|\mathbf{z}^k - \mathbf{w}^{k-1}\right\|^2\right] + \frac{\alpha}{16\eta_z} \mathbb{E}\left[\left\|\mathbf{z}^k - \mathbf{z}^{k-1}\right\|^2\right] \\
&\leq \frac{1}{\eta_z} \mathbb{E}\left[\left\|\mathbf{z}^k - \mathbf{z}^*\right\|^2\right] + \frac{\omega}{\eta_z} \mathbb{E}\left[\left\|\mathbf{w}^k - \mathbf{z}^*\right\|^2\right] - \frac{\omega}{\eta_z} \mathbb{E}\left[\left\|\mathbf{z}^{k+1} - \mathbf{w}^k\right\|^2\right] \\
&\quad - \left(\frac{\omega}{\eta_z} - \nu\right) \mathbb{E}\left[\left\|\mathbf{z}^k - \mathbf{z}^*\right\|^2\right] - \frac{5/16 - \omega}{\eta_z} \mathbb{E}\left[\left\|\mathbf{z}^{k+1} - \mathbf{z}^k\right\|^2\right] \\
&\quad + \frac{\alpha}{4\eta_z} \mathbb{E}\left[\left\|\mathbf{z}^k - \mathbf{z}^{k-1}\right\|^2\right] - (2\mu - \nu)\mathbb{E}\left[\left\|\mathbf{z}^{k+1} - \mathbf{z}^*\right\|^2\right] \\
&\quad + 2\mathbb{E}\left[\langle \mathbf{y}^{k+1} - \mathbf{y}^*, \mathbf{z}^{k+1} - \mathbf{z}^*\rangle\right] \\
&\quad + 2\mathbb{E}\left[\langle \mathbf{F}(\mathbf{z}^{k+1}) - \mathbf{F}(\mathbf{z}^k) - (\mathbf{y}^{k+1} - \mathbf{y}^k), \mathbf{z}^{k+1} - \mathbf{z}^*\rangle\right] \\
&\quad - 2\alpha\mathbb{E}\left[\langle \mathbf{F}(\mathbf{z}^k) - \mathbf{F}(\mathbf{z}^{k-1}) - (\mathbf{y}^k - \mathbf{y}^{k-1}), \mathbf{z}^k - \mathbf{z}^*\rangle\right]
\end{aligned}
$$

$$+ \frac{\alpha}{2\eta_y} \mathbb{E}\left[\left\|\mathbf{y}^k - \mathbf{y}^{k-1}\right\|^2\right] + \frac{4\eta_z \overline{L}^2}{b} \mathbb{E}\left[\left\|\mathbf{z}^k - \mathbf{w}^{k-1}\right\|^2\right].$$

Choice of $\omega \leq \frac{1}{16}$ and small rearrangement give

$$\frac{1}{\eta_z}\mathbb{E}\left[\left\|\mathbf{z}^{k+1} - \mathbf{z}^*\right\|^2\right] + (2\mu - \nu)\mathbb{E}\left[\left\|\mathbf{z}^{k+1} - \mathbf{z}^*\right\|^2\right] + \frac{1}{4\eta_z}\mathbb{E}\left[\left\|\mathbf{z}^{k+1} - \mathbf{z}^k\right\|^2\right]$$
$$- 2\mathbb{E}\left[\langle \mathbf{F}(\mathbf{z}^{k+1}) - \mathbf{F}(\mathbf{z}^k) - (\mathbf{y}^{k+1} - \mathbf{y}^k), \mathbf{z}^{k+1} - \mathbf{z}^*\rangle\right]$$
$$\leq \frac{1}{\eta_z}\mathbb{E}\left[\left\|\mathbf{z}^k - \mathbf{z}^*\right\|^2\right] - \left(\frac{\omega}{\eta_z} - \nu\right)\mathbb{E}\left[\left\|\mathbf{z}^k - \mathbf{z}^*\right\|^2\right]$$
$$+ \alpha \cdot \frac{1}{4\eta_z}\mathbb{E}\left[\left\|\mathbf{z}^k - \mathbf{z}^{k-1}\right\|^2\right] - \alpha \cdot 2\mathbb{E}\left[\langle \mathbf{F}(\mathbf{z}^k) - \mathbf{F}(\mathbf{z}^{k-1}) - (\mathbf{y}^k - \mathbf{y}^{k-1}), \mathbf{z}^k - \mathbf{z}^*\rangle\right]$$
$$+ \frac{\omega}{\eta_z}\mathbb{E}\left[\left\|\mathbf{w}^k - \mathbf{z}^*\right\|^2\right] - \frac{\omega}{\eta_z}\mathbb{E}\left[\left\|\mathbf{z}^{k+1} - \mathbf{w}^k\right\|^2\right] + 2\mathbb{E}\left[\langle \mathbf{y}^{k+1} - \mathbf{y}^*, \mathbf{z}^{k+1} - \mathbf{z}^*\rangle\right]$$
$$+ \frac{\alpha}{2\eta_y}\mathbb{E}\left[\left\|\mathbf{y}^k - \mathbf{y}^{k-1}\right\|^2\right] + \frac{4\eta_z \overline{L}^2}{b}\mathbb{E}\left[\left\|\mathbf{z}^k - \mathbf{w}^{k-1}\right\|^2\right].$$

**Part 2.** Update for $m^{k+1}$ and Assumption 2.4 on time-varying graph give

$$\left\|m^{k+1}\right\|_{\mathbf{P}}^2 \leq (1 - \chi^{-1}(T))\left\|m^k + \eta_x \Delta_x^k\right\|_{\mathbf{P}}^2$$
$$\leq (1 - \chi^{-1}(T))\left((1 + (2\chi(T))^{-1})\left\|m^k\right\|_{\mathbf{P}}^2 + (1 + 2\chi(T))\left\|\eta_x \Delta_x^k\right\|_{\mathbf{P}}^2\right)$$
$$\leq (1 - (2\chi(T))^{-1})\left\|m^k\right\|_{\mathbf{P}}^2 + 2\eta_x^2 \chi(T)\left\|\Delta_x^k\right\|_{\mathbf{P}}^2.$$

After rearranging we get

$$\left\|m^k\right\|_{\mathbf{P}}^2 \leq (1 - (4\chi(T))^{-1})4\chi(T)\left\|m^k\right\|_{\mathbf{P}}^2 - 4\chi(T)\left\|m^{k+1}\right\|_{\mathbf{P}}^2 + 8\eta_x^2 \chi^2(T)\left\|\Delta_x^k\right\|_{\mathbf{P}}^2. \tag{39}$$

**Part 3.** Updates for $\mathbf{x}^{k+1}$ and $m^{k+1}$ of Algorithm 2 imply $\hat{\mathbf{x}}^{k+1} = \hat{\mathbf{x}}^k - \eta_x \mathbf{P}\Delta_x^k$. By this together with update for $\mathbf{y}^{k+1}$ of Algorithm 2 we obtain

$$\frac{1}{\eta_y}\left\|\mathbf{y}^{k+1} - \mathbf{y}^*\right\|^2 + \frac{1}{\eta_x}\left\|\hat{\mathbf{x}}^{k+1} - \mathbf{x}^*\right\|^2$$
$$= \frac{1}{\eta_y}\left\|\mathbf{y}^k - \mathbf{y}^*\right\|^2 + \frac{1}{\eta_x}\left\|\hat{\mathbf{x}}^k - \mathbf{x}^*\right\|^2 - \frac{1}{\eta_y}\left\|\mathbf{y}^{k+1} - \mathbf{y}^k\right\|^2 + \frac{1}{\eta_x}\left\|\hat{\mathbf{x}}^{k+1} - \hat{\mathbf{x}}^k\right\|^2$$
$$+ \frac{2}{\eta_y}\langle \mathbf{y}^{k+1} - \mathbf{y}^k, \mathbf{y}^{k+1} - \mathbf{y}^*\rangle + \frac{2}{\eta_x}\langle \hat{\mathbf{x}}^{k+1} - \hat{\mathbf{x}}^k, \hat{\mathbf{x}}^k - \mathbf{x}^*\rangle$$
$$\leq \frac{1}{\eta_y}\left\|\mathbf{y}^k - \mathbf{y}^*\right\|^2 + \frac{1}{\eta_x}\left\|\hat{\mathbf{x}}^k - \mathbf{x}^*\right\|^2 - \frac{1}{\eta_y}\left\|\mathbf{y}^{k+1} - \mathbf{y}^k\right\|^2$$
$$- 2\langle \Delta_y^k, \mathbf{y}^{k+1} - \mathbf{y}^*\rangle - 2\langle \mathbf{P}\Delta_x^k, \mathbf{x}^k - \mathbf{x}^*\rangle + 2\left\|\Delta_x^k\right\|_{\mathbf{P}}\left\|m^k\right\|_{\mathbf{P}} + \eta_x\left\|\Delta_x^k\right\|_{\mathbf{P}}^2.$$

Using the definitions of $\Delta_y^k$ and $\Delta_x^k$ we get

$$\frac{1}{\eta_y}\left\|\mathbf{y}^{k+1} - \mathbf{y}^*\right\|^2 + \frac{1}{\eta_x}\left\|\hat{\mathbf{x}}^{k+1} - \mathbf{x}^*\right\|^2$$
$$\leq \frac{1}{\eta_y}\left\|\mathbf{y}^k - \mathbf{y}^*\right\|^2 + \frac{1}{\eta_x}\left\|\hat{\mathbf{x}}^k - \mathbf{x}^*\right\|^2 - \frac{1}{\eta_y}\left\|\mathbf{y}^{k+1} - \mathbf{y}^k\right\|^2 + 2\left\|\Delta_x^k\right\|_{\mathbf{P}}\left\|m^k\right\|_{\mathbf{P}} + \eta_x\left\|\Delta_x^k\right\|_{\mathbf{P}}^2$$
$$- 2\langle \nu^{-1}(\mathbf{y}_c^k + \mathbf{x}_c^k) + \mathbf{z}^{k+1} + \gamma(\mathbf{y}^k + \mathbf{x}^k + \nu\mathbf{z}^k), \mathbf{y}^{k+1} - \mathbf{y}^*\rangle$$
$$- 2\langle \nu^{-1}\mathbf{P}(\mathbf{y}_c^k + \mathbf{x}_c^k) + \beta\mathbf{P}(\mathbf{x}^k + \Delta^{k+1/2}), \mathbf{x}^k - \mathbf{x}^*\rangle$$
$$= \frac{1}{\eta_y}\left\|\mathbf{y}^k - \mathbf{y}^*\right\|^2 + \frac{1}{\eta_x}\left\|\hat{\mathbf{x}}^k - \mathbf{x}^*\right\|^2 - \frac{1}{\eta_y}\left\|\mathbf{y}^{k+1} - \mathbf{y}^k\right\|^2 + 2\left\|\Delta_x^k\right\|_{\mathbf{P}}\left\|m^k\right\|_{\mathbf{P}} + \eta_x\left\|\Delta_x^k\right\|_{\mathbf{P}}^2$$
$$- 2\langle \nu^{-1}(\mathbf{y}_c^k + \mathbf{x}_c^k) + \mathbf{z}^{k+1} + \gamma(\mathbf{y}^k + \mathbf{x}^k + \nu\mathbf{z}^k), \mathbf{y}^{k+1} - \mathbf{y}^*\rangle$$
$$- 2\langle \nu^{-1}\mathbf{P}(\mathbf{y}_c^k + \mathbf{x}_c^k), \mathbf{x}^k - \mathbf{x}^*\rangle - 2\langle \beta\mathbf{P}(\mathbf{x}^k + \Delta^{k+1/2}), \mathbf{x}^k - \mathbf{x}^*\rangle.$$

Definitions (37), (38) gives $\mathbf{PF}(\mathbf{z}^*) + \mathbf{x}^* = 0$ $\mathbf{P}(\mathbf{y}^* + \mathbf{x}^*) = 0$, $\nu^{-1}(\mathbf{y}^* + \mathbf{x}^*) + \mathbf{z}^* = 0$. Additionally, using $\mathbf{Px}^k = \mathbf{x}^k$, we get

$$\frac{1}{\eta_y}\left\|\mathbf{y}^{k+1} - \mathbf{y}^*\right\|^2 + \frac{1}{\eta_x}\left\|\hat{\mathbf{x}}^{k+1} - \mathbf{x}^*\right\|^2$$

$$\leq \frac{1}{\eta_y}\left\|\mathbf{y}^k - \mathbf{y}^*\right\|^2 + \frac{1}{\eta_x}\left\|\hat{\mathbf{x}}^k - \mathbf{x}^*\right\|^2 - \frac{1}{\eta_y}\left\|\mathbf{y}^{k+1} - \mathbf{y}^k\right\|^2 + 2\left\|\Delta_x^k\right\|_{\mathbf{P}}\left\|m^k\right\|_{\mathbf{P}} + \eta_x\left\|\Delta_x^k\right\|_{\mathbf{P}}^2$$
$$- 2\langle\nu^{-1}(\mathbf{y}_c^k + \mathbf{x}_c^k) + \mathbf{z}^{k+1} + \gamma(\mathbf{y}^k + \mathbf{x}^k + \nu\mathbf{z}^k), \mathbf{y}^{k+1} - \mathbf{y}^*\rangle$$
$$- 2\langle\nu^{-1}(\mathbf{y}_c^k + \mathbf{x}_c^k), \mathbf{x}^k - \mathbf{x}^*\rangle - 2\langle\beta(\mathbf{x}^k + \mathbf{P}\Delta^{k+1/2}), \mathbf{x}^k - \mathbf{x}^*\rangle$$
$$= \frac{1}{\eta_y}\left\|\mathbf{y}^k - \mathbf{y}^*\right\|^2 + \frac{1}{\eta_x}\left\|\hat{\mathbf{x}}^k - \mathbf{x}^*\right\|^2 - \frac{1}{\eta_y}\left\|\mathbf{y}^{k+1} - \mathbf{y}^k\right\|^2 + 2\left\|\Delta_x^k\right\|_{\mathbf{P}}\left\|m^k\right\|_{\mathbf{P}} + \eta_x\left\|\Delta_x^k\right\|_{\mathbf{P}}^2$$
$$- 2\langle\nu^{-1}(\mathbf{y}_c^k + \mathbf{x}_c^k) + \mathbf{z}^{k+1} - \nu^{-1}(\mathbf{y}^* + \mathbf{x}^*) - \mathbf{z}^*, \mathbf{y}^{k+1} - \mathbf{y}^*\rangle$$
$$- 2\gamma\langle\mathbf{y}^k + \mathbf{x}^k + \nu\mathbf{z}^k - (\mathbf{y}^* + \mathbf{x}^* + \nu\mathbf{z}^*), \mathbf{y}^{k+1} - \mathbf{y}^*\rangle$$
$$- 2\langle\nu^{-1}(\mathbf{y}_c^k + \mathbf{x}_c^k - \mathbf{y}^* - \mathbf{x}^*), \mathbf{x}^k - \mathbf{x}^*\rangle$$
$$- 2\beta\langle\mathbf{x}^k + \mathbf{P}\Delta^{k+1/2} - \mathbf{PF}(\mathbf{z}^*) - \mathbf{x}^*, \mathbf{x}^k - \mathbf{x}^*\rangle.$$

And then

$$\frac{1}{\eta_y}\left\|\mathbf{y}^{k+1} - \mathbf{y}^*\right\|^2 + \frac{1}{\eta_x}\left\|\hat{\mathbf{x}}^{k+1} - \mathbf{x}^*\right\|^2$$
$$\leq \frac{1}{\eta_y}\left\|\mathbf{y}^k - \mathbf{y}^*\right\|^2 + \frac{1}{\eta_x}\left\|\hat{\mathbf{x}}^k - \mathbf{x}^*\right\|^2 - \frac{1}{\eta_y}\left\|\mathbf{y}^{k+1} - \mathbf{y}^k\right\|^2 + 2\left\|\Delta_x^k\right\|_{\mathbf{P}}\left\|m^k\right\|_{\mathbf{P}} + \eta_x\left\|\Delta_x^k\right\|_{\mathbf{P}}^2$$
$$- 2\langle\mathbf{z}^{k+1} - \mathbf{z}^*, \mathbf{y}^{k+1} - \mathbf{y}^*\rangle - 2\nu^{-1}\langle\mathbf{y}_c^k + \mathbf{x}_c^k - \mathbf{y}^* - \mathbf{x}^*, \mathbf{y}^{k+1} + \mathbf{x}^k - \mathbf{y}^* - \mathbf{x}^*\rangle$$
$$- 2\gamma\langle\mathbf{y}^k + \mathbf{x}^k + \nu\mathbf{z}^k - \mathbf{y}^* - \mathbf{x}^* - \nu\mathbf{z}^*, \mathbf{y}^{k+1} - \mathbf{y}^*\rangle$$
$$- 2\beta\langle\mathbf{x}^k + \mathbf{P}\Delta^{k+1/2} - \mathbf{x}^* - \mathbf{PF}(\mathbf{z}^*), \mathbf{x}^k - \mathbf{x}^*\rangle.$$

Using parallelogram rule we get

$$\frac{1}{\eta_y}\left\|\mathbf{y}^{k+1} - \mathbf{y}^*\right\|^2 + \frac{1}{\eta_x}\left\|\hat{\mathbf{x}}^{k+1} - \mathbf{x}^*\right\|^2$$
$$\leq \frac{1}{\eta_y}\left\|\mathbf{y}^k - \mathbf{y}^*\right\|^2 + \frac{1}{\eta_x}\left\|\hat{\mathbf{x}}^k - \mathbf{x}^*\right\|^2 - \frac{1}{\eta_y}\left\|\mathbf{y}^{k+1} - \mathbf{y}^k\right\|^2 + 2\left\|\Delta_x^k\right\|_{\mathbf{P}}\left\|m^k\right\|_{\mathbf{P}} + \eta_x\left\|\Delta_x^k\right\|_{\mathbf{P}}^2$$
$$- 2\langle\mathbf{z}^{k+1} - \mathbf{z}^*, \mathbf{y}^{k+1} - \mathbf{y}^*\rangle - 2\nu^{-1}\langle\mathbf{y}_c^k + \mathbf{x}_c^k - \mathbf{y}^* - \mathbf{x}^*, \mathbf{y}^{k+1} + \mathbf{x}^k - \mathbf{y}^* - \mathbf{x}^*\rangle$$
$$- \gamma\left(\left\|\mathbf{y}^k - \mathbf{y}^*\right\|^2 + \left\|\mathbf{y}^{k+1} - \mathbf{y}^*\right\|^2 - \left\|\mathbf{y}^{k+1} - \mathbf{y}^k\right\|^2\right)$$
$$+ \gamma\left(\left\|\mathbf{x}^k - \mathbf{x}^* + \nu(\mathbf{z}^k - \mathbf{z}^*)\right\|^2 + \left\|\mathbf{y}^{k+1} - \mathbf{y}^*\right\|^2\right)$$
$$- 2\beta\left\|\mathbf{x}^k - \mathbf{x}^*\right\|^2 + \beta\left\|\mathbf{x}^k - \mathbf{x}^*\right\|^2 + \beta\left\|\Delta^{k+1/2} - \mathbf{F}(\mathbf{z}^*)\right\|^2$$
$$\leq \frac{1}{\eta_y}\left\|\mathbf{y}^k - \mathbf{y}^*\right\|^2 + \frac{1}{\eta_x}\left\|\hat{\mathbf{x}}^k - \mathbf{x}^*\right\|^2 - \left(\frac{1}{\eta_y} - \gamma\right)\left\|\mathbf{y}^{k+1} - \mathbf{y}^k\right\|^2$$
$$+ 2\left\|\Delta_x^k\right\|_{\mathbf{P}}\left\|m^k\right\|_{\mathbf{P}} + \eta_x\left\|\Delta_x^k\right\|_{\mathbf{P}}^2$$
$$- 2\langle\mathbf{z}^{k+1} - \mathbf{z}^*, \mathbf{y}^{k+1} - \mathbf{y}^*\rangle - 2\nu^{-1}\langle\mathbf{y}_c^k + \mathbf{x}_c^k - \mathbf{y}^* - \mathbf{x}^*, \mathbf{y}^{k+1} + \mathbf{x}^k - \mathbf{y}^* - \mathbf{x}^*\rangle$$
$$- \gamma\left\|\mathbf{y}^k - \mathbf{y}^*\right\|^2 - (\beta - 2\gamma)\left\|\mathbf{x}^k - \mathbf{x}^*\right\|^2 + 2\gamma\nu^2\left\|\mathbf{z}^k - \mathbf{z}^*\right\|^2 + \beta\left\|\Delta^{k+1/2} - \mathbf{F}(\mathbf{z}^*)\right\|^2.$$

The definition of $\beta = 5\gamma$ gives

$$\frac{1}{\eta_y}\left\|\mathbf{y}^{k+1} - \mathbf{y}^*\right\|^2 + \frac{1}{\eta_x}\left\|\hat{\mathbf{x}}^{k+1} - \mathbf{x}^*\right\|^2$$
$$\leq \frac{1}{\eta_y}\left\|\mathbf{y}^k - \mathbf{y}^*\right\|^2 + \frac{1}{\eta_x}\left\|\hat{\mathbf{x}}^k - \mathbf{x}^*\right\|^2 - \left(\frac{1}{\eta_y} - \gamma\right)\left\|\mathbf{y}^{k+1} - \mathbf{y}^k\right\|^2$$
$$+ 2\left\|\Delta_x^k\right\|_{\mathbf{P}}\left\|m^k\right\|_{\mathbf{P}} + \eta_x\left\|\Delta_x^k\right\|_{\mathbf{P}}^2$$
$$- 2\langle\mathbf{z}^{k+1} - \mathbf{z}^*, \mathbf{y}^{k+1} - \mathbf{y}^*\rangle - 2\nu^{-1}\langle\mathbf{y}_c^k + \mathbf{x}_c^k - \mathbf{y}^* - \mathbf{x}^*, \mathbf{y}^{k+1} + \mathbf{x}^k - \mathbf{y}^* - \mathbf{x}^*\rangle$$
$$- \gamma\left\|\mathbf{y}^k - \mathbf{y}^*\right\|^2 - 3\gamma\left\|\mathbf{x}^k - \mathbf{x}^*\right\|^2 + 2\gamma\nu^2\left\|\mathbf{z}^k - \mathbf{z}^*\right\|^2 + 5\gamma\left\|\Delta^{k+1/2} - \mathbf{F}(\mathbf{z}^*)\right\|^2.$$

Using the definition (35) of $\hat{\mathbf{x}}^k$, we get

$$\frac{1}{\eta_y}\left\|\mathbf{y}^{k+1} - \mathbf{y}^*\right\|^2 + \frac{1}{\eta_x}\left\|\hat{\mathbf{x}}^{k+1} - \mathbf{x}^*\right\|^2$$

$$\leq \frac{1}{\eta_y}\left\|\mathbf{y}^k-\mathbf{y}^*\right\|^2+\frac{1}{\eta_x}\left\|\hat{\mathbf{x}}^k-\mathbf{x}^*\right\|^2-\left(\frac{1}{\eta_y}-\gamma\right)\left\|\mathbf{y}^{k+1}-\mathbf{y}^k\right\|^2$$

$$+2\left\|\Delta_x^k\right\|_{\mathbf{P}}\left\|m^k\right\|_{\mathbf{P}}+\eta_x\left\|\Delta_x^k\right\|_{\mathbf{P}}^2$$

$$-2\langle\mathbf{z}^{k+1}-\mathbf{z}^*,\mathbf{y}^{k+1}-\mathbf{y}^*\rangle-2\nu^{-1}\langle\mathbf{y}_c^k+\mathbf{x}_c^k-\mathbf{y}^*-\mathbf{x}^*,\mathbf{y}^{k+1}+\mathbf{x}^k-\mathbf{y}^*-\mathbf{x}^*\rangle$$

$$-\gamma\left\|\mathbf{y}^k-\mathbf{y}^*\right\|^2-\gamma\left\|\hat{\mathbf{x}}^k-\mathbf{x}^*\right\|^2-\gamma\left\|\mathbf{x}^k-\mathbf{x}^*\right\|^2+2\gamma\left\|m^k\right\|_{\mathbf{P}}^2+2\gamma\nu^2\left\|\mathbf{z}^k-\mathbf{z}^*\right\|^2$$

$$+5\gamma\left\|\Delta^{k+1/2}-\mathbf{F}(\mathbf{z}^*)\right\|^2$$

$$=\left(\frac{1}{\eta_y}-\gamma\right)\left\|\mathbf{y}^k-\mathbf{y}^*\right\|^2+\left(\frac{1}{\eta_x}-\gamma\right)\left\|\hat{\mathbf{x}}^k-\mathbf{x}^*\right\|^2-\left(\frac{1}{\eta_y}-\gamma\right)\left\|\mathbf{y}^{k+1}-\mathbf{y}^k\right\|^2$$

$$+2\gamma\left\|m^k\right\|_{\mathbf{P}}^2+2\left\|\Delta_x^k\right\|_{\mathbf{P}}\left\|m^k\right\|_{\mathbf{P}}+\eta_x\left\|\Delta_x^k\right\|_{\mathbf{P}}^2-2\langle\mathbf{z}^{k+1}-\mathbf{z}^*,\mathbf{y}^{k+1}-\mathbf{y}^*\rangle$$

$$-2\nu^{-1}\langle\mathbf{y}_c^k+\mathbf{x}_c^k-\mathbf{y}^*-\mathbf{x}^*,\mathbf{y}^{k+1}+\mathbf{x}^k-\mathbf{y}^*-\mathbf{x}^*\rangle$$

$$-\gamma\left\|\mathbf{x}^k-\mathbf{x}^*\right\|^2+2\gamma\nu^2\left\|\mathbf{z}^k-\mathbf{z}^*\right\|^2+5\gamma\left\|\Delta^{k+1/2}-\mathbf{F}(\mathbf{z}^*)\right\|^2.$$

With updates of $\mathbf{x}_f^{k+1}$ and $\mathbf{y}_f^{k+1}$ from Algorithm 2 we have

$$\frac{1}{\eta_y}\left\|\mathbf{y}^{k+1}-\mathbf{y}^*\right\|^2+\frac{1}{\eta_x}\left\|\hat{\mathbf{x}}^{k+1}-\mathbf{x}^*\right\|^2$$

$$\leq\left(\frac{1}{\eta_y}-\gamma\right)\left\|\mathbf{y}^k-\mathbf{y}^*\right\|^2+\left(\frac{1}{\eta_x}-\gamma\right)\left\|\hat{\mathbf{x}}^k-\mathbf{x}^*\right\|^2-\left(\frac{1}{\eta_y}-\gamma\right)\left\|\mathbf{y}^{k+1}-\mathbf{y}^k\right\|^2$$

$$+2\gamma\left\|m^k\right\|_{\mathbf{P}}^2+2\left\|\Delta_x^k\right\|_{\mathbf{P}}\left\|m^k\right\|_{\mathbf{P}}+\eta_x\left\|\Delta_x^k\right\|_{\mathbf{P}}^2-2\langle\mathbf{z}^{k+1}-\mathbf{z}^*,\mathbf{y}^{k+1}-\mathbf{y}^*\rangle$$

$$-\gamma\left\|\mathbf{x}^k-\mathbf{x}^*\right\|^2+2\gamma\nu^2\left\|\mathbf{z}^k-\mathbf{z}^*\right\|^2+5\gamma\left\|\Delta^{k+1/2}-\mathbf{F}(\mathbf{z}^*)\right\|^2$$

$$-\frac{2\nu^{-1}}{\tau}\langle\mathbf{y}_c^k+\mathbf{x}_c^k-\mathbf{y}^*-\mathbf{x}^*,\mathbf{y}_f^{k+1}+\mathbf{x}_f^{k+1}-\mathbf{y}_c^k-\mathbf{x}_c^k+\theta(\mathbf{W}(k)\otimes\mathbf{I}_d)(\mathbf{y}_c^k+\mathbf{x}_c^k)\rangle$$

$$-2\nu^{-1}\langle\mathbf{y}_c^k+\mathbf{x}_c^k-\mathbf{y}^*-\mathbf{x}^*,\mathbf{y}^k+\mathbf{x}^k-\mathbf{y}^*-\mathbf{x}^*\rangle$$

$$=\left(\frac{1}{\eta_y}-\gamma\right)\left\|\mathbf{y}^k-\mathbf{y}^*\right\|^2+\left(\frac{1}{\eta_x}-\gamma\right)\left\|\hat{\mathbf{x}}^k-\mathbf{x}^*\right\|^2-\left(\frac{1}{\eta_y}-\gamma\right)\left\|\mathbf{y}^{k+1}-\mathbf{y}^k\right\|^2$$

$$+2\gamma\left\|m^k\right\|_{\mathbf{P}}^2+2\left\|\Delta_x^k\right\|_{\mathbf{P}}\left\|m^k\right\|_{\mathbf{P}}+\eta_x\left\|\Delta_x^k\right\|_{\mathbf{P}}^2-2\langle\mathbf{z}^{k+1}-\mathbf{z}^*,\mathbf{y}^{k+1}-\mathbf{y}^*\rangle$$

$$-\gamma\left\|\mathbf{x}^k-\mathbf{x}^*\right\|^2+2\gamma\nu^2\left\|\mathbf{z}^k-\mathbf{z}^*\right\|^2+5\gamma\left\|\Delta^{k+1/2}-\mathbf{F}(\mathbf{z}^*)\right\|^2$$

$$-\frac{2\theta\nu^{-1}}{\tau}\langle\mathbf{y}_c^k+\mathbf{x}_c^k-\mathbf{y}^*-\mathbf{x}^*,(\mathbf{W}(k)\otimes\mathbf{I}_d)(\mathbf{y}_c^k+\mathbf{x}_c^k)\rangle$$

$$-\frac{2\nu^{-1}}{\tau}\langle\mathbf{y}_c^k+\mathbf{x}_c^k-\mathbf{y}^*-\mathbf{x}^*,\mathbf{y}_f^{k+1}+\mathbf{x}_f^{k+1}-\mathbf{y}_c^k-\mathbf{x}_c^k\rangle$$

$$-2\nu^{-1}\langle\mathbf{y}_c^k+\mathbf{x}_c^k-\mathbf{y}^*-\mathbf{x}^*,\mathbf{y}^k+\mathbf{x}^k-\mathbf{y}^*-\mathbf{x}^*\rangle$$

Using definitions (37), (38) we get $\mathbf{y}^*+\mathbf{x}^*\in\mathcal{L}$ and then

$$\frac{1}{\eta_y}\left\|\mathbf{y}^{k+1}-\mathbf{y}^*\right\|^2+\frac{1}{\eta_x}\left\|\hat{\mathbf{x}}^{k+1}-\mathbf{x}^*\right\|^2$$

$$\leq\left(\frac{1}{\eta_y}-\gamma\right)\left\|\mathbf{y}^k-\mathbf{y}^*\right\|^2+\left(\frac{1}{\eta_x}-\gamma\right)\left\|\hat{\mathbf{x}}^k-\mathbf{x}^*\right\|^2-\left(\frac{1}{\eta_y}-\gamma\right)\left\|\mathbf{y}^{k+1}-\mathbf{y}^k\right\|^2$$

$$+2\gamma\left\|m^k\right\|_{\mathbf{P}}^2+2\left\|\Delta_x^k\right\|_{\mathbf{P}}\left\|m^k\right\|_{\mathbf{P}}+\eta_x\left\|\Delta_x^k\right\|_{\mathbf{P}}^2-2\langle\mathbf{z}^{k+1}-\mathbf{z}^*,\mathbf{y}^{k+1}-\mathbf{y}^*\rangle$$

$$-\gamma\left\|\mathbf{x}^k-\mathbf{x}^*\right\|^2+2\gamma\nu^2\left\|\mathbf{z}^k-\mathbf{z}^*\right\|^2+5\gamma\left\|\Delta^{k+1/2}-\mathbf{F}(\mathbf{z}^*)\right\|^2$$

$$-\frac{2\theta\nu^{-1}}{\tau}\langle\mathbf{y}_c^k+\mathbf{x}_c^k,(\mathbf{W}(k)\otimes\mathbf{I}_d)(\mathbf{y}_c^k+\mathbf{x}_c^k)\rangle$$

$$-\frac{2\nu^{-1}}{\tau}\langle\mathbf{y}_c^k+\mathbf{x}_c^k-\mathbf{y}^*-\mathbf{x}^*,\mathbf{y}_f^{k+1}+\mathbf{x}_f^{k+1}-\mathbf{y}_c^k-\mathbf{x}_c^k\rangle$$

$$-2\nu^{-1}\langle\mathbf{y}_c^k+\mathbf{x}_c^k-\mathbf{y}^*-\mathbf{x}^*,\mathbf{y}^k+\mathbf{x}^k-\mathbf{y}^*-\mathbf{x}^*\rangle$$

$$= \left(\frac{1}{\eta_y} - \gamma\right)\left\|\mathbf{y}^k - \mathbf{y}^*\right\|^2 + \left(\frac{1}{\eta_x} - \gamma\right)\left\|\hat{\mathbf{x}}^k - \mathbf{x}^*\right\|^2 - \left(\frac{1}{\eta_y} - \gamma\right)\left\|\mathbf{y}^{k+1} - \mathbf{y}^k\right\|^2$$

$$+ 2\gamma\left\|m^k\right\|_{\mathbf{P}}^2 + 2\left\|\Delta_x^k\right\|_{\mathbf{P}}\left\|m^k\right\|_{\mathbf{P}} + \eta_x\left\|\Delta_x^k\right\|_{\mathbf{P}}^2 - 2\langle\mathbf{z}^{k+1} - \mathbf{z}^*, \mathbf{y}^{k+1} - \mathbf{y}^*\rangle$$

$$- \gamma\left\|\mathbf{x}^k - \mathbf{x}^*\right\|^2 + 2\gamma\nu^2\left\|\mathbf{z}^k - \mathbf{z}^*\right\|^2 + 5\gamma\left\|\Delta^{k+1/2} - \mathbf{F}(\mathbf{z}^*)\right\|^2$$

$$- \frac{2\theta\nu^{-1}}{\tau}\left\|\mathbf{y}_c^k + \mathbf{x}_c^k\right\|_{(\mathbf{W}(k)\otimes\mathbf{I}_d)}^2$$

$$- \frac{2\nu^{-1}}{\tau}\langle\mathbf{y}_c^k + \mathbf{x}_c^k - \mathbf{y}^* - \mathbf{x}^*, \mathbf{y}_f^{k+1} + \mathbf{x}_f^{k+1} - \mathbf{y}_c^k - \mathbf{x}_c^k\rangle$$

$$- 2\nu^{-1}\langle\mathbf{y}_c^k + \mathbf{x}_c^k - \mathbf{y}^* - \mathbf{x}^*, \mathbf{y}^k + \mathbf{x}^k - \mathbf{y}^* - \mathbf{x}^*\rangle$$

By parallelogram rule we obtain

$$\frac{1}{\eta_y}\left\|\mathbf{y}^{k+1} - \mathbf{y}^*\right\|^2 + \frac{1}{\eta_x}\left\|\hat{\mathbf{x}}^{k+1} - \mathbf{x}^*\right\|^2$$

$$\leq \left(\frac{1}{\eta_y} - \gamma\right)\left\|\mathbf{y}^k - \mathbf{y}^*\right\|^2 + \left(\frac{1}{\eta_x} - \gamma\right)\left\|\hat{\mathbf{x}}^k - \mathbf{x}^*\right\|^2 - \left(\frac{1}{\eta_y} - \gamma\right)\left\|\mathbf{y}^{k+1} - \mathbf{y}^k\right\|^2$$

$$+ 2\gamma\left\|m^k\right\|_{\mathbf{P}}^2 + 2\left\|\Delta_x^k\right\|_{\mathbf{P}}\left\|m^k\right\|_{\mathbf{P}} + \eta_x\left\|\Delta_x^k\right\|_{\mathbf{P}}^2 - 2\langle\mathbf{z}^{k+1} - \mathbf{z}^*, \mathbf{y}^{k+1} - \mathbf{y}^*\rangle$$

$$- \gamma\left\|\mathbf{x}^k - \mathbf{x}^*\right\|^2 + 2\gamma\nu^2\left\|\mathbf{z}^k - \mathbf{z}^*\right\|^2 + 5\gamma\left\|\Delta^{k+1/2} - \mathbf{F}(\mathbf{z}^*)\right\|^2$$

$$- \frac{\nu^{-1}}{\tau}\left\|\mathbf{y}_f^{k+1} + \mathbf{x}_f^{k+1} - \mathbf{y}^* - \mathbf{x}^*\right\|^2 + \frac{\nu^{-1}}{\tau}\left\|\mathbf{y}_c^k + \mathbf{x}_c^k - \mathbf{y}^* - \mathbf{x}^*\right\|^2$$

$$+ \frac{\nu^{-1}}{\tau}\left\|\mathbf{y}_f^{k+1} + \mathbf{x}_f^{k+1} - \mathbf{y}_c^k - \mathbf{x}_c^k\right\|^2$$

$$- \frac{2\theta\nu^{-1}}{\tau}\left\|\mathbf{y}_c^k + \mathbf{x}_c^k\right\|_{(\mathbf{W}(k)\otimes\mathbf{I}_d)}^2 - 2\nu^{-1}\langle\mathbf{y}_c^k + \mathbf{x}_c^k - \mathbf{y}^* - \mathbf{x}^*, \mathbf{y}^k + \mathbf{x}^k - \mathbf{y}^* - \mathbf{x}^*\rangle$$

$$\leq \left(\frac{1}{\eta_y} - \gamma\right)\left\|\mathbf{y}^k - \mathbf{y}^*\right\|^2 + \left(\frac{1}{\eta_x} - \gamma\right)\left\|\hat{\mathbf{x}}^k - \mathbf{x}^*\right\|^2 - \left(\frac{1}{\eta_y} - \gamma\right)\left\|\mathbf{y}^{k+1} - \mathbf{y}^k\right\|^2$$

$$+ 2\gamma\left\|m^k\right\|_{\mathbf{P}}^2 + 2\left\|\Delta_x^k\right\|_{\mathbf{P}}\left\|m^k\right\|_{\mathbf{P}} + \eta_x\left\|\Delta_x^k\right\|_{\mathbf{P}}^2 - 2\langle\mathbf{z}^{k+1} - \mathbf{z}^*, \mathbf{y}^{k+1} - \mathbf{y}^*\rangle$$

$$- \gamma\left\|\mathbf{x}^k - \mathbf{x}^*\right\|^2 + 2\gamma\nu^2\left\|\mathbf{z}^k - \mathbf{z}^*\right\|^2 + 5\gamma\left\|\Delta^{k+1/2} - \mathbf{F}(\mathbf{z}^*)\right\|^2$$

$$- \frac{\nu^{-1}}{\tau}\left\|\mathbf{y}_f^{k+1} + \mathbf{x}_f^{k+1} - \mathbf{y}^* - \mathbf{x}^*\right\|^2 + \frac{\nu^{-1}}{\tau}\left\|\mathbf{y}_c^k + \mathbf{x}_c^k - \mathbf{y}^* - \mathbf{x}^*\right\|^2$$

$$+ \frac{2\nu^{-1}}{\tau}\left\|\mathbf{y}_f^{k+1} - \mathbf{y}_c^k\right\|^2 + \frac{2\nu^{-1}}{\tau}\left\|\mathbf{x}_f^{k+1} - \mathbf{x}_c^k\right\|^2 - \frac{2\theta\nu^{-1}}{\tau}\left\|\mathbf{y}_c^k + \mathbf{x}_c^k\right\|_{(\mathbf{W}(k)\otimes\mathbf{I}_d)}^2$$

$$- 2\nu^{-1}\langle\mathbf{y}_c^k + \mathbf{x}_c^k - \mathbf{y}^* - \mathbf{x}^*, \mathbf{y}^k + \mathbf{x}^k - \mathbf{y}^* - \mathbf{x}^*\rangle$$

Again by expressions for $\mathbf{x}_f^{k+1}$ and $\mathbf{y}_f^{k+1}$

$$\frac{1}{\eta_y}\left\|\mathbf{y}^{k+1} - \mathbf{y}^*\right\|^2 + \frac{1}{\eta_x}\left\|\hat{\mathbf{x}}^{k+1} - \mathbf{x}^*\right\|^2$$

$$\leq \left(\frac{1}{\eta_y} - \gamma\right)\left\|\mathbf{y}^k - \mathbf{y}^*\right\|^2 + \left(\frac{1}{\eta_x} - \gamma\right)\left\|\hat{\mathbf{x}}^k - \mathbf{x}^*\right\|^2 - \left(\frac{1}{\eta_y} - \gamma\right)\left\|\mathbf{y}^{k+1} - \mathbf{y}^k\right\|^2$$

$$+ 2\gamma\left\|m^k\right\|_{\mathbf{P}}^2 + 2\left\|\Delta_x^k\right\|_{\mathbf{P}}\left\|m^k\right\|_{\mathbf{P}} + \eta_x\left\|\Delta_x^k\right\|_{\mathbf{P}}^2 - 2\langle x^{k+1} - \mathbf{x}^*, \mathbf{y}^{k+1} - \mathbf{y}^*\rangle$$

$$- \gamma\left\|\mathbf{x}^k - \mathbf{x}^*\right\|^2 + 2\gamma\nu^2\left\|\mathbf{z}^k - \mathbf{z}^*\right\|^2 + 5\gamma\left\|\Delta^{k+1/2} - \mathbf{F}(\mathbf{x}^*)\right\|^2$$

$$- \frac{\nu^{-1}}{\tau}\left\|\mathbf{y}_f^{k+1} + \mathbf{x}_f^{k+1} - \mathbf{y}^* - \mathbf{x}^*\right\|^2 + \frac{\nu^{-1}}{\tau}\left\|\mathbf{y}_c^k + \mathbf{x}_c^k - \mathbf{y}^* - \mathbf{x}^*\right\|^2$$

$$+ 2\nu^{-1}\tau\left\|\mathbf{y}^{k+1} - \mathbf{y}^k\right\|^2 + \frac{2\theta^2\nu^{-1}}{\tau}\left\|\mathbf{y}_c^k + \mathbf{x}_c^k\right\|_{(\mathbf{W}(k)\otimes\mathbf{I}_d)^2}^2 - \frac{2\theta\nu^{-1}}{\tau}\left\|\mathbf{y}_c^k + \mathbf{x}_c^k\right\|_{(\mathbf{W}(k)\otimes\mathbf{I}_d)}^2$$

$$- 2\nu^{-1}\langle\mathbf{y}_c^k + \mathbf{x}_c^k - \mathbf{y}^* - \mathbf{x}^*, \mathbf{y}^k + \mathbf{x}^k - \mathbf{y}^* - \mathbf{x}^*\rangle.$$

Using the contraction property of the gossip matrix and the definition of $\theta = \frac{1}{2}$ we get

$$\frac{1}{\eta_y}\left\|\mathbf{y}^{k+1} - \mathbf{y}^*\right\|^2 + \frac{1}{\eta_x}\left\|\hat{\mathbf{x}}^{k+1} - \mathbf{x}^*\right\|^2$$

$$\leq \left(\frac{1}{\eta_y} - \gamma\right)\left\|\mathbf{y}^k - \mathbf{y}^*\right\|^2 + \left(\frac{1}{\eta_x} - \gamma\right)\left\|\hat{\mathbf{x}}^k - \mathbf{x}^*\right\|^2 - \left(\frac{1}{\eta_y} - \gamma - 2\nu^{-1}\tau\right)\left\|\mathbf{y}^{k+1} - \mathbf{y}^k\right\|^2$$
$$+ 2\gamma\left\|m^k\right\|_{\mathbf{P}}^2 + 2\left\|\Delta_x^k\right\|_{\mathbf{P}}\left\|m^k\right\|_{\mathbf{P}} + \eta_x\left\|\Delta_x^k\right\|_{\mathbf{P}}^2 - \frac{\nu^{-1}}{2\tau\chi(T)}\left\|\mathbf{y}_c^k + \mathbf{x}_c^k\right\|_{\mathbf{P}}^2$$
$$- 2\langle\mathbf{z}^{k+1} - \mathbf{z}^*, \mathbf{y}^{k+1} - \mathbf{y}^*\rangle - \gamma\left\|\mathbf{x}^k - \mathbf{x}^*\right\|^2 + 2\gamma\nu^2\left\|\mathbf{z}^k - \mathbf{z}^*\right\|^2 + 5\gamma\left\|\Delta^{k+1/2} - \mathbf{F}(\mathbf{z}^*)\right\|^2$$
$$- \frac{\nu^{-1}}{\tau}\left\|\mathbf{y}_f^{k+1} + \mathbf{x}_f^{k+1} - \mathbf{y}^* - \mathbf{x}^*\right\|^2 + \frac{\nu^{-1}}{\tau}\left\|\mathbf{y}_c^k + \mathbf{x}_c^k - \mathbf{y}^* - \mathbf{x}^*\right\|^2$$
$$- 2\nu^{-1}\langle\mathbf{y}_c^k + \mathbf{x}_c^k - \mathbf{y}^* - \mathbf{x}^*, \mathbf{y}^k + \mathbf{x}^k - \mathbf{y}^* - \mathbf{x}^*\rangle.$$

By updates for $\mathbf{x}_c^k$ and $\mathbf{y}_c^k$ we get

$$\frac{1}{\eta_y}\left\|\mathbf{y}^{k+1} - \mathbf{y}^*\right\|^2 + \frac{1}{\eta_x}\left\|\hat{\mathbf{x}}^{k+1} - \mathbf{x}^*\right\|^2$$
$$\leq \left(\frac{1}{\eta_y} - \gamma\right)\left\|\mathbf{y}^k - \mathbf{y}^*\right\|^2 + \left(\frac{1}{\eta_x} - \gamma\right)\left\|\hat{\mathbf{x}}^k - \mathbf{x}^*\right\|^2 - \left(\frac{1}{\eta_y} - \gamma - 2\nu^{-1}\tau\right)\left\|\mathbf{y}^{k+1} - \mathbf{y}^k\right\|^2$$
$$+ 2\gamma\left\|m^k\right\|_{\mathbf{P}}^2 + 2\left\|\Delta_x^k\right\|_{\mathbf{P}}\left\|m^k\right\|_{\mathbf{P}} + \eta_x\left\|\Delta_x^k\right\|_{\mathbf{P}}^2 - \frac{\nu^{-1}}{2\tau\chi(T)}\left\|\mathbf{y}_c^k + \mathbf{x}_c^k\right\|_{\mathbf{P}}^2$$
$$- 2\langle\mathbf{z}^{k+1} - \mathbf{z}^*, \mathbf{y}^{k+1} - \mathbf{y}^*\rangle - \gamma\left\|\mathbf{x}^k - \mathbf{x}^*\right\|^2 + 2\gamma\nu^2\left\|\mathbf{z}^k - \mathbf{z}^*\right\|^2 + 5\gamma\left\|\Delta^{k+1/2} - \mathbf{F}(\mathbf{z}^*)\right\|^2$$
$$- \frac{\nu^{-1}}{\tau}\left\|\mathbf{y}_f^{k+1} + \mathbf{x}_f^{k+1} - \mathbf{y}^* - \mathbf{x}^*\right\|^2 + \frac{\nu^{-1}}{\tau}\left\|\mathbf{y}_c^k + \mathbf{x}_c^k - \mathbf{y}^* - \mathbf{x}^*\right\|^2$$
$$- 2\nu^{-1}\langle\mathbf{y}_c^k + \mathbf{x}_c^k - \mathbf{y}^* - \mathbf{x}^*, \mathbf{y}_c^k + \mathbf{x}_c^k - \mathbf{y}^* - \mathbf{x}^*\rangle$$
$$- 2\nu^{-1}\langle\mathbf{y}_c^k + \mathbf{x}_c^k - \mathbf{y}^* - \mathbf{x}^*, \mathbf{y}^k + \mathbf{x}^k - \mathbf{y}_c^k - \mathbf{x}_c^k\rangle$$
$$= \left(\frac{1}{\eta_y} - \gamma\right)\left\|\mathbf{y}^k - \mathbf{y}^*\right\|^2 + \left(\frac{1}{\eta_x} - \gamma\right)\left\|\hat{\mathbf{x}}^k - \mathbf{x}^*\right\|^2 - \left(\frac{1}{\eta_y} - \gamma - 2\nu^{-1}\tau\right)\left\|\mathbf{y}^{k+1} - \mathbf{y}^k\right\|^2$$
$$+ 2\gamma\left\|m^k\right\|_{\mathbf{P}}^2 + 2\left\|\Delta_x^k\right\|_{\mathbf{P}}\left\|m^k\right\|_{\mathbf{P}} + \eta_x\left\|\Delta_x^k\right\|_{\mathbf{P}}^2 - \frac{\nu^{-1}}{2\tau\chi(T)}\left\|\mathbf{y}_c^k + \mathbf{x}_c^k\right\|_{\mathbf{P}}^2$$
$$- 2\langle\mathbf{z}^{k+1} - \mathbf{z}^*, \mathbf{y}^{k+1} - \mathbf{y}^*\rangle - \gamma\left\|\mathbf{x}^k - \mathbf{x}^*\right\|^2 + 2\gamma\nu^2\left\|\mathbf{z}^k - \mathbf{z}^*\right\|^2 + 5\gamma\left\|\Delta^{k+1/2} - \mathbf{F}(\mathbf{z}^*)\right\|^2$$
$$- \frac{\nu^{-1}}{\tau}\left\|\mathbf{y}_f^{k+1} + \mathbf{x}_f^{k+1} - \mathbf{y}^* - \mathbf{x}^*\right\|^2 + \frac{\nu^{-1}}{\tau}\left\|\mathbf{y}_c^k + \mathbf{x}_c^k - \mathbf{y}^* - \mathbf{x}^*\right\|^2$$
$$- 2\nu^{-1}\left\|\mathbf{y}_c^k + \mathbf{x}_c^k - \mathbf{y}^* - \mathbf{x}^*\right\|^2$$
$$+ \frac{2\nu^{-1}(1 - \tau)}{\tau}\langle\mathbf{y}_c^k + \mathbf{x}_c^k - \mathbf{y}^* - \mathbf{x}^*, \mathbf{y}_f^k + \mathbf{x}_f^k - \mathbf{y}_c^k - \mathbf{x}_c^k\rangle.$$

Using parallelogram rule we obtain

$$\frac{1}{\eta_y}\left\|\mathbf{y}^{k+1} - \mathbf{y}^*\right\|^2 + \frac{1}{\eta_x}\left\|\hat{\mathbf{x}}^{k+1} - \mathbf{x}^*\right\|^2$$
$$\leq \left(\frac{1}{\eta_y} - \gamma\right)\left\|\mathbf{y}^k - \mathbf{y}^*\right\|^2 + \left(\frac{1}{\eta_x} - \gamma\right)\left\|\hat{\mathbf{x}}^k - \mathbf{x}^*\right\|^2 - \left(\frac{1}{\eta_y} - \gamma - 2\nu^{-1}\tau\right)\left\|\mathbf{y}^{k+1} - \mathbf{y}^k\right\|^2$$
$$+ 2\gamma\left\|m^k\right\|_{\mathbf{P}}^2 + 2\left\|\Delta_x^k\right\|_{\mathbf{P}}\left\|m^k\right\|_{\mathbf{P}} + \eta_x\left\|\Delta_x^k\right\|_{\mathbf{P}}^2 - \frac{\nu^{-1}}{2\tau\chi(T)}\left\|\mathbf{y}_c^k + \mathbf{x}_c^k\right\|_{\mathbf{P}}^2$$
$$- 2\langle\mathbf{z}^{k+1} - \mathbf{z}^*, \mathbf{y}^{k+1} - \mathbf{y}^*\rangle - \gamma\left\|\mathbf{x}^k - \mathbf{x}^*\right\|^2 + 2\gamma\nu^2\left\|\mathbf{z}^k - \mathbf{z}^*\right\|^2 + 5\gamma\left\|\Delta^{k+1/2} - \mathbf{F}(\mathbf{z}^*)\right\|^2$$
$$- \frac{\nu^{-1}}{\tau}\left\|\mathbf{y}_f^{k+1} + \mathbf{x}_f^{k+1} - \mathbf{y}^* - \mathbf{x}^*\right\|^2 + \frac{\nu^{-1}}{\tau}\left\|\mathbf{y}_c^k + \mathbf{x}_c^k - \mathbf{y}^* - \mathbf{x}^*\right\|^2$$
$$- 2\nu^{-1}\left\|\mathbf{y}_c^k + \mathbf{x}_c^k - \mathbf{y}^* - \mathbf{x}^*\right\|^2$$
$$+ \frac{\nu^{-1}(1 - \tau)}{\tau}\left\|\mathbf{y}_f^k + \mathbf{x}_f^k - \mathbf{y}^* - \mathbf{x}^*\right\|^2 - \frac{\nu^{-1}(1 - \tau)}{\tau}\left\|\mathbf{y}_c^k + \mathbf{x}_c^k - \mathbf{y}^* - \mathbf{x}^*\right\|^2$$
$$\leq \left(\frac{1}{\eta_y} - \gamma\right)\left\|\mathbf{y}^k - \mathbf{y}^*\right\|^2 + \left(\frac{1}{\eta_x} - \gamma\right)\left\|\hat{\mathbf{x}}^k - \mathbf{x}^*\right\|^2 - \left(\frac{1}{\eta_y} - \gamma - 2\nu^{-1}\tau\right)\left\|\mathbf{y}^{k+1} - \mathbf{y}^k\right\|^2$$
$$+ 2\gamma\left\|m^k\right\|_{\mathbf{P}}^2 + 2\left\|\Delta_x^k\right\|_{\mathbf{P}}\left\|m^k\right\|_{\mathbf{P}} + \eta_x\left\|\Delta_x^k\right\|_{\mathbf{P}}^2 - \frac{\nu^{-1}}{2\tau\chi(T)}\left\|\mathbf{y}_c^k + \mathbf{x}_c^k\right\|_{\mathbf{P}}^2$$

$$- 2\langle \mathbf{z}^{k+1} - \mathbf{z}^*, \mathbf{y}^{k+1} - \mathbf{y}^* \rangle - \gamma \left\| \mathbf{x}^k - \mathbf{x}^* \right\|^2 + 2\gamma\nu^2 \left\| \mathbf{z}^k - \mathbf{z}^* \right\|^2 + 5\gamma \left\| \Delta^{k+1/2} - \mathbf{F}(\mathbf{z}^*) \right\|^2$$

$$- \frac{\nu^{-1}}{\tau} \left\| \mathbf{y}_f^{k+1} + \mathbf{x}_f^{k+1} - \mathbf{y}^* - \mathbf{x}^* \right\|^2 + \frac{\nu^{-1}(1-\tau)}{\tau} \left\| \mathbf{y}_f^k + \mathbf{x}_f^k - \mathbf{y}^* - \mathbf{x}^* \right\|^2.$$

Using Young's inequality we get

$$\frac{1}{\eta_y} \left\| \mathbf{y}^{k+1} - \mathbf{y}^* \right\|^2 + \frac{1}{\eta_x} \left\| \hat{\mathbf{x}}^{k+1} - \mathbf{x}^* \right\|^2$$

$$\leq \left( \frac{1}{\eta_y} - \gamma \right) \left\| \mathbf{y}^k - \mathbf{y}^* \right\|^2 + \left( \frac{1}{\eta_x} - \gamma \right) \left\| \hat{\mathbf{x}}^k - \mathbf{x}^* \right\|^2 - \left( \frac{1}{\eta_y} - \gamma - 2\nu^{-1}\tau \right) \left\| \mathbf{y}^{k+1} - \mathbf{y}^k \right\|^2$$

$$+ 2\gamma \left\| m^k \right\|_{\mathbf{P}}^2 + 4\eta_x\chi(T) \left\| \Delta_x^k \right\|_{\mathbf{P}}^2 + (4\eta_x\chi(T))^{-1} \left\| m^k \right\|_{\mathbf{P}}^2 + \eta_x \left\| \Delta_x^k \right\|_{\mathbf{P}}^2$$

$$- \frac{\nu^{-1}}{2\tau\chi(T)} \left\| \mathbf{y}_c^k + \mathbf{x}_c^k \right\|_{\mathbf{P}}^2 - 2\langle \mathbf{z}^{k+1} - \mathbf{z}^*, \mathbf{y}^{k+1} - \mathbf{y}^* \rangle - \gamma \left\| \mathbf{x}^k - \mathbf{x}^* \right\|^2$$

$$+ 2\gamma\nu^2 \left\| \mathbf{z}^k - \mathbf{z}^* \right\|^2 + 5\gamma \left\| \Delta^{k+1/2} - \mathbf{F}(\mathbf{z}^*) \right\|^2 - \frac{\nu^{-1}}{\tau} \left\| \mathbf{y}_f^{k+1} + \mathbf{x}_f^{k+1} - \mathbf{y}^* - \mathbf{x}^* \right\|^2$$

$$+ \frac{\nu^{-1}(1-\tau)}{\tau} \left\| \mathbf{y}_f^k + \mathbf{x}_f^k - \mathbf{y}^* - \mathbf{x}^* \right\|^2$$

$$= \left( \frac{1}{\eta_y} - \gamma \right) \left\| \mathbf{y}^k - \mathbf{y}^* \right\|^2 + \left( \frac{1}{\eta_x} - \gamma \right) \left\| \hat{\mathbf{x}}^k - \mathbf{x}^* \right\|^2 - \left( \frac{1}{\eta_y} - \gamma - 2\nu^{-1}\tau \right) \left\| \mathbf{y}^{k+1} - \mathbf{y}^k \right\|^2$$

$$+ 2\gamma \left\| m^k \right\|_{\mathbf{P}}^2 + 4\eta_x\chi(T) \left\| \Delta_x^k \right\|_{\mathbf{P}}^2 + (4\eta_x\chi(T))^{-1} \left\| m^k \right\|_{\mathbf{P}}^2 + \eta_x \left\| \Delta_x^k \right\|_{\mathbf{P}}^2$$

$$- \frac{\nu^{-1}}{2\tau\chi(T)} \left\| \mathbf{y}_c^k + \mathbf{x}_c^k \right\|_{\mathbf{P}}^2 - 2\langle \mathbf{z}^{k+1} - \mathbf{z}^*, \mathbf{y}^{k+1} - \mathbf{y}^* \rangle - \gamma \left\| \mathbf{x}^k - \mathbf{x}^* \right\|^2$$

$$+ 2\gamma\nu^2 \left\| \mathbf{z}^k - \mathbf{z}^* \right\|^2 + 5\gamma \left\| \Delta^{k+1/2} - \mathbf{F}(\mathbf{z}^*) \right\|^2 - \frac{\nu^{-1}}{\tau} \left\| \mathbf{y}_f^{k+1} + \mathbf{x}_f^{k+1} - \mathbf{y}^* - \mathbf{x}^* \right\|^2$$

$$+ \frac{\nu^{-1}(1-\tau)}{\tau} \left\| \mathbf{y}_f^k + \mathbf{x}_f^k - \mathbf{y}^* - \mathbf{x}^* \right\|^2.$$

By the assumption on $\eta_x \leq \frac{1}{8\chi(T)\gamma}$ we get that $2\gamma \leq \frac{1}{4\eta_x}$ and

$$\frac{1}{\eta_y} \left\| \mathbf{y}^{k+1} - \mathbf{y}^* \right\|^2 + \frac{1}{\eta_x} \left\| \hat{\mathbf{x}}^{k+1} - \mathbf{x}^* \right\|^2$$

$$\leq \left( \frac{1}{\eta_y} - \gamma \right) \left\| \mathbf{y}^k - \mathbf{y}^* \right\|^2 + \left( \frac{1}{\eta_x} - \gamma \right) \left\| \hat{\mathbf{x}}^k - \mathbf{x}^* \right\|^2 - \left( \frac{1}{\eta_y} - \gamma - 2\nu^{-1}\tau \right) \left\| \mathbf{y}^{k+1} - \mathbf{y}^k \right\|^2$$

$$+ 4\eta_x\chi(T) \left\| \Delta_x^k \right\|_{\mathbf{P}}^2 + (2\eta_x\chi(T))^{-1} \left\| m^k \right\|_{\mathbf{P}}^2 + \eta_x \left\| \Delta_x^k \right\|_{\mathbf{P}}^2 - \frac{\nu^{-1}}{2\tau\chi(T)} \left\| \mathbf{y}_c^k + \mathbf{x}_c^k \right\|_{\mathbf{P}}^2$$

$$- 2\langle \mathbf{z}^{k+1} - \mathbf{z}^*, \mathbf{y}^{k+1} - \mathbf{y}^* \rangle - \gamma \left\| \mathbf{x}^k - \mathbf{x}^* \right\|^2 + 2\gamma\nu^2 \left\| \mathbf{z}^k - \mathbf{z}^* \right\|^2$$

$$+ 5\gamma \left\| \Delta^{k+1/2} - \mathbf{F}(\mathbf{z}^*) \right\|^2 - \frac{\nu^{-1}}{\tau} \left\| \mathbf{y}_f^{k+1} + \mathbf{x}_f^{k+1} - \mathbf{y}^* - \mathbf{x}^* \right\|^2$$

$$+ \frac{\nu^{-1}(1-\tau)}{\tau} \left\| \mathbf{y}_f^k + \mathbf{x}_f^k - \mathbf{y}^* - \mathbf{x}^* \right\|^2.$$

Using (39) we get

$$\frac{1}{\eta_y} \left\| \mathbf{y}^{k+1} - \mathbf{y}^* \right\|^2 + \frac{1}{\eta_x} \left\| \hat{\mathbf{x}}^{k+1} - \mathbf{x}^* \right\|^2$$

$$\leq \left( \frac{1}{\eta_y} - \gamma \right) \left\| \mathbf{y}^k - \mathbf{y}^* \right\|^2 + \left( \frac{1}{\eta_x} - \gamma \right) \left\| \hat{\mathbf{x}}^k - \mathbf{x}^* \right\|^2 - \left( \frac{1}{\eta_y} - \gamma - 2\nu^{-1}\tau \right) \left\| \mathbf{y}^{k+1} - \mathbf{y}^k \right\|^2$$

$$+ 8\eta_x\chi(T) \left\| \Delta_x^k \right\|_{\mathbf{P}}^2 + (1 - (4\chi(T))^{-1})\frac{2}{\eta_x} \left\| m^k \right\|_{\mathbf{P}}^2 - \frac{2}{\eta_x} \left\| m^{k+1} \right\|_{\mathbf{P}}^2 + \eta_x \left\| \Delta_x^k \right\|_{\mathbf{P}}^2$$

$$- \frac{\nu^{-1}}{2\tau\chi(T)} \left\| \mathbf{y}_c^k + \mathbf{x}_c^k \right\|_{\mathbf{P}}^2 - 2\langle \mathbf{z}^{k+1} - \mathbf{z}^*, \mathbf{y}^{k+1} - \mathbf{y}^* \rangle - \gamma \left\| \mathbf{x}^k - \mathbf{x}^* \right\|^2 + 2\gamma\nu^2 \left\| \mathbf{z}^k - \mathbf{z}^* \right\|^2$$

$$+ 5\gamma \left\| \Delta^{k+1/2} - \mathbf{F}(\mathbf{z}^*) \right\|^2 - \frac{\nu^{-1}}{\tau} \left\| \mathbf{y}_f^{k+1} + \mathbf{x}_f^{k+1} - \mathbf{y}^* - \mathbf{x}^* \right\|^2$$

$$+ \frac{\nu^{-1}(1-\tau)}{\tau} \left\| \mathbf{y}_f^k + \mathbf{x}_f^k - \mathbf{y}^* - \mathbf{x}^* \right\|^2.$$

Using the definition of $\Delta_x^k$ with definition of $\beta = 5\gamma$ we get

$$\frac{1}{\eta_y}\left\|\mathbf{y}^{k+1} - \mathbf{y}^*\right\|^2 + \frac{1}{\eta_x}\left\|\hat{\mathbf{x}}^{k+1} - \mathbf{x}^*\right\|^2$$

$$\leq \left(\frac{1}{\eta_y} - \gamma\right)\left\|\mathbf{y}^k - \mathbf{y}^*\right\|^2 + \left(\frac{1}{\eta_x} - \gamma\right)\left\|\hat{\mathbf{x}}^k - \mathbf{x}^*\right\|^2 - \left(\frac{1}{\eta_y} - \gamma - 2\nu^{-1}\tau\right)\left\|\mathbf{y}^{k+1} - \mathbf{y}^k\right\|^2$$

$$+ 9\eta_x\chi(T)\left\|\nu^{-1}(\mathbf{y}_c^k + \mathbf{x}_c^k) + 5\gamma(\mathbf{x}^k + \Delta^{k+1/2})\right\|_{\mathbf{P}}^2 + (1 - (4\chi(T))^{-1})\frac{2}{\eta_x}\left\|m^k\right\|_{\mathbf{P}}^2$$

$$- \frac{2}{\eta_x}\left\|m^{k+1}\right\|_{\mathbf{P}}^2 - \frac{\nu^{-1}}{2\tau\chi(T)}\left\|\mathbf{y}_c^k + \mathbf{x}_c^k\right\|_{\mathbf{P}}^2 - 2\langle\mathbf{z}^{k+1} - \mathbf{z}^*, \mathbf{y}^{k+1} - \mathbf{y}^*\rangle - \gamma\left\|\mathbf{x}^k - \mathbf{x}^*\right\|^2$$

$$+ 2\gamma\nu^2\left\|\mathbf{z}^k - \mathbf{z}^*\right\|^2 + 5\gamma\left\|\Delta^{k+1/2} - \mathbf{F}(\mathbf{z}^*)\right\|^2 - \frac{\nu^{-1}}{\tau}\left\|\mathbf{y}_f^{k+1} + \mathbf{x}_f^{k+1} - \mathbf{y}^* - \mathbf{x}^*\right\|^2$$

$$+ \frac{\nu^{-1}(1-\tau)}{\tau}\left\|\mathbf{y}_f^k + \mathbf{x}_f^k - \mathbf{y}^* - \mathbf{x}^*\right\|^2.$$

By definition $\mathbf{P}\mathbf{F}(\mathbf{z}^*) + \mathbf{x}^* = 0$ we obtain

$$\frac{1}{\eta_y}\left\|\mathbf{y}^{k+1} - \mathbf{y}^*\right\|^2 + \frac{1}{\eta_x}\left\|\hat{\mathbf{x}}^{k+1} - \mathbf{x}^*\right\|^2$$

$$\leq \left(\frac{1}{\eta_y} - \gamma\right)\left\|\mathbf{y}^k - \mathbf{y}^*\right\|^2 + \left(\frac{1}{\eta_x} - \gamma\right)\left\|\hat{\mathbf{x}}^k - \mathbf{x}^*\right\|^2 - \left(\frac{1}{\eta_y} - \gamma - 2\nu^{-1}\tau\right)\left\|\mathbf{y}^{k+1} - \mathbf{y}^k\right\|^2$$

$$+ 18\eta_x\nu^{-2}\chi(T)\left\|\mathbf{y}_c^k + \mathbf{x}_c^k\right\|_{\mathbf{P}}^2 + 450\eta_x\chi(T)\gamma^2\left\|\mathbf{x}^k - \mathbf{x}^* - \mathbf{P}\mathbf{F}(\mathbf{z}^*) + \Delta^{k+1/2}\right\|_{\mathbf{P}}^2$$

$$+ (1 - (4\chi(T))^{-1})\frac{2}{\eta_x}\left\|m^k\right\|_{\mathbf{P}}^2 - \frac{2}{\eta_x}\left\|m^{k+1}\right\|_{\mathbf{P}}^2 - \frac{\nu^{-1}}{2\tau\chi(T)}\left\|\mathbf{y}_c^k + \mathbf{x}_c^k\right\|_{\mathbf{P}}^2$$

$$- 2\langle\mathbf{z}^{k+1} - \mathbf{z}^*, \mathbf{y}^{k+1} - \mathbf{y}^*\rangle - \gamma\left\|\mathbf{x}^k - \mathbf{x}^*\right\|^2 + 2\gamma\nu^2\left\|\mathbf{z}^k - \mathbf{z}^*\right\|^2 + 5\gamma\left\|\Delta^{k+1/2} - \mathbf{F}(\mathbf{z}^*)\right\|^2$$

$$- \frac{\nu^{-1}}{\tau}\left\|\mathbf{y}_f^{k+1} + \mathbf{x}_f^{k+1} - \mathbf{y}^* - \mathbf{x}^*\right\|^2 + \frac{\nu^{-1}(1-\tau)}{\tau}\left\|\mathbf{y}_f^k + \mathbf{x}_f^k - \mathbf{y}^* - \mathbf{x}^*\right\|^2$$

$$\leq \left(\frac{1}{\eta_y} - \gamma\right)\left\|\mathbf{y}^k - \mathbf{y}^*\right\|^2 + \left(\frac{1}{\eta_x} - \gamma\right)\left\|\hat{\mathbf{x}}^k - \mathbf{x}^*\right\|^2 - \left(\frac{1}{\eta_y} - \gamma - 2\nu^{-1}\tau\right)\left\|\mathbf{y}^{k+1} - \mathbf{y}^k\right\|^2$$

$$+ 18\eta_x\nu^{-2}\chi(T)\left\|\mathbf{y}_c^k + \mathbf{x}_c^k\right\|_{\mathbf{P}}^2 + 900\eta_x\chi(T)\gamma^2\left\|\mathbf{x}^k - \mathbf{x}^*\right\|_{\mathbf{P}}^2$$

$$+ 900\eta_x\chi(T)\gamma^2\left\|\Delta^{k+1/2} - \mathbf{P}\mathbf{F}(\mathbf{z}^*)\right\|_{\mathbf{P}}^2 + (1 - (4\chi(T))^{-1})\frac{2}{\eta_x}\left\|m^k\right\|_{\mathbf{P}}^2$$

$$- \frac{2}{\eta_x}\left\|m^{k+1}\right\|_{\mathbf{P}}^2 - \frac{\nu^{-1}}{2\tau\chi(T)}\left\|\mathbf{y}_c^k + \mathbf{x}_c^k\right\|_{\mathbf{P}}^2 - 2\langle\mathbf{z}^{k+1} - \mathbf{z}^*, \mathbf{y}^{k+1} - \mathbf{y}^*\rangle - \gamma\left\|\mathbf{x}^k - \mathbf{x}^*\right\|^2$$

$$+ 2\gamma\nu^2\left\|\mathbf{z}^k - \mathbf{z}^*\right\|^2 + 5\gamma\left\|\Delta^{k+1/2} - \mathbf{F}(\mathbf{z}^*)\right\|^2$$

$$- \frac{\nu^{-1}}{\tau}\left\|\mathbf{y}_f^{k+1} + \mathbf{x}_f^{k+1} - \mathbf{y}^* - \mathbf{x}^*\right\|^2 + \frac{\nu^{-1}(1-\tau)}{\tau}\left\|\mathbf{y}_f^k + \mathbf{x}_f^k - \mathbf{y}^* - \mathbf{x}^*\right\|^2.$$

With the assumption on $\eta_x \leq \frac{1}{900\chi(T)\gamma}; \frac{\nu}{36\tau\chi^2(T)}$ we get

$$\frac{1}{\eta_y}\left\|\mathbf{y}^{k+1} - \mathbf{y}^*\right\|^2 + \frac{1}{\eta_x}\left\|\hat{\mathbf{x}}^{k+1} - \mathbf{x}^*\right\|^2$$

$$\leq \left(\frac{1}{\eta_y} - \gamma\right)\left\|\mathbf{y}^k - \mathbf{y}^*\right\|^2 + \left(\frac{1}{\eta_x} - \gamma\right)\left\|\hat{\mathbf{x}}^k - \mathbf{x}^*\right\|^2 - \left(\frac{1}{\eta_y} - \gamma - 2\nu^{-1}\tau\right)\left\|\mathbf{y}^{k+1} - \mathbf{y}^k\right\|^2$$

$$+ \frac{\nu^{-1}}{2\tau\chi(T)}\left\|\mathbf{y}_c^k + \mathbf{x}_c^k\right\|_{\mathbf{P}}^2 + \gamma\left\|\mathbf{x}^k - \mathbf{x}^*\right\|_{\mathbf{P}}^2 + \gamma\left\|\Delta^{k+1/2} - \mathbf{P}\mathbf{F}(\mathbf{z}^*)\right\|_{\mathbf{P}}^2$$

$$+ (1 - (4\chi(T))^{-1})\frac{2}{\eta_x}\left\|m^k\right\|_{\mathbf{P}}^2 - \frac{2}{\eta_x}\left\|m^{k+1}\right\|_{\mathbf{P}}^2 - \frac{\nu^{-1}}{2\tau\chi(T)}\left\|\mathbf{y}_c^k + \mathbf{x}_c^k\right\|_{\mathbf{P}}^2$$

$$- 2\langle\mathbf{z}^{k+1} - \mathbf{z}^*, \mathbf{y}^{k+1} - \mathbf{y}^*\rangle - \gamma\left\|\mathbf{x}^k - \mathbf{x}^*\right\|^2 + 2\gamma\nu^2\left\|\mathbf{z}^k - \mathbf{z}^*\right\|^2 + 5\gamma\left\|\Delta^{k+1/2} - \mathbf{F}(\mathbf{z}^*)\right\|^2$$

$$- \frac{\nu^{-1}}{\tau}\left\|\mathbf{y}_f^{k+1} + \mathbf{x}_f^{k+1} - \mathbf{y}^* - \mathbf{x}^*\right\|^2 + \frac{\nu^{-1}(1-\tau)}{\tau}\left\|\mathbf{y}_f^k + \mathbf{x}_f^k - \mathbf{y}^* - \mathbf{x}^*\right\|^2.$$

With property of the projector: $\mathbf{P}\mathbf{P} = \mathbf{P}$, we get

$$\frac{1}{\eta_y}\left\|\mathbf{y}^{k+1} - \mathbf{y}^*\right\|^2 + \frac{1}{\eta_x}\left\|\hat{\mathbf{x}}^{k+1} - \mathbf{x}^*\right\|^2$$

$$\leq \left(\frac{1}{\eta_y} - \gamma\right) \left\|\mathbf{y}^k - \mathbf{y}^*\right\|^2 + \left(\frac{1}{\eta_x} - \gamma\right) \left\|\hat{\mathbf{x}}^k - \mathbf{x}^*\right\|^2 - \left(\frac{1}{\eta_y} - \gamma - 2\nu^{-1}\tau\right) \left\|\mathbf{y}^{k+1} - \mathbf{y}^k\right\|^2$$

$$+ (1 - (4\chi(T))^{-1})\frac{2}{\eta_x} \left\|m^k\right\|_{\mathbf{P}}^2 - \frac{2}{\eta_x} \left\|m^{k+1}\right\|_{\mathbf{P}}^2 - 2\langle \mathbf{z}^{k+1} - \mathbf{z}^*, \mathbf{y}^{k+1} - \mathbf{y}^* \rangle$$

$$+ 2\gamma\nu^2 \left\|\mathbf{z}^k - \mathbf{z}^*\right\|^2 + 6\gamma \left\|\Delta^{k+1/2} - \mathbf{F}(\mathbf{z}^*)\right\|^2$$

$$- \frac{\nu^{-1}}{\tau} \left\|\mathbf{y}_f^{k+1} + \mathbf{x}_f^{k+1} - \mathbf{y}^* - \mathbf{x}^*\right\|^2 + \frac{\nu^{-1}(1-\tau)}{\tau} \left\|\mathbf{y}_f^k + \mathbf{x}_f^k - \mathbf{y}^* - \mathbf{x}^*\right\|^2 .$$

Taking full expectation and using (33) we have

$$\frac{1}{\eta_y}\mathbb{E}\left[\left\|\mathbf{y}^{k+1} - \mathbf{y}^*\right\|^2\right] + \frac{1}{\eta_x}\mathbb{E}\left[\left\|\hat{\mathbf{x}}^{k+1} - \mathbf{x}^*\right\|^2\right]$$

$$\leq \left(\frac{1}{\eta_y} - \gamma\right) \mathbb{E}\left[\left\|\mathbf{y}^k - \mathbf{y}^*\right\|^2\right] + \left(\frac{1}{\eta_x} - \gamma\right) \mathbb{E}\left[\left\|\hat{\mathbf{x}}^k - \mathbf{x}^*\right\|^2\right]$$

$$- \left(\frac{1}{\eta_y} - \gamma - 2\nu^{-1}\tau\right) \mathbb{E}\left[\left\|\mathbf{y}^{k+1} - \mathbf{y}^k\right\|^2\right] + (1 - (4\chi(T))^{-1})\frac{2}{\eta_x}\mathbb{E}\left[\left\|m^k\right\|_{\mathbf{P}}^2\right]$$

$$- \frac{2}{\eta_x}\mathbb{E}\left[\left\|m^{k+1}\right\|_{\mathbf{P}}^2\right] - 2\mathbb{E}\left[\langle \mathbf{z}^{k+1} - \mathbf{z}^*, \mathbf{y}^{k+1} - \mathbf{y}^* \rangle\right] + 2\gamma\nu^2 \mathbb{E}\left[\left\|\mathbf{z}^k - \mathbf{z}^*\right\|^2\right]$$

$$+ 12\gamma L^2 \mathbb{E}\left[\left\|\mathbf{z}^{k+1} - \mathbf{z}^*\right\|^2\right] + \frac{12\gamma\overline{L}^2}{b}\mathbb{E}\left[\left\|\mathbf{z}^{k+1} - \mathbf{w}^k\right\|^2\right]$$

$$- \frac{\nu^{-1}}{\tau}\mathbb{E}\left[\left\|\mathbf{y}_f^{k+1} + \mathbf{x}_f^{k+1} - \mathbf{y}^* - \mathbf{x}^*\right\|^2\right] + \frac{\nu^{-1}(1-\tau)}{\tau}\mathbb{E}\left[\left\|\mathbf{y}_f^k + \mathbf{x}_f^k - \mathbf{y}^* - \mathbf{x}^*\right\|^2\right] .$$

Using the definition of $\eta_y$ we get that $\gamma \leq \frac{1}{4\eta_y}$ and $2\nu^{-1}\tau \leq \frac{1}{4\eta_y}$

$$\frac{1}{\eta_y}\mathbb{E}\left[\left\|\mathbf{y}^{k+1} - \mathbf{y}^*\right\|^2\right] + \frac{1}{\eta_x}\mathbb{E}\left[\left\|\hat{\mathbf{x}}^{k+1} - \mathbf{x}^*\right\|^2\right]$$

$$\leq \left(\frac{1}{\eta_y} - \gamma\right) \mathbb{E}\left[\left\|\mathbf{y}^k - \mathbf{y}^*\right\|^2\right] + \left(\frac{1}{\eta_x} - \gamma\right) \mathbb{E}\left[\left\|\hat{\mathbf{x}}^k - \mathbf{x}^*\right\|^2\right] - \frac{1}{2\eta_y}\mathbb{E}\left[\left\|\mathbf{y}^{k+1} - \mathbf{y}^k\right\|^2\right]$$

$$+ (1 - (4\chi(T))^{-1})\frac{2}{\eta_x}\mathbb{E}\left[\left\|m^k\right\|_{\mathbf{P}}^2\right] - \frac{2}{\eta_x}\mathbb{E}\left[\left\|m^{k+1}\right\|_{\mathbf{P}}^2\right] - 2\mathbb{E}\left[\langle \mathbf{z}^{k+1} - \mathbf{z}^*, \mathbf{y}^{k+1} - \mathbf{y}^* \rangle\right]$$

$$+ 2\gamma\nu^2 \mathbb{E}\left[\left\|\mathbf{z}^k - \mathbf{z}^*\right\|^2\right] + 12\gamma L^2 \mathbb{E}\left[\left\|\mathbf{z}^{k+1} - \mathbf{z}^*\right\|^2\right] + \frac{12\gamma\overline{L}^2}{b}\mathbb{E}\left[\left\|\mathbf{z}^{k+1} - \mathbf{w}^k\right\|^2\right]$$

$$- \frac{\nu^{-1}}{\tau}\mathbb{E}\left[\left\|\mathbf{y}_f^{k+1} + \mathbf{x}_f^{k+1} - \mathbf{y}^* - \mathbf{x}^*\right\|^2\right] + \frac{\nu^{-1}(1-\tau)}{\tau}\mathbb{E}\left[\left\|\mathbf{y}_f^k + \mathbf{x}_f^k - \mathbf{y}^* - \mathbf{x}^*\right\|^2\right] .$$

**Part 4.** After combining parts 1 and 3 of this proof we get

$$\frac{1}{\eta_y}\mathbb{E}\left[\left\|\mathbf{y}^{k+1} - \mathbf{y}^*\right\|^2\right] + \frac{1}{\eta_x}\mathbb{E}\left[\left\|\hat{\mathbf{x}}^{k+1} - \mathbf{x}^*\right\|^2\right] + \frac{1}{\eta_z}\mathbb{E}\left[\left\|\mathbf{z}^{k+1} - \mathbf{z}^*\right\|^2\right] + (2\mu - \nu)\mathbb{E}\left[\left\|\mathbf{z}^{k+1} - \mathbf{z}^*\right\|^2\right]$$

$$+ \frac{1}{4\eta_z}\mathbb{E}\left[\left\|\mathbf{z}^{k+1} - \mathbf{z}^k\right\|^2\right] - 2\mathbb{E}\left[\langle \mathbf{F}(\mathbf{z}^{k+1}) - \mathbf{F}(\mathbf{z}^k) - (\mathbf{y}^{k+1} - \mathbf{y}^k), \mathbf{z}^{k+1} - \mathbf{z}^* \rangle\right]$$

$$\leq \left(\frac{1}{\eta_y} - \gamma\right) \mathbb{E}\left[\left\|\mathbf{y}^k - \mathbf{y}^*\right\|^2\right] + \left(\frac{1}{\eta_x} - \gamma\right) \mathbb{E}\left[\left\|\hat{\mathbf{x}}^k - \mathbf{x}^*\right\|^2\right] - \frac{1}{2\eta_y}\mathbb{E}\left[\left\|\mathbf{y}^{k+1} - \mathbf{y}^k\right\|^2\right]$$

$$+ (1 - (4\chi(T))^{-1})\frac{2}{\eta_x}\mathbb{E}\left[\left\|m^k\right\|_{\mathbf{P}}^2\right] - \frac{2}{\eta_x}\mathbb{E}\left[\left\|m^{k+1}\right\|_{\mathbf{P}}^2\right] - 2\mathbb{E}\left[\langle \mathbf{z}^{k+1} - \mathbf{z}^*, \mathbf{y}^{k+1} - \mathbf{y}^* \rangle\right]$$

$$+ 2\gamma\nu^2 \mathbb{E}\left[\left\|\mathbf{z}^k - \mathbf{z}^*\right\|^2\right] + 12\gamma L^2 \mathbb{E}\left[\left\|\mathbf{z}^{k+1} - \mathbf{z}^*\right\|^2\right] + \frac{12\gamma\overline{L}^2}{b}\mathbb{E}\left[\left\|\mathbf{z}^{k+1} - \mathbf{w}^k\right\|^2\right]$$

$$- \frac{\nu^{-1}}{\tau}\mathbb{E}\left[\left\|\mathbf{y}_f^{k+1} + \mathbf{x}_f^{k+1} - \mathbf{y}^* - \mathbf{x}^*\right\|^2\right] + \frac{\nu^{-1}(1-\tau)}{\tau}\mathbb{E}\left[\left\|\mathbf{y}_f^k + \mathbf{x}_f^k - \mathbf{y}^* - \mathbf{x}^*\right\|^2\right] .$$

$$+ \frac{1}{\eta_z}\mathbb{E}\left[\left\|\mathbf{z}^k - \mathbf{z}^*\right\|^2\right] - \left(\frac{\omega}{\eta_z} - \nu\right) \mathbb{E}\left[\left\|\mathbf{z}^k - \mathbf{z}^*\right\|^2\right]$$

$$+ \alpha \cdot \frac{1}{4\eta_z}\mathbb{E}\left[\left\|\mathbf{z}^k - \mathbf{z}^{k-1}\right\|^2\right] - \alpha \cdot 2\mathbb{E}\left[\langle \mathbf{F}(\mathbf{z}^k) - \mathbf{F}(\mathbf{z}^{k-1}) - (\mathbf{y}^k - \mathbf{y}^{k-1}), \mathbf{z}^k - \mathbf{z}^* \rangle\right]$$

$$+ \frac{\omega}{\eta_z}\mathbb{E}\left[\left\|\mathbf{w}^k - \mathbf{z}^*\right\|^2\right] - \frac{\omega}{\eta_z}\mathbb{E}\left[\left\|\mathbf{z}^{k+1} - \mathbf{w}^k\right\|^2\right] + 2\mathbb{E}\left[\langle \mathbf{y}^{k+1} - \mathbf{y}^*, \mathbf{z}^{k+1} - \mathbf{z}^* \rangle\right]$$

$$+ \frac{\alpha}{2\eta_y}\mathbb{E}\left[\left\|\mathbf{y}^k - \mathbf{y}^{k-1}\right\|^2\right] + \frac{4\eta_z\overline{L}^2}{b}\mathbb{E}\left[\left\|\mathbf{z}^k - \mathbf{w}^{k-1}\right\|^2\right].$$

Small rearrangement gives

$$\frac{1}{\eta_y}\mathbb{E}\left[\left\|\mathbf{y}^{k+1} - \mathbf{y}^*\right\|^2\right] + \frac{1}{\eta_x}\mathbb{E}\left[\left\|\hat{\mathbf{x}}^{k+1} - \mathbf{x}^*\right\|^2\right] + \frac{1}{2\eta_y}\mathbb{E}\left[\left\|\mathbf{y}^{k+1} - \mathbf{y}^k\right\|^2\right]$$

$$+ \frac{1}{4\eta_z}\mathbb{E}\left[\left\|\mathbf{z}^{k+1} - \mathbf{z}^k\right\|^2\right] - 2\mathbb{E}\left[\langle\mathbf{F}(\mathbf{z}^{k+1}) - \mathbf{F}(\mathbf{z}^k) - (\mathbf{y}^{k+1} - \mathbf{y}^k), \mathbf{z}^{k+1} - \mathbf{z}^*\rangle\right]$$

$$+ \frac{2}{\eta_x}\mathbb{E}\left[\left\|m^{k+1}\right\|_{\mathbf{P}}^2\right] + \frac{\nu^{-1}}{\tau}\mathbb{E}\left[\left\|\mathbf{y}_f^{k+1} + \mathbf{x}_f^{k+1} - \mathbf{y}^* - \mathbf{x}^*\right\|^2\right]$$

$$+ \left(1 - \frac{12\gamma\overline{L}^2\eta_z}{b\omega}\right)\frac{\omega}{\eta_z}\mathbb{E}\left[\left\|\mathbf{z}^{k+1} - \mathbf{w}^k\right\|^2\right] + \left(\frac{1}{\eta_z} + 2\mu - \nu - 12\gamma L^2\right)\mathbb{E}\left[\left\|\mathbf{z}^{k+1} - \mathbf{z}^*\right\|^2\right]$$

$$\leq (1 - \eta_y\gamma)\cdot\frac{1}{\eta_y}\mathbb{E}\left[\left\|\mathbf{y}^k - \mathbf{y}^*\right\|^2\right] + (1 - \eta_x\gamma)\cdot\frac{1}{\eta_x}\mathbb{E}\left[\left\|\hat{\mathbf{x}}^k - \mathbf{x}^*\right\|^2\right] + \alpha\cdot\frac{1}{2\eta_y}\mathbb{E}\left[\left\|\mathbf{y}^k - \mathbf{y}^{k-1}\right\|^2\right]$$

$$+ \alpha\cdot\frac{1}{4\eta_z}\mathbb{E}\left[\left\|\mathbf{z}^k - \mathbf{z}^{k-1}\right\|^2\right] - \alpha\cdot2\mathbb{E}\left[\langle\mathbf{F}(\mathbf{z}^k) - \mathbf{F}(\mathbf{z}^{k-1}) - (\mathbf{y}^k - \mathbf{y}^{k-1}), \mathbf{z}^k - \mathbf{z}^*\rangle\right]$$

$$+ (1 - (4\chi(T))^{-1})\cdot\frac{2}{\eta_x}\mathbb{E}\left[\left\|m^k\right\|_{\mathbf{P}}^2\right] + (1 - \tau)\cdot\frac{\nu^{-1}}{\tau}\mathbb{E}\left[\left\|\mathbf{y}_f^k + \mathbf{x}_f^k - \mathbf{y}^* - \mathbf{x}^*\right\|^2\right]$$

$$+ \frac{4\eta_z\overline{L}^2}{b}\mathbb{E}\left[\left\|\mathbf{z}^k - \mathbf{w}^{k-1}\right\|^2\right] + \left(\frac{1}{\eta_z} - \frac{\omega}{\eta_z} + \nu + 2\gamma\nu^2\right)\mathbb{E}\left[\left\|\mathbf{z}^k - \mathbf{z}^*\right\|^2\right]$$

$$+ \frac{\omega}{\eta_z}\mathbb{E}\left[\left\|\mathbf{w}^k - \mathbf{z}^*\right\|^2\right].$$

Now, we add $\frac{\gamma + \eta\nu}{p\eta_z}\mathbb{E}\left[\left\|\mathbf{w}^{k+1} - \mathbf{z}^*\right\|^2\right]$ to both sides and use update for $\mathbf{w}^{k+1}$

$$\frac{\omega + \eta_z\nu}{p\eta_z}\mathbb{E}\left[\mathbb{E}_{\mathbf{w}^{k+1}}\left\|\mathbf{w}^{k+1} - \mathbf{z}^*\right\|^2\right] = \frac{\omega + \eta_z\nu}{\eta_z}\mathbb{E}\left[\left\|\mathbf{z}^k - \mathbf{z}^*\right\|^2\right] + \frac{(\omega + \eta_z\nu)(1 - p)}{\eta_z p}\mathbb{E}\left[\left\|\mathbf{w}^k - \mathbf{z}^*\right\|^2\right],$$

and then

$$\frac{1}{\eta_y}\mathbb{E}\left[\left\|\mathbf{y}^{k+1} - \mathbf{y}^*\right\|^2\right] + \frac{1}{\eta_x}\mathbb{E}\left[\left\|\hat{\mathbf{x}}^{k+1} - \mathbf{x}^*\right\|^2\right] + \frac{1}{2\eta_y}\mathbb{E}\left[\left\|\mathbf{y}^{k+1} - \mathbf{y}^k\right\|^2\right]$$

$$+ \frac{1}{4\eta_z}\mathbb{E}\left[\left\|\mathbf{z}^{k+1} - \mathbf{z}^k\right\|^2\right] - 2\mathbb{E}\left[\langle\mathbf{F}(\mathbf{z}^{k+1}) - \mathbf{F}(\mathbf{z}^k) - (\mathbf{y}^{k+1} - \mathbf{y}^k), \mathbf{z}^{k+1} - \mathbf{z}^*\rangle\right]$$

$$+ \frac{2}{\eta_x}\mathbb{E}\left[\left\|m^{k+1}\right\|_{\mathbf{P}}^2\right] + \frac{\nu^{-1}}{\tau}\mathbb{E}\left[\left\|\mathbf{y}_f^{k+1} + \mathbf{x}_f^{k+1} - \mathbf{y}^* - \mathbf{x}^*\right\|^2\right]$$

$$+ \left(1 - \frac{12\gamma\overline{L}^2\eta_z}{b\omega}\right)\frac{\omega}{\eta_z}\mathbb{E}\left[\left\|\mathbf{z}^{k+1} - \mathbf{w}^k\right\|^2\right] + \left(\frac{1}{\eta_z} + 2\mu - \nu - 12\gamma L^2\right)\mathbb{E}\left[\left\|\mathbf{z}^{k+1} - \mathbf{z}^*\right\|^2\right]$$

$$+ \frac{\omega + \eta_z\nu}{p\eta_z}\mathbb{E}\left[\left\|\mathbf{w}^{k+1} - \mathbf{z}^*\right\|^2\right]$$

$$\leq (1 - \eta_y\gamma)\cdot\frac{1}{\eta_y}\mathbb{E}\left[\left\|\mathbf{y}^k - \mathbf{y}^*\right\|^2\right] + (1 - \eta_x\gamma)\cdot\frac{1}{\eta_x}\mathbb{E}\left[\left\|\hat{\mathbf{x}}^k - \mathbf{x}^*\right\|^2\right] + \alpha\cdot\frac{1}{2\eta_y}\mathbb{E}\left[\left\|\mathbf{y}^k - \mathbf{y}^{k-1}\right\|^2\right]$$

$$+ \alpha\cdot\frac{1}{4\eta_z}\mathbb{E}\left[\left\|\mathbf{z}^k - \mathbf{z}^{k-1}\right\|^2\right] - \alpha\cdot2\mathbb{E}\left[\langle\mathbf{F}(\mathbf{z}^k) - \mathbf{F}(\mathbf{z}^{k-1}) - (\mathbf{y}^k - \mathbf{y}^{k-1}), \mathbf{z}^k - \mathbf{z}^*\rangle\right]$$

$$+ (1 - (4\chi(T))^{-1})\cdot\frac{2}{\eta_x}\mathbb{E}\left[\left\|m^k\right\|_{\mathbf{P}}^2\right] + (1 - \tau)\cdot\frac{\nu^{-1}}{\tau}\mathbb{E}\left[\left\|\mathbf{y}_f^k + \mathbf{x}_f^k - \mathbf{y}^* - \mathbf{x}^*\right\|^2\right]$$

$$+ \frac{4\eta_z\overline{L}^2}{b}\mathbb{E}\left[\left\|\mathbf{z}^k - \mathbf{w}^{k-1}\right\|^2\right] + \left(\frac{1}{\eta_z} + 2\nu + 2\gamma\nu^2\right)\mathbb{E}\left[\left\|\mathbf{z}^k - \mathbf{z}^*\right\|^2\right]$$

$$+ \left(1 - \frac{p\eta_z\nu}{\omega + \eta_z\nu}\right)\cdot\frac{\omega + \eta_z\nu}{p\eta_z}\mathbb{E}\left[\left\|\mathbf{w}^k - \mathbf{z}^*\right\|^2\right].$$

With $\gamma \leq \frac{\mu}{48L^2}$ and $\nu \leq \frac{\mu}{4}$ we gat that

$$\left(1 + 2\mu\eta_x - \nu\eta_x - 12\gamma L^2\eta_x\right) \geq 1 + \frac{3\mu\eta_x}{2},$$

$$\left(\frac{1}{\eta_x} + 2\nu + 2\gamma\nu^2\right) \leq \left(1 + \frac{\mu\eta_x}{2} + \frac{2\mu^3\eta_x}{16 \cdot 48L^2}\right)\frac{1}{\eta_x} \leq (1 + \mu\eta_x)\frac{1}{\eta_x}.$$

Additionally, by

$$(1 + \mu\eta_x) \leq \left(1 + \frac{3\mu\eta_x}{2}\right)\left(1 - \frac{\mu\eta_x}{8}\right)$$

we obtain

$$\frac{1}{\eta_y}\mathbb{E}\left[\left\|\mathbf{y}^{k+1} - \mathbf{y}^*\right\|^2\right] + \frac{1}{\eta_x}\mathbb{E}\left[\left\|\hat{\mathbf{x}}^{k+1} - \mathbf{x}^*\right\|^2\right] + \frac{1}{2\eta_y}\mathbb{E}\left[\left\|\mathbf{y}^{k+1} - \mathbf{y}^k\right\|^2\right]$$

$$+ \frac{1}{4\eta_z}\mathbb{E}\left[\left\|\mathbf{z}^{k+1} - \mathbf{z}^k\right\|^2\right] - 2\mathbb{E}\left[\langle\mathbf{F}(\mathbf{z}^{k+1}) - \mathbf{F}(\mathbf{z}^k) - (\mathbf{y}^{k+1} - \mathbf{y}^k), \mathbf{z}^{k+1} - \mathbf{z}^*\rangle\right]$$

$$+ \frac{2}{\eta_x}\mathbb{E}\left[\left\|m^{k+1}\right\|_{\mathbf{P}}^2\right] + \frac{\nu^{-1}}{\tau}\mathbb{E}\left[\left\|\mathbf{y}_f^{k+1} + \mathbf{x}_f^{k+1} - \mathbf{y}^* - \mathbf{x}^*\right\|^2\right]$$

$$+ \left(1 - \frac{12\gamma\overline{L}^2\eta_z}{b\omega}\right)\frac{\omega}{\eta_z}\mathbb{E}\left[\left\|\mathbf{z}^{k+1} - \mathbf{w}^k\right\|^2\right] + \left(1 + \frac{3\mu\eta_x}{2}\right)\mathbb{E}\left[\left\|\mathbf{z}^{k+1} - \mathbf{z}^*\right\|^2\right]$$

$$+ \frac{\omega + \eta_z\nu}{p\eta_z}\mathbb{E}\left[\left\|\mathbf{w}^{k+1} - \mathbf{z}^*\right\|^2\right]$$

$$\leq (1 - \eta_y\gamma)\cdot\frac{1}{\eta_y}\mathbb{E}\left[\left\|\mathbf{y}^k - \mathbf{y}^*\right\|^2\right] + (1 - \eta_x\gamma)\cdot\frac{1}{\eta_x}\mathbb{E}\left[\left\|\hat{\mathbf{x}}^k - \mathbf{x}^*\right\|^2\right] + \alpha\cdot\frac{1}{2\eta_y}\mathbb{E}\left[\left\|\mathbf{y}^k - \mathbf{y}^{k-1}\right\|^2\right]$$

$$+ \alpha\cdot\frac{1}{4\eta_z}\mathbb{E}\left[\left\|\mathbf{z}^k - \mathbf{z}^{k-1}\right\|^2\right] - \alpha\cdot2\mathbb{E}\left[\langle\mathbf{F}(\mathbf{z}^k) - \mathbf{F}(\mathbf{z}^{k-1}) - (\mathbf{y}^k - \mathbf{y}^{k-1}), \mathbf{z}^k - \mathbf{z}^*\rangle\right]$$

$$+ (1 - (4\chi(T))^{-1})\cdot\frac{2}{\eta_x}\mathbb{E}\left[\left\|m^k\right\|_{\mathbf{P}}^2\right] + (1 - \tau)\cdot\frac{\nu^{-1}}{\tau}\mathbb{E}\left[\left\|\mathbf{y}_f^k + \mathbf{x}_f^k - \mathbf{y}^* - \mathbf{x}^*\right\|^2\right]$$

$$+ \frac{4\eta_z\overline{L}^2}{b}\mathbb{E}\left[\left\|\mathbf{z}^k - \mathbf{w}^{k-1}\right\|^2\right] + \left(1 - \frac{\mu\eta_x}{8}\right)\cdot\left(1 + \frac{3\mu\eta_x}{2}\right)\mathbb{E}\left[\left\|\mathbf{z}^k - \mathbf{z}^*\right\|^2\right]$$

$$+ \left(1 - \frac{p\eta_z\nu}{\omega + \eta_z\nu}\right)\cdot\frac{\omega + \eta_z\nu}{p\eta_z}\mathbb{E}\left[\left\|\mathbf{w}^k - \mathbf{z}^*\right\|^2\right].$$

The choice $\eta_z \leq \frac{\sqrt{\alpha\omega b}}{4\overline{L}}$ and $\gamma \leq \frac{b\omega}{24\overline{L}^2\eta_z}$ gives

$$\frac{1}{\eta_y}\mathbb{E}\left[\left\|\mathbf{y}^{k+1} - \mathbf{y}^*\right\|^2\right] + \frac{1}{\eta_x}\mathbb{E}\left[\left\|\hat{\mathbf{x}}^{k+1} - \mathbf{x}^*\right\|^2\right] + \frac{1}{2\eta_y}\mathbb{E}\left[\left\|\mathbf{y}^{k+1} - \mathbf{y}^k\right\|^2\right]$$

$$+ \frac{1}{4\eta_z}\mathbb{E}\left[\left\|\mathbf{z}^{k+1} - \mathbf{z}^k\right\|^2\right] - 2\mathbb{E}\left[\langle\mathbf{F}(\mathbf{z}^{k+1}) - \mathbf{F}(\mathbf{z}^k) - (\mathbf{y}^{k+1} - \mathbf{y}^k), \mathbf{z}^{k+1} - \mathbf{z}^*\rangle\right]$$

$$+ \frac{2}{\eta_x}\mathbb{E}\left[\left\|m^{k+1}\right\|_{\mathbf{P}}^2\right] + \frac{\nu^{-1}}{\tau}\mathbb{E}\left[\left\|\mathbf{y}_f^{k+1} + \mathbf{x}_f^{k+1} - \mathbf{y}^* - \mathbf{x}^*\right\|^2\right]$$

$$+ \frac{\omega}{2\eta_z}\mathbb{E}\left[\left\|\mathbf{z}^{k+1} - \mathbf{w}^k\right\|^2\right] + \left(1 + \frac{3\mu\eta_x}{2}\right)\mathbb{E}\left[\left\|\mathbf{z}^{k+1} - \mathbf{z}^*\right\|^2\right]$$

$$+ \frac{\omega + \eta_z\nu}{p\eta_z}\mathbb{E}\left[\left\|\mathbf{w}^{k+1} - \mathbf{z}^*\right\|^2\right]$$

$$\leq (1 - \eta_y\gamma)\cdot\frac{1}{\eta_y}\mathbb{E}\left[\left\|\mathbf{y}^k - \mathbf{y}^*\right\|^2\right] + (1 - \eta_x\gamma)\cdot\frac{1}{\eta_x}\mathbb{E}\left[\left\|\hat{\mathbf{x}}^k - \mathbf{x}^*\right\|^2\right] + \alpha\cdot\frac{1}{2\eta_y}\mathbb{E}\left[\left\|\mathbf{y}^k - \mathbf{y}^{k-1}\right\|^2\right]$$

$$+ \alpha\cdot\frac{1}{4\eta_z}\mathbb{E}\left[\left\|\mathbf{z}^k - \mathbf{z}^{k-1}\right\|^2\right] - \alpha\cdot2\mathbb{E}\left[\langle\mathbf{F}(\mathbf{z}^k) - \mathbf{F}(\mathbf{z}^{k-1}) - (\mathbf{y}^k - \mathbf{y}^{k-1}), \mathbf{z}^k - \mathbf{z}^*\rangle\right]$$

$$+ (1 - (4\chi(T))^{-1})\cdot\frac{2}{\eta_x}\mathbb{E}\left[\left\|m^k\right\|_{\mathbf{P}}^2\right] + (1 - \tau)\cdot\frac{\nu^{-1}}{\tau}\mathbb{E}\left[\left\|\mathbf{y}_f^k + \mathbf{x}_f^k - \mathbf{y}^* - \mathbf{x}^*\right\|^2\right]$$

$$+ \alpha\cdot\frac{\omega}{2\eta_z}\mathbb{E}\left[\left\|\mathbf{z}^k - \mathbf{w}^{k-1}\right\|^2\right] + \left(1 - \frac{\mu\eta_x}{8}\right)\cdot\left(1 + \frac{3\mu\eta_x}{2}\right)\mathbb{E}\left[\left\|\mathbf{z}^k - \mathbf{z}^*\right\|^2\right]$$

$$+ \left(1 - \frac{p\eta_z\nu}{\omega + \eta_z\nu}\right)\cdot\frac{\omega + \eta_z\nu}{p\eta_z}\mathbb{E}\left[\left\|\mathbf{w}^k - \mathbf{z}^*\right\|^2\right].$$

Putting $p = \omega$ and using definition of the Lyapunov function, we get

$$\mathbb{E}[\Psi^{k+1}] \leq \max\left[\left(1 - \frac{p\eta_z\nu}{p + \eta_z\nu}\right); 1 - \eta_y\gamma; 1 - \eta_x\gamma; 1 - \frac{\mu\eta_z}{8}; 1 - \tau; 1 - \frac{1}{4\chi(T)}\right]\cdot\mathbb{E}[\Psi^k].$$

Finally,

$$
\begin{aligned}
\Psi^k &\geq \frac{1}{\eta_x}\left\|x^k - \mathbf{x}^*\right\|^2 + \frac{1}{4\eta_x}\left\|x^{k-1} - x^k\right\|^2 + \frac{1}{2\eta_y}\left\|y^{k-1} - y^k\right\|^2 \\
&\quad - 2\langle \mathbf{F}(x^k) - \mathbf{F}(x^{k-1}) - (y^k - y^{k-1}), x^k - \mathbf{x}^*\rangle \\
&\geq \frac{1}{\eta_x}\left\|x^k - \mathbf{x}^*\right\|^2 + \frac{1}{4\eta_x}\left\|x^{k-1} - x^k\right\|^2 + \frac{1}{2\eta_y}\left\|y^{k-1} - y^k\right\|^2 \\
&\quad - \frac{1}{2\eta_y}\left\|y^k - y^{k-1}\right\|^2 - 2\eta_y\left\|x^k - \mathbf{x}^*\right\|^2 - \frac{1}{4\eta_x}\left\|x^{k-1} - x^k\right\|^2 - 4\eta_x L^2\left\|x^k - \mathbf{x}^*\right\|^2 \\
&= \left(1 - 2\eta_x\eta_y - 4\eta_x^2 L^2\right)\frac{1}{\eta_x}\left\|x^k - \mathbf{x}^*\right\|^2 \\
&\geq \frac{1}{4\eta_x}\left\|x^k - \mathbf{x}^*\right\|^2.
\end{aligned}
$$

$\square$

**Theorem E.3** (Theorem 4.3)**.** *Consider the problem (10) (or (1) + (5)) under Assumptions 2.1 and 2.2 over a sequence of time-varying graphs $\mathcal{G}(k)$ with gossip matrices $\mathbf{W}(k)$. Let $\{\mathbf{z}^k\}$ be the sequence generated by Algorithm 2 with parameters*

$$
T \geq B; \quad \omega = p \leq \frac{1}{16}; \quad \theta = \frac{1}{2}; \quad \beta = 5\gamma; \quad \nu = \frac{\mu}{4}; \quad \tau = \min\left\{\frac{\mu}{32L\chi(T)}; \frac{\mu\sqrt{bp}}{32\overline{L}}\right\};
$$

$$
\eta_z = \min\left\{\frac{1}{8L\chi(T)}, \frac{1}{32\eta_y}, \frac{\sqrt{\alpha b\omega}}{8\overline{L}}\right\}; \quad \eta_y = \min\left\{\frac{1}{4\gamma}, \frac{\nu}{8\tau}\right\}; \quad \eta_x = \min\left\{\frac{1}{900\chi(T)\gamma}, \frac{\nu}{36\tau\chi^2(T)}\right\};
$$

$$
\gamma = \min\left\{\frac{\mu}{16L^2}; \frac{b\omega}{24\overline{L}^2\eta_z}\right\},
$$

$$
\alpha = \max\left[\left(1 - \frac{p\eta_z\nu}{p + \eta_z\nu}\right); 1 - \eta_y\gamma; 1 - \eta_x\gamma; 1 - \frac{\mu\eta_z}{8}; 1 - \tau; 1 - \frac{1}{4\chi(T)}\right].
$$

*Let the choice of $T$ guarantees contraction property (Assumption 2.4 point 4) with $\chi(T)$. Then, given $\varepsilon > 0$, the number of iterations for $\mathbb{E}[\|\mathbf{z}^k - \mathbf{z}^*\|^2] \leq \varepsilon$ is*

$$
O\left(\left[\chi^2(T) + \frac{1}{p} + \chi(T)\frac{L}{\mu} + \frac{1}{\sqrt{bp}}\frac{\overline{L}}{\mu}\right]\log\frac{1}{\varepsilon}\right).
$$

*Proof.* It is easy to check that $\alpha \geq \frac{1}{2}$, then also one can verify that the choice of $\omega, \theta, \beta, \nu, \tau, \eta_z, \eta_y, \eta_x, \gamma, \alpha$ satisfies the conditions of Lemma E.2. We can get that the iteration complexity of Algorithm 2:

$$
O\left(\left[\chi(T) + \frac{1}{p} + \frac{1}{\eta_y\gamma} + \frac{1}{\eta_x\gamma} + \frac{1}{\eta_z\mu} + \frac{1}{\eta_z\nu} + \frac{1}{\tau} + \frac{1}{1-\alpha}\right]\log\frac{1}{\varepsilon}\right).
$$

Substituting $\alpha, \nu, \omega$, we get

$$
O\left(\left[\chi(T) + \frac{1}{p} + \frac{1}{\eta_y\gamma} + \frac{1}{\eta_x\gamma} + \frac{1}{\eta_z\mu} + \frac{1}{\tau}\right]\log\frac{1}{\varepsilon}\right).
$$

Putting $\eta_z$

$$
O\left(\left[\chi(T) + \frac{1}{p} + \chi(T)\frac{L}{\mu} + \frac{1}{\sqrt{bp}}\frac{\overline{L}}{\mu} + \frac{1}{\eta_y\gamma} + \frac{1}{\eta_x\gamma} + \frac{\eta_y}{\mu} + \frac{1}{\tau}\right]\log\frac{1}{\varepsilon}\right).
$$

Substituting $\eta_y$ and $\eta_x$

$$
O\left(\left[\chi(T) + \frac{1}{p} + \chi(T)\frac{L}{\mu} + \frac{1}{\sqrt{bp}}\frac{\overline{L}}{\mu} + \frac{\tau\chi^2(T)}{\mu\gamma} + \frac{\eta_y}{\mu} + \frac{1}{\tau}\right]\log\frac{1}{\varepsilon}\right).
$$

Using $\eta_y \leq \frac{\nu}{8\tau}$

$$
O\left(\left[\chi(T) + \frac{1}{p} + \chi(T)\frac{L}{\mu} + \frac{1}{\sqrt{bp}}\frac{\overline{L}}{\mu} + \frac{\tau\chi^2(T)}{\mu\gamma} + \frac{1}{\tau}\right]\log\frac{1}{\varepsilon}\right).
$$

Substituting $\tau$ we get

$$
O\left(\left[\chi(T) + \frac{1}{p} + \chi(T)\frac{L}{\mu} + \frac{1}{\sqrt{bp}}\frac{\overline{L}}{\mu} + \frac{\tau\chi^2(T)}{\mu\gamma}\right]\log\frac{1}{\varepsilon}\right).
$$

Putting $\gamma$ we obtain

$$O\left(\left[\chi(T) + \frac{1}{p} + \chi(T)\frac{L}{\mu} + \frac{1}{\sqrt{bp}}\frac{\overline{L}}{\mu} + \frac{\tau\overline{L}^2\eta_z\chi^2(T)}{\mu bp} + \frac{\tau L^2\chi^2(T)}{\mu^2}\right]\log\frac{1}{\varepsilon}\right).$$

With $\eta_z \leq \frac{\sqrt{abp}}{8\overline{L}}$ and $\tau \leq \frac{\mu\sqrt{bp}}{\overline{L}}$ we have

$$O\left(\left[\chi^2(T) + \frac{1}{p} + \chi(T)\frac{L}{\mu} + \frac{1}{\sqrt{bp}}\frac{\overline{L}}{\mu} + \frac{\tau L^2\chi^2(T)}{\mu^2}\right]\log\frac{1}{\varepsilon}\right).$$

Finally, $\tau \leq \frac{1}{8L\chi(T)}$

$$O\left(\left[\chi^2(T) + \frac{1}{p} + \chi(T)\frac{L}{\mu} + \frac{1}{\sqrt{bp}}\frac{\overline{L}}{\mu}\right]\log\frac{1}{\varepsilon}\right).$$

$\square$

# F Additional experiments

## F.1 Variance reduction

Here we give additional experiments for Section 5.1 on other matrices.

Figure 4: Comparison epoch complexities of Algorithm 1, EG-Alc-Alg1, EG-Alc-Alg2 and EG-Car on (11) with matrices from [55] (two upper lines correspond to test matrix 1, the two bottom lines – to test matrix 2). Dashed lines give convergence with theoretical parameters, solid lines – with tuned parameters.

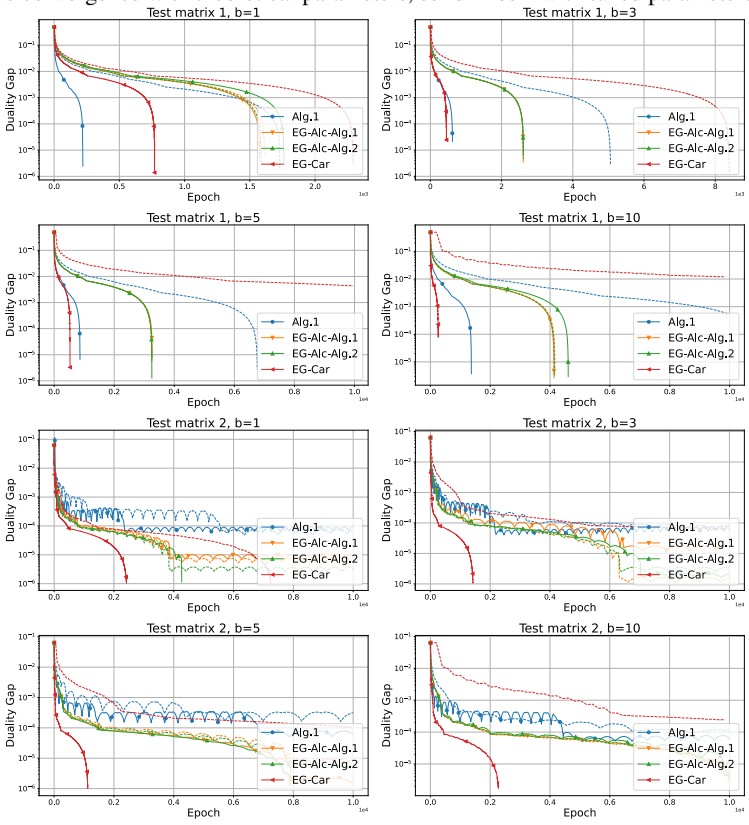

## F.2 Decentralized methods

### F.2.1 Fixed networks

Here we give additional experiments for Section 5.2.1.

Figure 5: Comparison communication complexities of Algorithm 1, EGD-GT, EGD-Con and Sliding on (12) with over fixed grid networks.

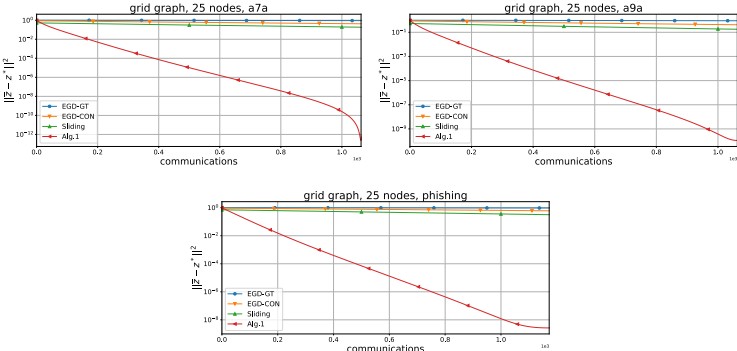

Figure 6: Comparison communication complexities of Algorithm 1, EGD-GT, EGD-Con and Sliding on (12) with over fixed ring networks.

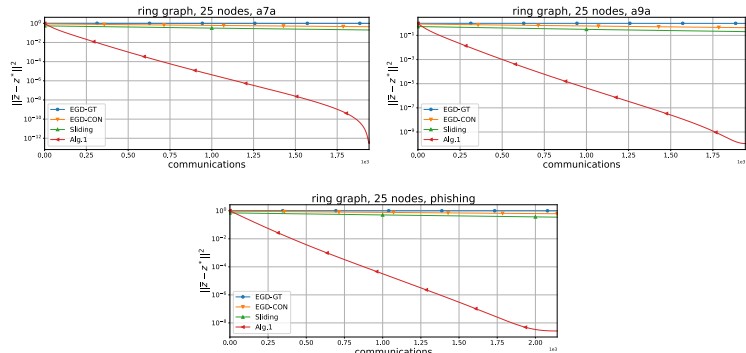

Figure 7: Comparison communication complexities of Algorithm 1, EGD-GT, EGD-Con and Sliding on (12) with over fixed star networks.

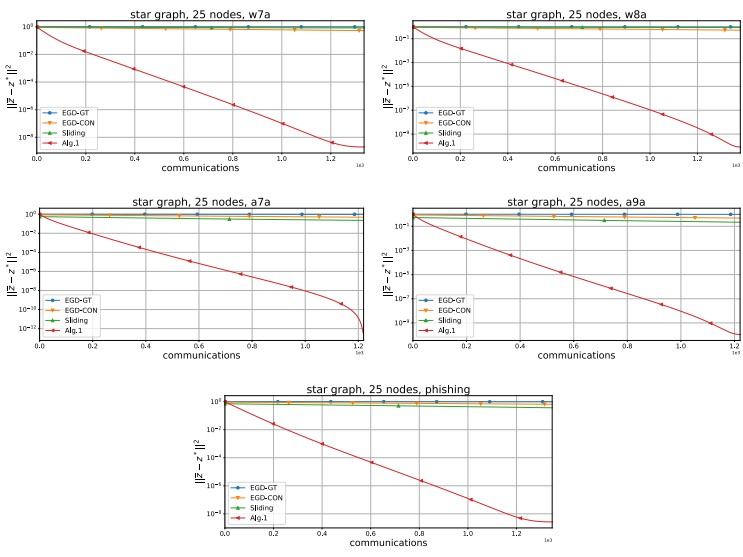

### F.2.2 Time-varying networks

Here we give additional experiments for Section 5.2.2.

Figure 8: Comparison communication complexities of Algorithm 2, EGD-GT, EGD-Con and Sliding on (12) over time-varying grid networks.

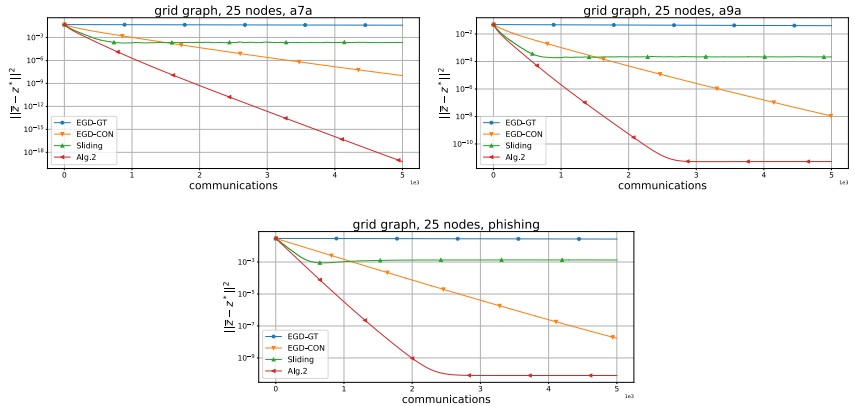

Figure 9: Comparison communication complexities of Algorithm 1, EGD-GT, EGD-Con and Sliding on (12) with over fixed ring networks.

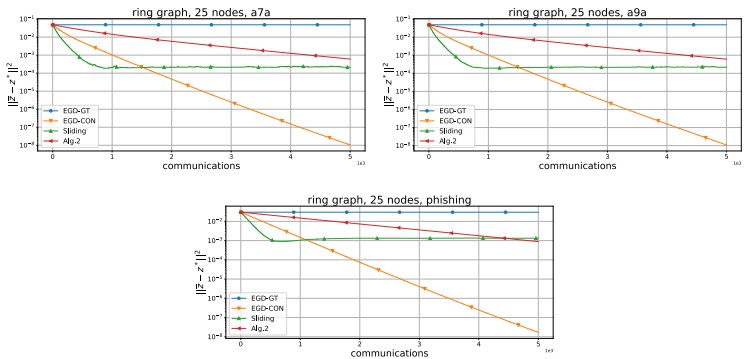

# G Tables

Table 2: Summary communication and local complexities for finding an $\varepsilon$-solution for strongly monotone **deterministic decentralized** variational inequality (1) on fixed and time-varying networks. Convergence is measured by the distance to the solution. *Notation:* $\mu$ = constant of strong monotonicity of the operator $F$, $L$ = Lipschitz constants for all $L_{m,i}$, $\chi$ = characteristic number of the network (see Assumptions 2.3 and 2.4).

| | | Reference | Communication complexity | Local complexity | Weaknesses |
|---|---|---|---|---|---|
| **Fixed** | **Upper** | Mukherjee and Chakraborty [48] [1,2] | $\mathcal{O}\left(\chi^{\frac{4}{3}}\frac{L^{\frac{4}{3}}}{\mu^{\frac{4}{3}}}\log\frac{1}{\varepsilon}\right)$ | $\mathcal{O}\left(\chi^{\frac{4}{3}}\frac{L^{\frac{4}{3}}}{\mu^{\frac{4}{3}}}\log\frac{1}{\varepsilon}\right)$ | bad comm. rates bad local rates |
| | | Beznosikov et al. [14] [1,2] | $\mathcal{O}\left(\sqrt{\chi}\frac{L}{\mu}\log^2\frac{1}{\varepsilon}\right)$ | $\mathcal{O}\left(\frac{L}{\mu}\log\frac{L+\mu}{\mu}\log\frac{1}{\varepsilon}\right)$ | multiple gossip no linear convergence |
| | | Beznosikov et al. [13] [1,3] | $\mathcal{O}\left(\sqrt{\chi}\frac{L}{\mu}\log^2\frac{1}{\varepsilon}\right)$ | $\mathcal{O}\left(\frac{L}{\mu}\log\frac{1}{\varepsilon}\right)$ | multiple gossip no linear convergence |
| | | Rogozin et al. [62] [1,2,4] | $\mathcal{O}\left(\sqrt{\chi}\frac{L}{\mu}\log\frac{1}{\varepsilon}\right)$ | $\mathcal{O}\left(\sqrt{\chi}\frac{L}{\mu}\log\frac{1}{\varepsilon}\right)$ | multiple gossip bounded gradient |
| | | This paper | $\mathcal{O}\left(\sqrt{\chi}\frac{L}{\mu}\log\frac{1}{\varepsilon}\right)$ | $\mathcal{O}\left(\sqrt{\chi}\frac{L}{\mu}\log\frac{1}{\varepsilon}\right)$ | |
| | | This paper | $\mathcal{O}\left(\sqrt{\chi}\frac{L}{\mu}\log\frac{1}{\varepsilon}\right)$ | $\mathcal{O}\left(\frac{L}{\mu}\log\frac{1}{\varepsilon}\right)$ | multiple gossip |
| | **Lower** | Beznosikov et al. [13] [3] | $\mathcal{O}\left(\sqrt{\chi}\frac{L}{\mu}\log\frac{1}{\varepsilon}\right)$ | $\mathcal{O}\left(\frac{L}{\mu}\log\frac{1}{\varepsilon}\right)$ | |
| | | This paper | $\mathcal{O}\left(\sqrt{\chi}\frac{L}{\mu}\log\frac{1}{\varepsilon}\right)$ | $\mathcal{O}\left(\frac{L}{\mu}\log\frac{1}{\varepsilon}\right)$ | |
| **Time-varying** | **Upper** | Beznosikov et al. [10] [3] | $\mathcal{O}\left(\chi\frac{L}{\mu}\log\frac{1}{\varepsilon} + \chi\frac{LD}{\mu^2\sqrt{\varepsilon}}\right)$ [5] | $\mathcal{O}\left(\chi\frac{L}{\mu}\log\frac{1}{\varepsilon} + \chi\frac{LD}{\mu^2\sqrt{\varepsilon}}\right)$ | $D$-homogeneity no linear convergence |
| | | Beznosikov et al. [12] [1,2] | $\mathcal{O}\left(\chi\frac{L}{\mu}\log^2\frac{1}{\varepsilon}\right)$ | $\mathcal{O}\left(\frac{L}{\mu}\log\frac{1}{\varepsilon}\right)$ | multiple gossip no linear convergence |
| | | This paper | $\mathcal{O}\left(\chi\frac{L}{\mu}\log\frac{1}{\varepsilon}\right)$ [5] | $\mathcal{O}\left(\chi\frac{L}{\mu}\log\frac{1}{\varepsilon}\right)$ | |
| | | This paper | $\mathcal{O}\left(\chi\frac{L}{\mu}\log\frac{1}{\varepsilon}\right)$ [5] | $\mathcal{O}\left(\frac{L}{\mu}\log\frac{1}{\varepsilon}\right)$ | multiple gossip |
| | **Lower** | Beznosikov et al. [12] [2] | $\mathcal{O}\left(\chi\frac{L}{\mu}\log\frac{1}{\varepsilon}\right)$ | $\mathcal{O}\left(\frac{L}{\mu}\log\frac{1}{\varepsilon}\right)$ | |
| | | This paper | $\mathcal{O}\left(\chi\frac{L}{\mu}\log\frac{1}{\varepsilon}\right)$ [5] | $\mathcal{O}\left(\frac{L}{\mu}\log\frac{1}{\varepsilon}\right)$ | |

[1] for saddle point problems
[2] deterministic
[3] stochastic, but not finite sum
[4] for convex-concave (monotone) case (we reanalyzed for strongly monotone case)
[5] $B$-connected graphs are also considered. For simplicity in comparison with other works, we put $B = 1$. To get estimates for $B \neq 1$, one need to change $\chi$ to $B\chi$

Table 3: Summary complexities for finding an $\varepsilon$-solution for strongly monotone **stochastic (finite-sum) non-distributed** variational inequality (1). Convergence is measured by the distance to the solution. *Notation:* $\mu$ = constant of strong monotonicity of the operator $F$, $L$ = Lipschitz constants for all $L_i$, $n$ = the size of the local dataset.

| Reference | Complexity | Weaknesses |
|---|---|---|
| Palaniappan and Bach [59] - SVRG [1] | $\mathcal{O}\left(n + b\frac{L^2}{\mu^2}\log\frac{1}{\varepsilon}\right)$ | bad complexity batching |
| Palaniappan and Bach [59] - Acc-SVRG [1] | $\mathcal{O}\left(n + \sqrt{bn}\frac{L}{\mu}\log^2\frac{1}{\varepsilon}\right)$ | envelope acceleration batching |
| Chavdarova et al. [20] [2] | $\mathcal{O}\left(n + b\frac{L^2}{\mu^2}\log\frac{1}{\varepsilon}\right)$ | bad complexity batching |
| Carmon et al. [17][1] | $\mathcal{O}\left(n + \sqrt{bn}\frac{L}{\mu}\log\frac{1}{\varepsilon}\right)$ | for games only batching |
| Carmon et al. [17][1] | $\tilde{\mathcal{O}}\left(n + \sqrt{bn}\frac{L}{\mu}\log\frac{1}{\varepsilon}\right)$ | batching |
| Yang et al. [70] [1,3] | $\mathcal{O}\left(b^{\frac{1}{3}}n^{\frac{2}{3}}\frac{L^3}{\mu^3}\log\frac{1}{\varepsilon}\right)$ | bad complexity batching |
| Alacaoglu and Malitsky [1] | $\mathcal{O}\left(n + \sqrt{bn}\frac{L}{\mu}\log\frac{1}{\varepsilon}\right)$ | batching |
| Alacaoglu et al. [2] | $\mathcal{O}\left(n + n\frac{L}{\mu}\log\frac{1}{\varepsilon}\right)$ | bad rates |
| Tominin et al. [65] [1] | $\mathcal{O}\left(n + \sqrt{bn}\frac{L}{\mu}\log^2\frac{1}{\varepsilon}\right)$ | envelope acceleration batching |
| Beznosikov et al. [11] [2] | $\mathcal{O}\left(n + b\frac{L^2}{\mu^2}\log\frac{1}{\varepsilon}\right)$ | bad complexity batching |
| This paper | $\mathcal{O}\left(n + \sqrt{n}\frac{L}{\mu}\log\frac{1}{\varepsilon}\right)$ | |

[1] for saddle point problems

[2] under $l$-cocoercivity assumption (in genral case $l = \frac{L^2}{\mu}$)

[3] under PL - condition

Table 4: Summary communication and local complexities for finding an $\varepsilon$-solution for strongly monotone **decentralized** variational inequality [1] on fixed and time-varying networks. Convergence is measured by the distance to the solution. *Notation:* $\mu$ = constant of strong monotonicity of the operator $F$, $L$ = Lipschitz constants for all $L_{m,i}$, $\chi$ = characteristic number of the network (see Assumptions 2.3 and 2.4).

| | | Reference | Communication complexity | Local complexity | Deter/Stoch? |
|---|---|---|---|---|---|
| **Fixed** | **Upper** | Mukherjee and Chakraborty [48] [1] | $\mathcal{O}\left(\chi^{\frac{4}{3}}\frac{L^{\frac{4}{3}}}{\mu^{\frac{4}{3}}}\log\frac{1}{\varepsilon}\right)$ | $\mathcal{O}\left(\chi^{\frac{4}{3}}\frac{L^{\frac{4}{3}}}{\mu^{\frac{4}{3}}}\log\frac{1}{\varepsilon}\right)$ | deterministic |
| | | Beznosikov et al. [14] [1] | $\mathcal{O}\left(\sqrt{\chi}\frac{L}{\mu}\log^2\frac{1}{\varepsilon}\right)$ | $\mathcal{O}\left(\frac{L}{\mu}\log\frac{L+\mu}{\mu}\log\frac{1}{\varepsilon}\right)$ | deterministic |
| | | Beznosikov et al. [13] [1] | $\mathcal{O}\left(\sqrt{\chi}\frac{L}{\mu}\log^2\frac{1}{\varepsilon}\right)$ | $\mathcal{O}\left(\frac{L}{\mu}\log\frac{1}{\varepsilon}+\frac{\sigma^2}{\mu^2 M\varepsilon}\right)$ | stochastic $\sigma^2$-bounded variance |
| | | Rogozin et al. [62] [1,2] | $\mathcal{O}\left(\sqrt{\chi}\frac{L}{\mu}\log\frac{1}{\varepsilon}\right)$ | $\mathcal{O}\left(\sqrt{\chi}\frac{L}{\mu}\log\frac{1}{\varepsilon}\right)$ | deterministic |
| | | This paper | $\mathcal{O}\left(\max[\sqrt{n};\sqrt{\chi}]\frac{L}{\mu}\log\frac{1}{\varepsilon}\right)$ | $\mathcal{O}\left(\max[\sqrt{n};\sqrt{\chi}]\frac{L}{\mu}\log\frac{1}{\varepsilon}\right)$ | stochastic finite-sum |
| | | This paper | $\mathcal{O}\left(\sqrt{\chi}\frac{L}{\mu}\log\frac{1}{\varepsilon}\right)$ | $\mathcal{O}\left(\sqrt{n}\frac{L}{\mu}\log\frac{1}{\varepsilon}\right)$ | stochastic finite-sum |
| | **Lower** | Beznosikov et al. [13] | $\mathcal{O}\left(\sqrt{\chi}\frac{L}{\mu}\log\frac{1}{\varepsilon}\right)$ | $\mathcal{O}\left(\frac{L}{\mu}\log\frac{1}{\varepsilon}+\frac{\sigma^2}{\mu^2 M\varepsilon}\right)$ | stochastic $\sigma^2$-bounded variance |
| | | This paper | $\mathcal{O}\left(\sqrt{\chi}\frac{L}{\mu}\log\frac{1}{\varepsilon}\right)$ | $\mathcal{O}\left(\sqrt{n}\frac{L}{\mu}\log\frac{1}{\varepsilon}\right)$ | stochastic finite-sum |
| **Time-varying** | **Upper** | Beznosikov et al. [10] | $\mathcal{O}\left(\chi\frac{L}{\mu}\log\frac{1}{\varepsilon}+\frac{L(\chi D+\sqrt{\chi}\sigma)}{\mu^2\sqrt{\varepsilon}}+\frac{\sigma^2}{\mu^2 M\varepsilon}\right)$ [3] | $\mathcal{O}\left(\chi\frac{L}{\mu}\log\frac{1}{\varepsilon}+\frac{L(\chi D+\sqrt{\chi}\sigma)}{\mu^2\sqrt{\varepsilon}}+\frac{\sigma^2}{\mu^2 M\varepsilon}\right)$ | $D$-homogeneity $\sigma^2$-bounded variance |
| | | Beznosikov et al. [12] [1] | $\mathcal{O}\left(\chi\frac{L}{\mu}\log^2\frac{1}{\varepsilon}\right)$ | $\mathcal{O}\left(\frac{L}{\mu}\log\frac{1}{\varepsilon}\right)$ | deterministic |
| | | This paper | $\mathcal{O}\left(\chi\frac{L}{\mu}\log\frac{1}{\varepsilon}\right)$ [3] | $\mathcal{O}\left(\max[\sqrt{n};\chi]\frac{L}{\mu}\log\frac{1}{\varepsilon}\right)$ | stochastic finite-sum |
| | | This paper | $\mathcal{O}\left(\chi\frac{L}{\mu}\log\frac{1}{\varepsilon}\right)$ [3] | $\mathcal{O}\left(\sqrt{n}\frac{L}{\mu}\log\frac{1}{\varepsilon}\right)$ | stochastic finite-sum |
| | **Lower** | Beznosikov et al. [12] | $\mathcal{O}\left(\chi\frac{L}{\mu}\log\frac{1}{\varepsilon}\right)$ | $\mathcal{O}\left(\frac{L}{\mu}\log\frac{1}{\varepsilon}\right)$ | deterministic |
| | | This paper | $\mathcal{O}\left(\chi\frac{L}{\mu}\log\frac{1}{\varepsilon}\right)$ [3] | $\mathcal{O}\left(\sqrt{n}\frac{L}{\mu}\log\frac{1}{\varepsilon}\right)$ | stochastic finite-sum |

[1] for saddle point problems

[2] for convex-concave (monotone) case (we reanalyzed for strongly monotone case)

[3] $B$-connected graphs are also considered. For simplicity in comparison with other works, we put $B=1$. To get estimates for $B \neq 1$, one need to change $\chi$ to $B\chi$