# OpenReview forum: "Optimal Algorithms for Decentralized Stochastic Variational Inequalities"
_NeurIPS.cc/2022/Conference — NeurIPS 2022 Accept_

### Official Review · Reviewer_4cfY · 2022-07-06

**Rating:** 5
**Confidence:** 3
**Soundness:** 3 good
**Presentation:** 2 fair
**Contribution:** 2 fair

**Summary:**

This paper considers the decentralized variational inequality problem and focuses on the setting when the VI is strongly monotone. Particularly, it studies the optimal stochastic algorithm under the finite sum setting through deriving a lower communication/computation complexity bound and an algorithm that achieves the bound. Numerical experiments are also presented for the proposed "optimal" algorithm.

**Questions:**

In addition to the above mentioned weaknesses, there are a number of typos/issues throughout the paper:

1. The definition of (7) can be problematic:

It is in fact described in the equation that the new iterate $z$ is taken from the span of $z'$, and $\sum_{i_m} F_{m,i_m}(z^{''})$ for some given $z', z^{''} \in {\cal M}_m$.

This seems to suggest that $z',z^{''}$ have to be fixed. This is different from the framework of [29,59,68] which defines $z$ to be in the span of $\{ z', \sum_{i_m} F_{m,i_m}(z'') : z', z'' \in {\cal M}_m \}$. That said, this seems to be merely a typo since the paper mainly adopting the results from [29,59,68].

2. In (3), it should be $\max_y$ instead of $\min_y$?

3. In line 761, "for the fixed it is real distance" misses the word "graph".

**Limitations:**

The limitations have been described in the reviews of "Weaknesses" and "Questions".

**Strengths And Weaknesses:**

**Strengths**: The settings considered for the optimal decentralized stochastic VI algorithms is new. Furthermore, the proposed "optimal" algorithms have demonstrated promising performance as reported from the numerical experiments.

**Weaknesses**: While the settings considered for the optimal algorithm is new, it seems that the technical novelty is limited. Moreover the reviewer finds a number of issues with the definition of the problem class considered. As follows,

1. The reviewer is concerned about the role of the convex function $g$ as stated in (1) in the lower bound analysis. In fact, as seen in (7)-(8) of Sec. 3, [also (19) in Appendix C], the class of algorithm under consideration seems to be independent of $g$, i.e., the lower bounds are applicable only when $g = 0$. This leads to the further confusion as the proposed "optimal" algorithms actually use a proximal operator depending on $g$, which made the algorithms to be outside of the algorithm class analyzed in Sec. 3. Strictly speaking, the said result is still giving a lower bound complexity to solving the VI problem, yet it should be motivated clearly in the analysis setup for why $g$ can be ignored.

2. From a quick scan on the appendix, it seems the key analysis depends mainly on [68]. The said reference has not been discussed in the main paper while it is necessary to discuss the main novelty here, e.g., in terms of the proof techniques?

---

> ### Author Response · Authors · 2022-07-30
> **For Reviewer 4cfY (Part II)**
>
> > **From a quick scan on the appendix, it seems the key analysis depends mainly on [68].** The said reference has not been discussed in the main paper while it is necessary to discuss the main novelty here, e.g., in terms of the proof techniques?
>
> [1] ([68] in the review) is devoted to lower bounds for deterministic non-distributed VIs. Our lower bounds is for distributed stochastic problems. We take the same step as papers [2] and [3] did in their time. From Nesterov's lower bounds [4] for the deterministic non-distributed minimization problems, they obtained lower bounds for the deterministic distributed minimization problems and then for the stochastic distributed ones.
>
> We added some more references and ideas about lower bounds (see lines 209 and 651 in the revision).
>
> > **It seems that the technical novelty is limited**
>
> We see comments and questions only about lower bounds, but **70 percent of our contribution is algorithms**! By themselves, these algorithms without lower bounds are a big and interesting contribution (because their upper bounds are SOTA), and lower bounds complement them well and show optimality.
>
>
> > **The definition of (7) can be problematic**
>
> This is a typo! We deleted the word “given” from the definition.
>
> > **2 more typos**
>
> We fixed it. Thanks!
>
> [1] Junyu Zhang, Mingyi Hong, and Shuzhong Zhang. On lower iteration complexity bounds for the saddle point problems.
>
> [2] Kevin Scaman, Francis Bach, Sébastien Bubeck, Yin Tat Lee, and Laurent Massoulié. Optimal algorithms for smooth and strongly convex distributed optimization in networks.
>
> [3] Hadrien Hendrikx, Francis Bach, and Laurent Massoulie. An optimal algorithm for decentralized finite sum optimization.
>
> [4] Yurii Nesterov. Introductory lectures on convex optimization : a basic course.

---

> ### Author Response · Authors · 2022-07-30
> **For Reviewer 4cfY (Part I)**
>
> We thank Reviewer **4cfY**  for review, time and the insightful comments, which will help to improve our work.
>
> Next, we answer the questions and shortcomings noted by Reviewer.
>
> > **The reviewer is concerned about the role of the convex function g  as stated in (1) in the lower bound analysis.** In fact, as seen in (7)-(8) of Sec. 3, [also (19) in Appendix C], the class of algorithm under consideration seems to be independent of g, i.e., the lower bounds are applicable only when g=0. This leads to the further confusion as the proposed "optimal" algorithms actually use a proximal operator depending on g, which made the algorithms to be outside of the algorithm class analyzed in Sec. 3. Strictly speaking, the said result is still giving a lower bound complexity to solving the VI problem, yet it should be motivated clearly in the analysis setup for why g can be ignored.
>
> 1) We thank Reviewer! We changed Definition 3.1 and added the ability to compute $\text{prox}_g$ in local computations. Please see (7) in Definition 3.1 of  the revision.
>
> 2) We also added to the revision that $g$ is a proximally friendly function, i.e. $\text{prox}_g$ can be computed for free. This is a classic assumption when considering composite problems, because usually the objective function is complex, and $g$ is, for example, some simple regularizer for which one can analytically calculate the formula for the proximal operator.
>
> 3) Adding the ability to calculate prox in Definition 3.1 does not affect the resulting lower bounds. When we get lower bounds we need to present a particular bad problem from the class of problems that makes all algorithms converge in a bad way.
> We consider the following class of problems: distributed variational inequalities, where the operator $F$ is strongly monotone (Assumption 2.2) and Lipschitz continuous
>  (Assumption 2.1), and the composite function $g$ is proper lower semicontinuous convex.
> As we have already said, in order to obtain lower bounds, we need to choose a bad example from this class of problems. We take some operator $F$, and $g = 0$. Is $g = 0$ proper lower semicontinuous convex? Yes. And $\text{prox}_g (z) = z$ for $g = 0$. It turns out that in such a situation, the calculation of the prox operator does not change anything. We indicate this in Appendix C, in particular, because of this we do not need to change the proof in any way.
>
>
> 4) If Reviewer wants to see how we can obtain lower bounds with  $g \neq 0$, then the following trick can be done. In the current version of the paper, when obtaining lower bounds, we solve the saddle point problem on $R^{d_x} \times R^{d_y}$:
> $$
> \min_{R^{d_x}} \max_{R^{d_y}} f(x,y).
> $$
> This problem has a solution $x^*_{orig}, y ^*_{orig}$. Let us define sets $\mathcal{X} = B(0, R)$ and $\mathcal{Y} = B(0, R)$ – balls with centers in $0$ and with radiuses are equal to $R$, where $R > \max( \|x^*_{orig}\|, \| y^*_{orig}\|)$. It means that $x^*, y^*$ in $\mathcal{X} \times \mathcal{Y}$.
> Let us consider the next problem for lower bounds:
> $$
> \min_{X} \max_{Y} f(x,y).
> $$
> The solution of this problem is still at the point $x^*_{orig}$ and $y^*_{orig}$. We can look at this problem differently and write in the following form
> $$
> \min_{R^{d_x}} \max_{R^{d_y}} f(x,y) + g_1 (x) + g_2(y),
> $$
> where $g_1$ and $g_2$ are indicator functions, i.e. $g_1 (x) = 0$ if $x \in \mathcal{X}$ and $g_1(x) = + \infty$ if $x \notin \mathcal{X}$ (the same for $g_2$ and $\mathcal{Y}$). When we rewrite
> $
> \min_{R^{d_x}} \max_{R^{d_y}} f(x,y) + g_1 (x) + g_2(y)
> $
> as the variational inequality, $f(x,y)$ goes to the operator $F$ and $g(z) = g_1 (x) + g_2(y)$. $g(z)$ is a convex function.
> Let us understand how the lower bounds change. Note that in this case $\text{prox}_g$ is the Euclidean projections onto the balls, i.e.
> $$
> \text{prox}_g(z) = \binom{\text{proj}_B (x)}{\text{proj}_B (y)}
> $$
> and $\text{proj}_B (x) = x$ if $x \in B$, but if $x \notin B$, then
> $$
> \text{proj}_B (x) = x \cdot \frac{R}{|| x||}.
> $$
> The main idea behind the lower bounds is the number of non-zero coordinates that we can guarantee in the final output. Note that our proximal operator cannot increase the number of non-zero coordinates. Then the reasoning from our proofs will be completely valid.

---

> ### Comment · Reviewer_4cfY · 2022-08-06
> **Thank you**
>
> Thanks for the response. The authors have addressed my previous concern and I have decided to raise my score.

---

> > ### Author Response · Authors · 2022-08-06
> > **Thank you for the reply!**
> >
> > We greatly appreciate the response, the comments, and the raising of the score! We kindly ask you to let us know if any issues are left that we could address. Perhaps we haven't solved all the problems mentioned in the review? We ask this because, at this point, we don't fully understand the score. The main contribution of our paper is the algorithms that are not only state-of-the-art but are provably optimal, i.e., they have the best possible complexities due to our lower bounds. However, we could not find any negative points about the algorithms in the review.

---

### Official Review · Reviewer_JEvK · 2022-07-11

**Rating:** 7
**Confidence:** 4
**Soundness:** 4 excellent
**Presentation:** 3 good
**Contribution:** 3 good

**Summary:**

Methods for variational inequalities are applied to solve problems in optimization, machine learning, image processing, game theory, etc. This paper provides a lower complexity bound in communication and computation for decentralized stochastic variational inequalities under some conditions. Then optimal algorithms are proposed to match these lower bounds. This paper considers a strong monotone operator for which linear convergence is derived.

**Questions:**

+ In the complexity, the term $n\log{1\over \epsilon}$ is not included in the computation, and the term $\sqrt{n\chi}\log{1\over \epsilon}$ is not included in the communication (they are both in Theorem 4,2). When $\sqrt{n}\geq {L\over \mu}$, this term can not be ignored.
+ It seems that the difference between Algorithm 4 and the algorithms in [1] is not just in the gradient estimator. In [1], the extragradient requires two proximal operators in each iteration, while the FBF has different update orders in the iteration. Since the non-distributed case is also interesting, it would be better to have more discussion in the supplementary material.

**Limitations:**

+ The parameter for the algorithm requires the strong convexity constant, which may not be easy to obtain. It may limit this algorithm's application for many nonconvex machine learning problems.

**Strengths And Weaknesses:**

Strengths
+ The lower bound in the complexity is provided. In general, finding the lower bound is not easy.
+  Optimization algorithms are proposed to match the lower bound.

Weakness
+ The strong assumption may limit the vast application of the proposed algorithms. It requires strong monotonicity for the convergence of the proposed algorithms. Is it possible to show the convergence for general problems?

---

> ### Author Response · Authors · 2022-07-30
> **For Reviewer JEvK**
>
> We thank Reviewer **JEvK**  for review, time and the insightful comments, which will help to improve our work.
>
> We are glad that Reviewer liked our results. Next, we answer the questions and shortcomings noted by Reviewer.
>
> > **The strong assumption may limit the vast application of the proposed algorithms.** It requires strong monotonicity for the convergence of the proposed algorithms. Is it possible to show the convergence for general problems?
>
> 1) If we are talking about the general monotone case, we are sure that we can get the result. It is clear that we cannot give the whole proof here, therefore we propose to consider the classical regularization trick to make sure that results can be obtained.
> The essence of this trick is to make the convex minimization problem strongly convex, the convex-concave saddle point problem strongly convex - strongly concave, and the monotone VI strongly monotone. For example, we have the convex - concave saddle point problem
> $$
> \min_{x \in X} \max_{y \in Y} f(x,y).
> $$
> We can change the goal function and consider
> $$
> \tilde f(x,y) = f(x,y) + \frac{\varepsilon}{8D^2}||x - x_0 ||^2 -  \frac{\varepsilon}{8D^2} ||y - y_0 ||^2,
> $$
> where $x_0, y_0$ is a starting point, $\varepsilon$ is a desired solution accuracy,  $D = \max (D_X, D_Y)$ with $D_X, D_Y$ Euclidean diameters of $X$ and $Y$.
> The new problem is $\frac{\varepsilon}{4D^2}$ strongly convex - strongly - concave. It can be seen that both regularizers do not spoil the original problem much, because the sum does not exceed $\varepsilon/4$. If we solve the new problem with accuracy $\varepsilon/2$, then we solve the original problem with accuracy $\varepsilon$.
> Similarly, we can consider the monotone operator $F$, and instead of it use the strongly monotone operator
> $$
> \tilde F = F + \frac{\varepsilon}{4 D^2} (z - z_0).
> $$
> And based on these conclusions it is easy to obtain estimates of convergence in the monotone case, just substitute $\mu = \frac{\varepsilon}{4D^2}$ in our estimates from the paper.
>
> 2) If we are talking about non-monotone operators, it is a more complicated issue. This is due to the fact that there are no results for them in the literature. Usually, non-monotonicity is considered with an additional minty/variational stability condition. People also consider saddle point problems under PL condition or convex-nonconcave saddle point problems. These are interesting directions for future research, but here we still need to decide which of the setups is a higher priority.
>
> > **In the complexity, the term $n \log 1/e$  is not included in the computation**, and the term $\sqrt{n\chi} \log 1/e$  is not included in the communication (they are both in Theorem 4,2). When $n\geq L\mu$, this term can not be ignored.
>
> If we understand correctly, this comment is related to Table 1. We modified Table 1 in the revision. At first we wanted to add the full estimates to Table 1, but this made the line longer and we had to make the text in Table 1 very small. Then we added a footnote (6), where we specify that the complexities can contain additional factors. Please see the revision of our paper.
>
> > **It seems that the difference between Algorithm 4 and the algorithms in [1] is not just in the gradient estimator.** In [1], the extragradient requires two proximal operators in each iteration, while the FBF has different update orders in the iteration. Since the non-distributed case is also interesting, it would be better to have more discussion in the supplementary material.
>
> We are glad that Reviewer found more differences between the algorithms from [1] with our non-distributed algorithm. But it is important to note that in the first place, our algorithm should be compared not with Algorithms 1 (extragradient) or 2 (FBF), but with Algorithm 4 (FoRB) from [1]. It is Algorithm 4 that is closest to us. Comparison with Algorithm 4 is already present in Appendix B (see line 627). Therefore, we did not add anything, we think this is the most honest option in relation to the authors of [1].
>
> > **The parameter for the algorithm requires the strong convexity constant**, which may not be easy to obtain. It may limit this algorithm's application for many nonconvex machine learning problems.
>
> It seems that we answered this issue when we discussed the general analysis above.
>
>
> [1] Ahmet Alacaoglu and Yura Malitsky. Stochastic variance reduction for variational inequality methods.

---

> > ### Comment · Reviewer_JEvK · 2022-08-09
> > **Thanks for the response**
> >
> > Thanks for the response. As in my previous review, I think this paper has value, and I still think this paper can be accepted. Variational inequality has more applications than optimization.

---

> > > ### Author Response · Authors · 2022-08-09
> > > **Thank you!**
> > >
> > > We're glad to hear it! Thanks again for the review!

---

### Official Review · Reviewer_LbuJ · 2022-07-12

**Rating:** 5
**Confidence:** 2
**Soundness:** 3 good
**Presentation:** 2 fair
**Contribution:** 2 fair

**Summary:**

This paper studies decentralized stochastic variational inequalities (SVI). The authors present lower bounds for the communication and computation complexities of decentralized SVI. The authors construct new algorithms that achieve the optimal rates matching the lower bounds, for both fixed and time-varying networks.

**Questions:**

-

**Limitations:**

Yes

**Strengths And Weaknesses:**

This paper studies an interesting problem and the results seem solid and comprehensive. The writing and exposition can be improved to give more context and intuition to the readers.

---

> ### Author Response · Authors · 2022-07-30
> **For Reviewer LbuJ**
>
> We thank Reviewer **LbuJ**  for review, time and the insightful comments, which will help to improve our work.
>
> We are glad that Reviewer liked our results. But Reviewer noted that wanted to see more context and intuition. Unfortunately, it's hard for us to see which points we should clarify in more detail. We just note that we have Section 1.4 where we talk about related results. We also have Section B (Appendix), where we give a link between our methods and those known in the literature.
>
> We would be grateful if Reviewer gave us more detailed comments on what we can add about context and intuition and gives us an opportunity to improve the paper.

---

### Author Response · Authors · 2022-07-30
**Rebuttal Revision**

Dear Reviewers, Area Chairs and Senior Area Chairs!

We published a revision of our paper. We made some small changes and highlighted them in blue. In particular,

1) We added a footnote (6) to Table 1 at the request of Reviewer **JEvK**.

2) We added that $g$ is a proximal friendly function.

3) We added $\text{prox}$ in Definition 3.1 at the request of Reviewer **4cfY**.

4) We added some more references and ideas about lower bounds (see lines 209 and 651) at the request of Reviewer **4cfY**.

5) Fixed 3 typos (thanks, Reviewer **4cfY**).

Thank you very much for your work! You really helped make our paper better.

---

### Author Response · Authors · 2022-08-07
**A kind reminder about rebuttals**

With this message, we would just like to kindly remind Reviewers that we would be happy if Reviewers would participate in the rebuttal discussion process. We are looking forward to hearing from Reviewers **LbuJ** and **JEvK**. We thank Reviewer **4cfY** for the responses to the rebuttal, we are also looking forward to hearing from Reviewer **4cfY** in reply to our clarifying question.

---

### Meta-Review · Area_Chair_jEcj · 2022-08-25

**Recommendation:** Accept
**Confidence:** Certain

**Metareview:**

The paper makes a significant contribution to the literature on distributed SVIs. The results provided are fairly comprehensive -- both lower bounds and algorithms achieving the lower bounds are provided. Hence, the paper is recommended for acceptance.

**Award:**

No

---

### Decision · Program_Chairs · 2022-09-14

Accept